# Arp2/3 and type-I myosins control chromosome mobility and end-resection at double-strand breaks in *S. cerevisiae*

Felix Y. Zhou [1,2], Marissa Ashton[1], Yiyang Jiang [1], Neha Arora [1], Kevin Clark [1], Kate B. Fitzpatrick [1] & James E. Haber [1] ✉

Using budding yeast, we show that Arp2/3 actin branching complex has an evolutionarily conserved role in promoting chromosome mobility of double-strand breaks (DSBs). The radius of confinement of a broken chromosome is reduced by inhibiting Arp2/3 or by auxin-induced degron depletion of the nucleation promoting factor Las17WASP or type-1 myosins. Arp2/3 and Las17 are required both to initiate and maintain 5' to 3' resection of DSB ends, whereas depleting Myo3 or Myo5 impairs broken chromosome motion without affecting resection. Conversely, inhibiting Exo1- and Dna2-dependent long-range resection reduces DSB mobility. Inactivating Arp2/3 before DSB induction leads to shortened checkpoint arrest, activating the Tel1ATM/Mre11 (TM) checkpoint. Shortened checkpoint arrest, but not reduced broken chromosome mobility per se, results in reduced interchromosomal homologous recombination. These results suggest that regulating the Arp2/3 complex plays a key role in the processing of DSB ends that is correlated with an increase in DSB mobility and DSB repair.

Repair of chromosomal double-strand breaks (DSBs) is required to maintain genome integrity. Homologous recombination (HR) requires that the ends of the DSB find a homologous donor sequence to use as a template for repair (reviewed in refs. 1,2). In budding yeast, a DSB causes an increase in chromatin mobility near a DSB break site[3,4]; this increase is thought to facilitate the Rad51-mediated search for homology by increasing the volume that the DSB ends explore within the nucleus, defined as the radius of confinement ($R_C$)[5]. While processes such as the activation of the Mec1ATR-dependent DNA damage checkpoint, the phosphorylation of histone H2A, and recruitment of repair proteins to the break have been shown to be required for DSB-induced chromatin mobility[3,4], there is still no mechanism that explains how a DSB causes increased chromatin mobility or how changes in mobility affect DSB repair (reviewed in refs. 5,6).

Several studies in metazoans have suggested that actin branching is directly involved in repairing double-strand breaks. In mammalian cells and *Drosophila*, inactivation of the actin nucleator Arp2/3 complex by the drug CK-666 lowers DSB mobility[7,8]. Treatment of *Drosophila* cells with CK-666 or RNAi depletion of nuclear myosin proteins impaired the directed movement of damaged DNA in heterochromatin to the nuclear periphery[7]. In addition, CK-666 caused an apparent reduction in the amount of single-stranded DNA (ssDNA) formed at DSB ends[8]. A proteomics study in *Xenopus* egg extracts has shown that all seven subunits of the Arp2/3 complex and actin are enriched at DSBs[8].

The branching of polymerized actin is dependent on the Arp2/3 complex, hereby referred to as Arp2/3, and a set of nucleation-promoting factors (NPF) (reviewed by Goode et al.[9]). The Arp2/3 actin nucleator Wiskott–Aldrich syndrome protein (WASP)[10,11] has been reported to affect the loading of the single-strand binding protein RPA onto single-stranded DNA in mammalian cells[12] and mutations in Las17, the budding yeast homolog of WASP, were shown to be hypersensitive to DNA damaging agents hydroxyurea (HU) and methyl methanesulfonate (MMS)[12]. Las17 mutants accumulated more

[1]Rosenstiel Basic Medical Sciences Research Center and Department of Biology, Brandeis University, Waltham, MA, USA. [2]Present address: Department of Radiation Oncology, Dana-Farber Cancer Institute, Harvard Medical School, Boston, MA, USA. ✉e-mail: haber@brandeis.edu

Rad52 foci when exposed to HU or MMS, suggesting a deficiency in repair by HR[12].

Here, we have investigated the role of Arp2/3 and its associated NPFs in *S. cerevisiae* in more detail. In budding yeast, Arp2/3 and the NPF Las17 and the type-I myosins (Myo3 and Myo5) (reviewed in ref. [13]) are primarily found in cortical actin patches and are required for clathrin-mediated endocytosis[9]. These factors are well-studied in endocytosis[13], but their roles in other cellular processes remain poorly defined; however, a recent study has shown that removing both type-I myosins from the nucleus causes a dramatic alteration in nuclear architecture and chromatin conformations[14].

Here we have used a fluorescently labeled checkpoint protein, Ddc2-GFP, to mark DSBs[15] and employed mean-square displacement (MSD) analysis (reviewed in ref. [16]) to track and measure changes in mobility of an irreparable site-specific DSB at the *MAT* locus on chromosome III. We found that blocking Arp2/3 activity, either through the drug CK-666 or through deletion or depletion of NPFs, lowered DSB mobility. Moreover, 5′ to 3′ resection was reduced when Arp2/3 was inactivated after DSB formation. When Arp2/3 activity is blocked before DSB induction, the initiation of resection is severely impaired. Reciprocally, we show that eliminating long-range resection, through deleting the chromatin remodeler *FUN30* or inactivating both Exo1 and Dna2 nucleases, lowers DSB mobility. We conclude that Arp2/3 has an evolutionarily conserved role in regulating both DSB mobility and DSB end-resection, and that directly impairing the rate of resection prevents the increased motion of DSB ends. In addition, we show that a reduction in Rc itself does not impair ectopic homologous recombination, whereas a block in resection, reducing the length of DNA damage checkpoint arrest, does inhibit repair.

## Results

### Labeling and tracking the mobility of double-stranded breaks
To generate, track, and quantify the mobility of a single-irreparable DSB, we modified the well-characterized strain JKM179, in which a single site-specific cut in the *MAT* locus on chromosome III is induced by a galactose-regulated HO endonuclease gene (Gal-HO)[17]. To visualize and track the DSB, we monitored the DNA damage checkpoint protein Ddc2 (Ddc2-GFP) (Fig. 1A), which we have previously shown forms a damage-dependent GFP focus in ~80% of cells 3 h after adding galactose[15]. An mCherry-tagged spindle-pole body protein Spc42 (Spc42-mCherry) was used as a fiducial marker to account for random nuclear motion. Cutting by Gal-HO is highly efficient, with nearly 90% of cells being cleaved 1 h after galactose addition[17–19]. In these strains, the homologous *HML* and *HMR* donors are deleted, thus preventing repair of *MAT* by HR. When HO is continually expressed, repair of the DSB by altering the cleavage site via nonhomologous end-joining occurs in only 0.2% of cells[20]. Since the percentage of cells with a Ddc2-GFP focus reached saturation (~80% of the population) 3 h after adding galactose, we measured DSB mobility 3 h after adding galactose and quantified the mobility of the DSB using mean-squared displacement (MSD) analysis (see "Methods" section for details) (Fig. 1B, C).

### Blocking Arp2/3 activity lowers DSB mobility
To test whether Arp2/3 activity has a conserved role in DSB mobility in budding yeast, we added 100 μM CK-666 to block Arp2/3 activity 20 min before imaging (160 min after adding galactose) (Fig. 1B, C). 100 μM of CK-666 is sufficient to block Arp2/3 activity in budding yeast[21]. CK-689 was used as an inactive control for Arp2/3 inhibition that binds to Arp2/3 without inhibiting Arp2/3 activity[22,23]. We found that when CK-689 (100 μM) was added to cells 20 min before imaging, it did not affect the mobility of the DSBs (Fig. 1C, D, and Supplementary Data 1). Addition of CK-666 significantly lowered the radius of confinement (Rc) of the DSB from 0.98 μm to 0.62 μm (Fig. 1D and Supplementary Data 1). The addition of CK-666 did not affect the mobility of the spindle-pole bodies relative to the bud neck (Supplementary

Fig. 1A, B, Supplementary Data 1), suggesting that the changes seen in DSB mobility after CK-666 treatment are due to the changes in the behavior of the DSB.

Since Arp2/3 is primarily known as a component of endocytosis in budding yeast, we asked whether the change in DSB mobility was due to impaired endocytosis. Deletion of *SLA2*, a key component of endocytosis[24], did not significantly affect the behavior of the DSB: Rc WT (0.87 μm) and *sla2Δ* (0.84 μm) (Fig. 1E, F, and Supplementary Data 1). This result suggests that endocytosis is not required for DSB mobility and that the role of Arp2/3 in DSB mobility is independent of its role in endocytosis.

### Arp2/3 affects changes in local and global chromatin mobility in response to a DSB
While the direct labeling of DSBs with Ddc2-GFP is suitable for tracking the mobility of DSBs, it cannot be used to establish a basal level of chromatin mobility in the absence of a DSB. To test whether Arp2/3 is required for increasing chromatin mobility near a DSB, we induced an irreparable DSB at *MAT* in a derivative of strain JKM179 that expresses the GFP-LacI protein, which binds to a *lacO* array inserted 4.4 kb away from the *MAT* locus[25] (Fig. 2A). As previously reported[25], we found that inducing a DSB caused an increase in local chromatin mobility near the DSB site: R_C 0.67 μm (uncut) and 1.1 μm (cut) (Fig. 2B, F, and Supplementary Data 1). Treating cells with CK-666 for 20 min did not significantly affect basal levels of chromatin mobility (Fig. 2B, F and Supplementary Data 1); however, treatment with CK-666 either 20 min before or 160 min after (Fig. 2C) DSB induction prevented increased DSB mobility, monitored at 180 min (Fig. 2D–F and Supplementary Data 1); however, as we show below, interfering with Arp2/3 activity before or after DSB induction had different effects on DSB repair processes. Thus, CK-666 impairs DSB-induced chromatin mobility.

### Nucleation-promoting factors are required DSB-induced increase in mobility
As in humans, yeast Arp2/3 requires the NPF WASP to nucleate actin branching[10,11]. Budding yeast Arp2/3 NPFs include Las17, the homolog of WASP[26], and the type-I myosins Myo3 and Myo5[13], which are known to interact with each other at pre-endocytic patches through verprolin (Vrp1)[13] (Fig. 3A). To test whether Las17 was required for DSB mobility, we added an auxin-inducible degron (AID) to the C-terminus of Las17 (Las17-AID)[27,28]. Adding 1 mM auxin (IAA) caused degradation of Las17-AID within 1 h (Fig. 3B). The AID tag did not affect the MSD curve of Las17-AID, but when IAA was added 2 h after HO induction, the mobility of the DSB measured at 3 h dropped significantly: Rc WT (0.76 μm), Las17-AID (0.79 μm), and Las17-AID + IAA (0.53 μm) (Fig. 3C, D, Supplementary Data 1).

To further characterize the role of Las17, we deleted functional domains that are responsible for Arp2/3 activation, specifically the acidic patch (CA) domain which directly binds to Arp2/3 to activate Arp2/3 and the WH2 (Wiskott–Aldrich syndrome homology region 2) domain which brings monomeric actin to Arp2/3[29] (Fig. 3A). A deletion of just the CA domain (*las17-CAΔ*) had no effect on the MSD and Rc of the DSB, but a deletion of both domains (*las17-WH2-CAΔ*) significantly lowered the mobility of the DSBs: R_C WT (0.95 μm), *las17-CAΔ* (0.90 μm), and R_C *las17-WH2-CAΔ* (0.53 μm) (Fig. 3E, F, Supplementary Data 1). Moreover, a deletion of the WH2 domain (*las17-WH2Δ*) significantly lowered the mobility of the DSBs: Rc *las17-WH2Δ* (0.5 μM) (Fig. 3E, F, Supplementary Data 1). This result suggests that Las17, as an Arp2/3 actin nucleator, and especially its WH2 domain, is required for DSB mobility.

### The type-I myosins Myo3 and Myo5 play a role in DSB mobility
Type-I myosins Myo3 and Myo5 promote actin nucleation by Arp2/3 during endocytosis in budding yeast[30] and depletion of the type-I myosins Myo1a and Myo1b in *Drosophila* increases sensitivity to

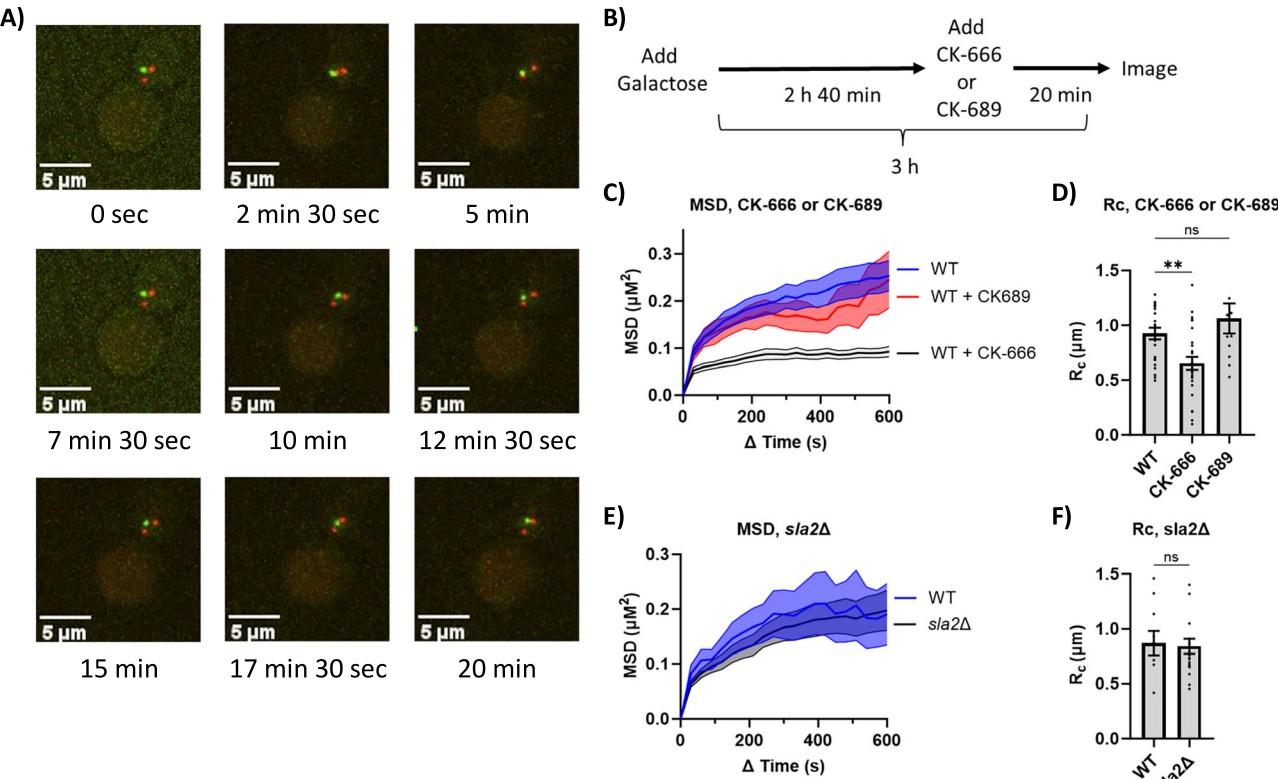

**Fig. 1 | Inhibiting Arp2/3 activity lowers DSB mobility. A** Stills from a 20-min time course of a budding yeast cell 3 h after DSB induction with Gal-HO. Green focus is Ddc2-GFP, and red foci are Spc42-mCherry. **B** Diagram of experimental setup to measure the MSD of a DSB labeled with Ddc2-GFP. CK-666 (100 μM) or CK-689 (100 μM) was added 2 h 40 min after adding galactose. **C** Mean-squared displacement (MSD) analysis (Δt = 30 s) of the DSB relative to the SPB in FZ015 (WT) strain (n = 23), WT strain with CK-689 (100 μM) (n = 11), and WT strain with CK-666 (100 μM) (n = 28) 3 h after galactose induction of an HO endonuclease-induced irreparable DSB at the *MAT* locus on chromosome III. Resected DSBs were bound by Ddc2-GFP, and the SPBs were labeled with Spc42-mCherry. Cells were imaged with a 488 nm and 561 nm laser, taking optical stacks of 1.5 μm with a step size of 300 nm every 30 s for 20 min. See "Methods" section for MSD analysis. Mean ± SEM is shown. Source data are provided as a Source Data file. **D** Radius of confinement (Rc) from MSD analysis of strains in (**C**). Statistical analysis for the radius of confinement derived in PRISM using a one-way Anova (ns p ≥ 0.05, *p < 0.05, **p < 0.01, and ***p < 0.001). WT vs. WT + CK-666 (p = 0.0082) and WT vs. WT + CK-689 (p = 0.4112). See Supplementary Data 1 and 2 for significance and $R_C$ values. Mean ± SEM is shown. Source data are provided as a Source Data file. **E** MSD analysis (Δt = 30 s) of the DSB relative to the SPB in FZ015 (n = 10) and FZ077 (*sla2Δ*) (n = 20) 3 h after Gal-HO induction. Imaging and analysis done as described in (**C**). Mean ± SEM is shown. Source data are provided as a Source Data file. **F** Rc from MSD analysis of strains in (**E**). Statistical analysis was done with a two-sided unpaired t-test. WT vs. *sla2Δ* (p = 0.8251). See Supplementary Data 1, 2 for significance and $R_C$ values. Mean ± SEM is shown. Source data are provided as a Source Data file.

ionizing radiation[7,31]. We found that a single deletion of Myo3 or Myo5 was sufficient to lower the mobility of the DSB: Rc WT (0.86 μm), *myo3Δ* (0.60 μm), and *myo5Δ* (0.6 μm) (Fig. 4A, F, and Supplementary Data 1). Using the GFP-LacI *lacO::MAT* strain, we showed that deletion of either Myo3 or Myo5 did not affect basal chromatin mobility in the absence of a DSB: Rc WT (0.78 μm), *myo3Δ* (0.74 μm), and *myo5Δ* (0.77 μm) (Fig. 4B, C, and Supplementary Data 1). The effect of deleting either type-I myosin was different from what is seen in endocytosis, where Myo3 and Myo5 are functionally redundant[32].

We then asked whether a second copy of one type-I myosin, added to a deletion of the other, would restore DSB mobility. We integrated a second copy of *MYO5* at *URA3* on chromosome V, under control of its own promoter, in a *myo3Δ* background and found that there was no significant difference in the Rc between the WT (0.86 μm) and the *myo3Δ MYO5* (+*MYO5*) (0.85 μm) strains. (Fig. 4D, F, Supplementary Data 1). Moreover, adding a second plasmid copy of *MYO3* with its endogenous promoter in a *myo5Δ* strain yielded similar results: Rc WT (0.86 μm) and *myo5Δ MYO3* (+*MYO3*) (0.9 μm) (Fig. 4E, F, Supplementary Data 1).

To determine which functional domains of Myo5 (Fig. 3A) are required for DSB mobility, we integrated a series of Myo5-domain deletion plasmids with their endogenous promotors[30] into a *myo5Δ* strain. In the presence of Myo3, we found that there was no significant difference in the $R_C$ between the wild-type strain and a deletion of the

Myo5 motor-domain, TH1, or the TH2 domain: WT (0.89 μm), *myo5-motorΔ* (0.86 μm), *myo5-TH1Δ* (0.85 μm), and *myo5-TH2Δ* (0.88 μm) (Supplementary Fig. 2A–C, E and Supplementary Data 1). However, a deletion of the SH3 domain (*myo5-SH3Δ*) did not restore the mobility of the DSBs back to wild-type levels; $R_C$ WT (0.89 μm) and *myo5-SH3Δ* (0.62 μm) (Supplementary Fig. 2D, E, Supplementary Data 1).

Because a double deletion of *MYO3* and *MYO5* is synthetically lethal in most backgrounds[33,34], we added an AID tag to Myo5 (Myo5-AID) in a *myo3Δ* strain. In a *MYO3* deletion strain, the addition of the AID tag to Myo5 did not affect the mobility of the DSB, and adding IAA 2 h after DSB induction reduced mobility, similar to the *MYO5* deletion. (Supplementary Fig. 3A–C, Supplementary Data 1). Unexpectedly, in the *myo3Δ* Myo5-AID strain, treatment with IAA increased the mobility of the DSB to near WT levels: Rc WT (0.81 μm), *myo3Δ* Myo5-AID (0.61 μm), and *myo3Δ* Myo5-AID + IAA (0.81 μm) (Fig. 4G, H, Supplementary Data 1). This unexpected result is likely a reflection of a critical role that type-I myosins play in budding yeast nuclear architecture. A recent study has shown that depleting the nucleus of both type-I myosins caused a large change in nuclear organization: centromeres became de-clustered and chromosome folding was perturbed[14]. We confirmed that there are significant changes in the behavior of the yeast nucleus when both type-I myosins were depleted, by measuring the mobility of the SPBs relative to the bud neck. SPB mobility was unaffected when *myo5* was deleted: Rc WT (0.95 μm) and *myo5Δ* (0.85 μm) (Supplementary Fig. 4A, B,

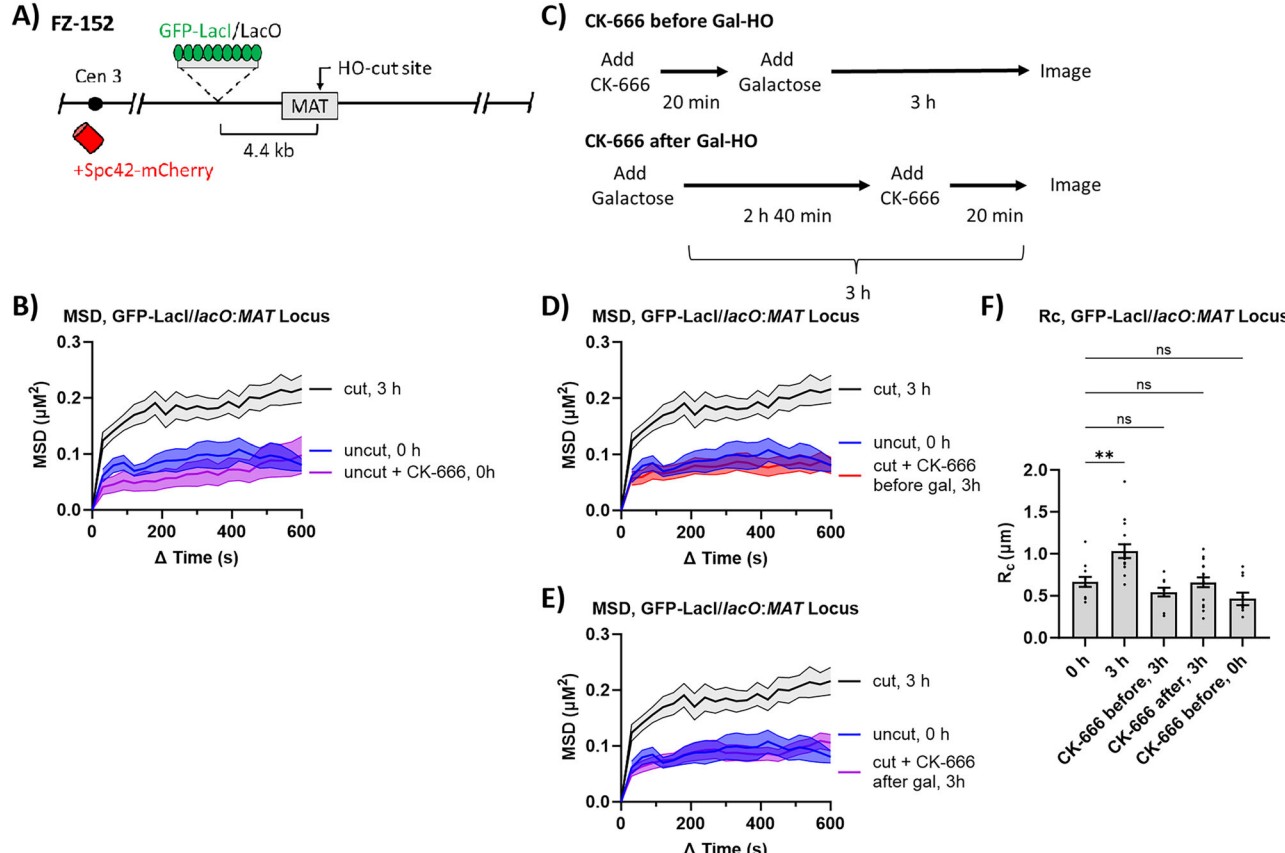

**Fig. 2 | Arp2/3 is required for DSB-induced chromatin mobility. A** GFP-tagged chromosome locus in FZ152. *LacO*-array is 4.4 kb away from a Gal-HO inducible DSB in the MAT locus on chromosome III labeled with GFP-LacI. SPB is labeled with Spc42-mCherry. **B** MSD analysis (Δt = 30 s) of the *MAT* locus relative to the SPB before Gal-HO induction (uncut, 0 h) (n = 12), treatment with CK-666 (100 μM) before Gal-HO induction (uncut + CK-666, 0 h) (n = 10), and 3 h after Gal-HO induction (cut, 3 h) (n = 15). Imaging and analysis done as described in Fig. 1C. Mean ± SEM is shown. Source data are provided as a Source Data file. **C** Experimental setup to measure the MSD of a GFP-LacI/*lacO* array 4.4 kb away from the HO-cut site on chromosome III. CK-666 was added 20 min before adding galactose, and MSD was measured 3 h after DSB induction (CK-666 before Gal-H), or CK-666 was added 2 h 40 min after galactose and MSD was measured 3 h after DSB induction (CK-666 after Gal-HO). **D** MSD analysis (Δt = 30 s) of the *MAT* locus relative to the SPB before Gal-HO induction (uncut, 0 h) (n = 12), 3 h after Gal-HO induction where CK-666 (100 μM) was added 20 min before galactose (cut + CK-666 before gal, 3 h) (n = 10), and 3 h after Gal-HO induction (cut, 3 h) (n = 15).

Imaging and analysis done as described in Fig. 1C. Mean ± SEM is shown. Cut and uncut data from (**B**) is shown again as a comparison for CK-666 treatment. Source data are provided as a Source Data file. **E** MSD analysis (Δt = 30 s) of the *MAT* locus relative to the SPB before Gal-HO induction (uncut, 0 h) (n = 12), 3 h after Gal-HO induction with CK-666 (100 μM) (cut + CK-666 after gal, 3 h) (n = 17), and 3 h after Gal-HO induction (cut, 3 h) (n = 15). Cells treated with CK-666 were given CK-666 20 min before imaging. Imaging and analysis done as described in Fig. 1C. Mean ± SEM is shown. Cut and uncut data from (**B**) is shown again as a comparison for CK-666 treatment. Source data are provided as a Source Data file. **F** Rc from MSD analysis of strains in (**B**), (**D**), and (**E**). Statistical analysis for the radius of confinement derived in PRISM using a one-way Anova (ns p ≥ 0.05, *p < 0.05, **p < 0.01, and ***p < 0.001). 0 h vs. 0 h + CK-666 (p = 0.18), 0 h vs. 3 h (p = 0.0012), 0 h vs. 3 h pre-CK-666 (p = 0.6039), and 0 h vs. 3 h post-CK-666 (p > 0.999). See Supplementary Data 1, 2 for significance and R_C values. Mean ± SEM is shown. Source data are provided as a Source Data file.

Supplementary Data 1). However, when both type-I myosins were depleted post-DSB induction, there was an increase in the mobility of the SPBs: Rc WT (0.93 μm), *myo3Δ* Myo5-AID (0.91 μm), and *myo3Δ* Myo5-AID (1.14 μm) (Supplementary Fig. 4C, D, Supplementary Data 1). These changes in nuclear architecture and spindle-pole mobility could account for the increase in DSB mobility seen in Fig. 4G, H, and therefore are likely not a reflection of a return to normal chromatin mobility when both myosins are absent.

### Arp2/3 and type-I myosins are required for damage-dependent focus formation with Ddc2-GFP or Rad51-GFP

Ddc2-GFP and Rad51-GFP form damage-dependent foci in ≥80% of cells 3 h after DSB induction[15]. However, when Arp2/3 activity was blocked by CK-666 20 min before DSB induction, we observed that Ddc2-GFP and Rad51-GFP formed foci in fewer than 15% of cells (Supplementary Fig. 5A–C). Similar inhibition of the DSB-induced foci was found in a deletion of *VRP1*, the protein that facilitates Las17

interactions with Myo3 and Myo5 (Supplementary Fig. 5B), or when IAA was added either to Las17-AID or *myo3Δ* Myo5-AID cultures.

To rule out that blocking Arp2/3 might interfere with either Gal-HO cutting (Supplementary Fig. 6A), we used a PCR assay to show that Gal-HO cutting was unaffected in Las17-AID or *myo3Δ* Myo5-AID mutants treated 1 h before DSB induction with IAA (Supplementary Fig. 6B–D). One known block to 5′ to 3′ resection in budding yeast is through the binding of the Yku70/Yku80 dimer in G_1 cells[35]. We asked if Ku might be implicated in preventing resection when Arp2/3 or Las17 were inhibited, but deletion of *KU70* did not restore damage-dependent Ddc2-GFP foci formation in CK-666-treated cells (Supplementary Fig. 5D).

### Arp2/3 and type-I myosins are required for both the initiation and the maintenance of resection

Since Gal-HO cutting is normal, blocking Arp2/3 activity might interfere with 5′ to 3′ resection of the DSB. We first monitored the

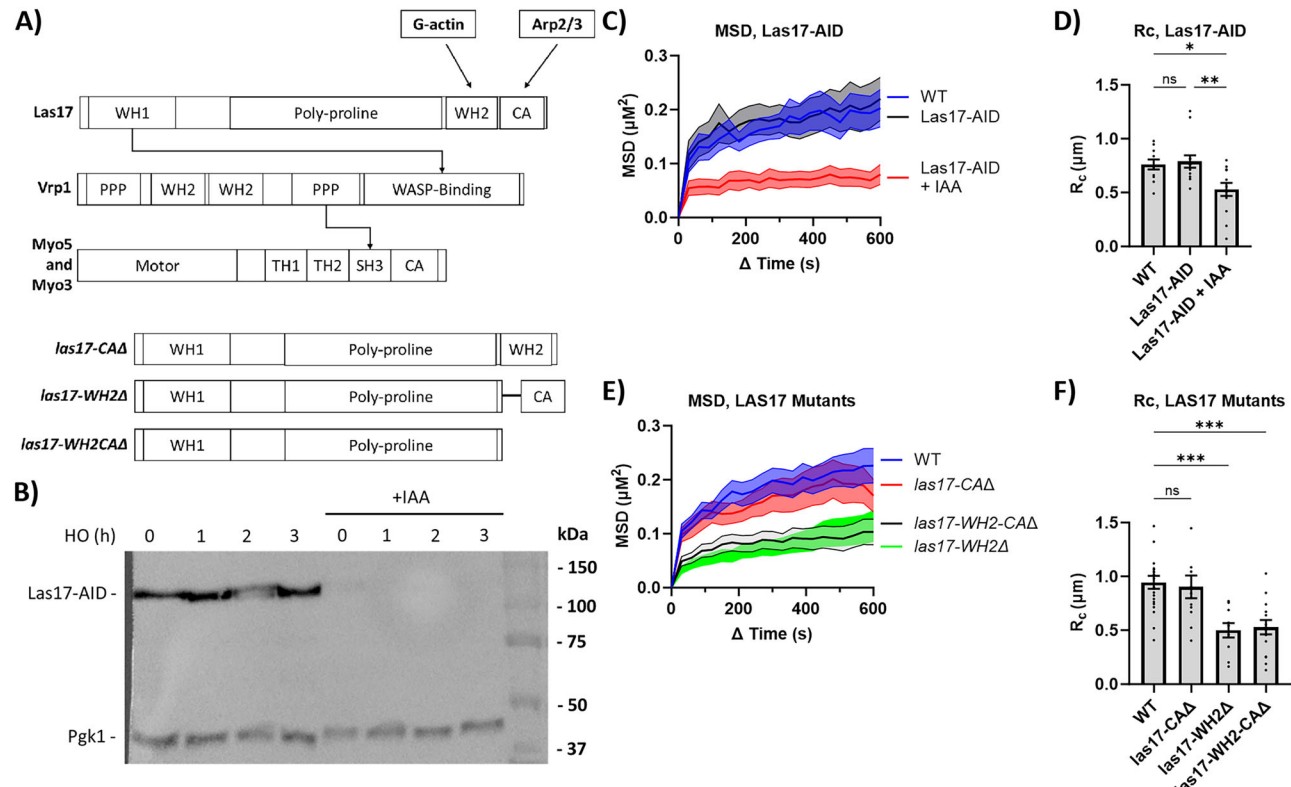

**Fig. 3 | Degradation of Las17 lowers DSB mobility. A** Functional domains of Las17, Vrp1, Myo3, and Myo5. Arrows indicate known interactions between these proteins. Not to scale. **B** Western blot analysis of Las17-AID ± auxin (IAA) (1 mM). IAA or an equivalent amount of 100% ethanol was added 1 h before adding galactose. Mouse α-Myc probed for Las17-AID. Mouse α-Pgk1 was probed as a loading control. Source data are provided as a Source Data file. **C** MSD analysis (Δt = 30 s) of DSB relative to the SPB in FZ015 (n = 11), FZ075 (Las17-AID) (n = 13), and FZ075 with IAA (1 mM) added 2 h after galactose (Las17-AID + IAA) (n = 13), 3 h after Gal-HO induction. Imaging and analysis done as described in Fig. 1C. Mean ± SEM is shown. Source data are provided as a Source Data file. **D** Rc from MSD analysis of strains in (**C**). Statistical analysis for the radius of confinement derived in PRISM using a one-way Anova (ns p ≥ 0.05, *p < 0.05, **p < 0.01, and ***p < 0.001). WT vs. Las17-AID (p = 0.9381), WT vs. Las17-AID + IAA (p = 0.0211), and Las17-AID vs. Las17-AID + IAA (p = 0.006). See Supplementary Data 1, 2 for significance and $R_C$ values. Mean ± SEM is shown. Source data are provided as a Source Data file. **E** MSD analysis (Δt = 30 s) of DSB relative to the SPB in FZ015 (n = 22), FZ099 (*las17-CAΔ*) (n = 12), FZ219 (*las17-WH2Δ*) (n = 10), and FZ100 (*las17-WH2-CAΔ*) (n = 15) 3 h after Gal-HO induction. Imaging and analysis done as described in Fig. 1C. Mean ± SEM is shown. Source data are provided as a Source Data file. **F** $R_C$ from MSD analysis of strains in (**E**). Statistical analysis for the radius of confinement derived in PRISM using a one-way Anova (ns p ≥ 0.05, *p < 0.05, **p < 0.01, and ***p < 0.001). WT vs. *las17-WH2-CAΔ* (p = 0.0002), WT vs. *las17-WH2Δ* (p = 0.0004), and WT vs. *las17-CAΔ* (p = 0.9693). See Supplementary Data 1, 2 for significance and $R_C$ values. Mean ± SEM is shown. Source data are provided as a Source Data file.

disappearance of the GFP-LacI/*lacO* focus in a strain where the *lacO* array was inserted 4.4 kb away from the HO cleavage site at *MAT* (Fig. 2A). As resection erodes the *lacO* array, GFP-LacI will lose its binding sites and the GFP-LacI/*lacO* focus will vanish. Beginning within 2 h after DSB induction, there was a steady drop in the percentage of cells with a GFP-LacI focus, down to 15% by 4 h (Supplementary Fig. 7A–C). However, the addition of CK-666 one hour before HO induction prevented the loss of the GFP-LacI focus, with 81% of cells retaining a GFP-LacI focus 4 h after DSB induction. In cells treated with CK-666 one hour after HO induction, there was an initial drop in the percentage of cells with a GFP-LacI focus, but 58% retained a GFP at 4 h (Supplementary Fig. 7A).

We used a restriction enzyme-based qPCR resection assay to measure the generation of ssDNA around the HO-cut site itself[36,37]. qPCR analysis was done using a series of primers flanking *Sty*I sites at different distances from the HO-cut site in the *MAT* locus (0.7 kb, 5 kb, and 10 kb) (Fig. 5A). As resection converts dsDNA to ssDNA, *Sty*I cleavage sites are lost, and PCR across the restriction site increases. We measured resection in Las17-AID, *myo3Δ*, *myo5Δ*, and *myo3Δ* Myo5-AID mutants. IAA was added either 1 h before or 2 h after adding galactose. Las17-AID without IAA had a similar resection profile as the wild type; however, when IAA was added either before or after galactose, long-range resection was severely impaired (Fig. 5B and Supplementary Data 3), suggesting that Las17 is required for the initiation and

maintenance of resection. To determine if there was a significant change in resection, we compared the amount of resection ($F_{resected}$) measured 6 h after DSB induction at sites 0.7 kb, 5 kb, and 10 kb away from the *MAT* locus (Fig. 5A) between the wild-type and the mutant conditions (Supplementary Data 3).

Single deletions of *MYO3* or *MYO5* did not greatly affect resection (Fig. 5C and Supplementary Data 3). Resection in the *myo3Δ* Myo5-AID mutant without IAA was comparable to resection in the WT, as expected from the *myo3Δ* resection data (Fig. 5C, D); but when IAA was added to *myo3Δ* Myo5-AID 1 h before DSB induction, there was little to no resection up to 6 h after DSB induction (Fig. 5D and Supplementary Data 3). Finally, when IAA was added 2 h after DSB induction, resection in *myo3Δ* Myo5-AID appeared to be greatly inhibited within an hour of degrading Myo5 (Fig. 5D and Supplementary Data 3). Together, these data suggest that Arp2/3 and type-I myosins have a role both in the initiation of resection and in its maintenance, although the general disruption of nuclear architecture[14] may also affect this measurement.

### Inhibition of resection lowers the mobility of DSBs

Since resection was affected by changes in Arp2/3 activity, we asked whether there was a correlation between the rate of resection and DSB mobility. Previously, we and others have shown that a deletion of the chromatin remodeler *FUN3O* greatly reduced Exo1 and Sgs1/Dna2-dependent long-range resection[38–40]. Although resection through the

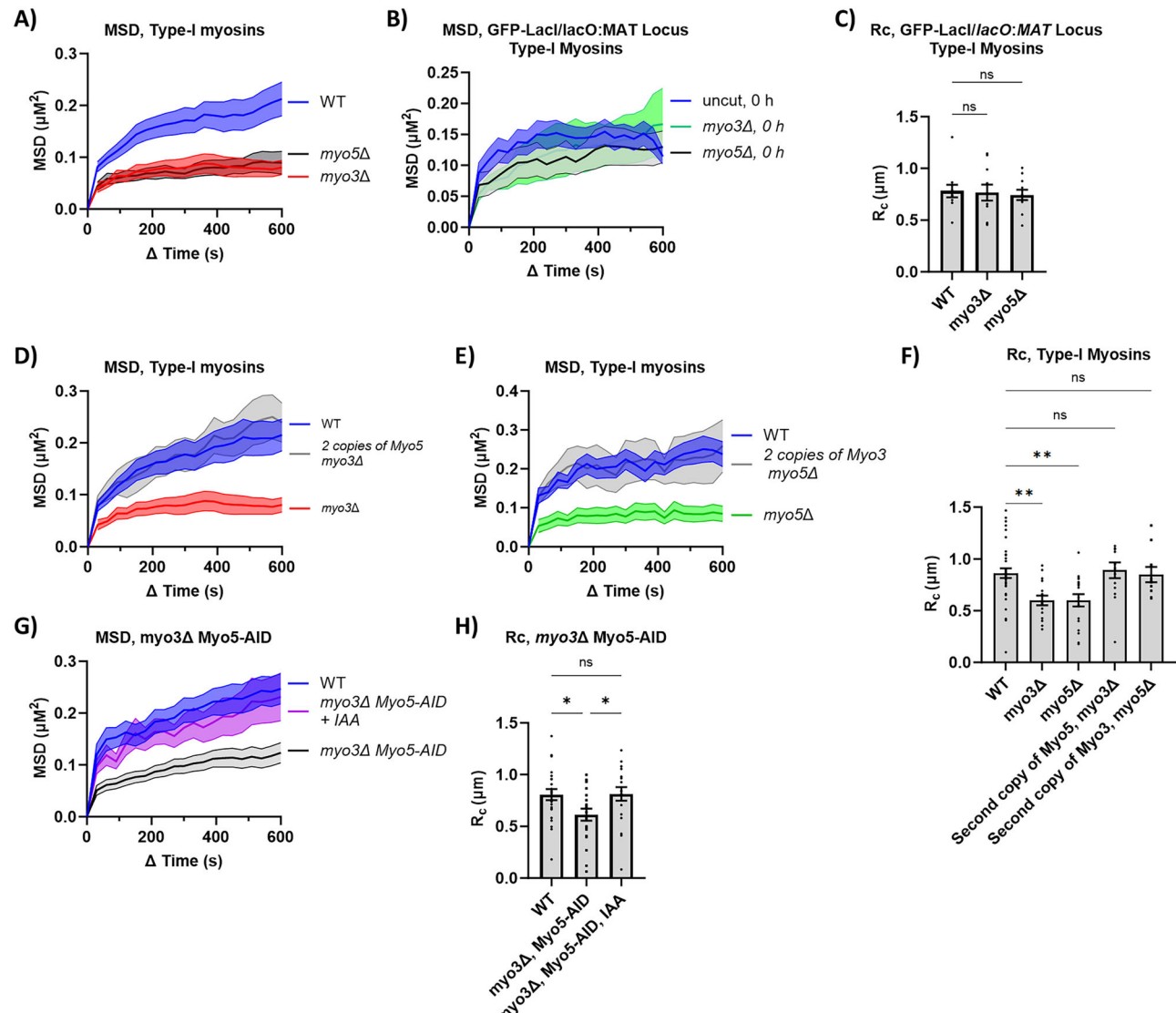

**Fig. 4 | The role of Myo3 and Myo5 in DSB mobility. A** MSD analysis (Δt = 30 s) of DSB relative to the SPB in FZ015 (WT) (n = 21), FZ033 (*myo3Δ*) (n = 16), and FZ034 (*myo5Δ*) (n = 18) 3 h after Gal-HO induction. Imaging and analysis done as described in Fig. 1C. Mean ± SEM is shown. Source data are provided as a Source Data file. **B** MSD analysis (Δt = 30 s) of basal mobility of the *MAT* locus relative to the SPB in FZ152 (WT) (n = 11), FZ215 (*myo5Δ*) (n = 11), and FZ216 (*myo3Δ*) (n = 11). Imaging and analysis done as described in Fig. 1C. Mean ± SEM is shown. Source data are provided as a Source Data file. **C** $R_C$ from MSD analysis of strains in (**B**). Statistical analysis for the radius of confinement derived in PRISM using a one-way Anova (ns p ≥ 0.05, *p < 0.05, **p < 0.01, and ***p < 0.001). WT vs. *myo5Δ* (p = 0.9782) and WT vs. *myo3Δ* (p = 0.8783). See Supplementary Data 1, 2 for significance and $R_C$ values. Mean ± SEM is shown. Source data are provided as a Source Data file. **D** MSD analysis (Δt = 30 s) of the DSB in FZ015 (WT) (n = 21), FZ033 (*myo3Δ*) (n = 16), and FZ047 (2 copies of Myo5 *myo3Δ*) (n = 15) 3 h after Gal-HO induction. Imaging and analysis done as described in Fig. 1C. Mean ± SEM is shown. Source data are provided as a Source Data file. **E** MSD analysis (Δt = 30 s) of the DSB in FZ015 (WT) (n = 12), FZ034 (*myo5Δ*) (n = 18), and FZ220 (2 copies of Myo3 *myo5Δ*) (n = 10) 3 h after Gal-HO

induction. Imaging and analysis done as described in Fig. 1C. Mean ± SEM is shown. Source data are provided as a Source Data file. **F** Rc from MSD analysis of strains in (**A**), (**D**), and (**E**). Statistical analysis for the radius of confinement derived in PRISM using a one-way Anova (ns p ≥ 0.05, *p < 0.05, **p < 0.01, and ***p < 0.001). WT vs. *myo5Δ* (p = 0.0048), WT vs. *myo3Δ* (p = 0.0033), WT vs. *myo3Δ*, second copy of Myo5 (p = 0.9933), and WT vs. *myo5Δ*, second copy of Myo3 (p = 0.9998). See Supplementary Data 1, 2 for significance and $R_C$ values. Mean ± SEM is shown. Source data are provided as a Source Data file. **G** MSD analysis (Δt = 30 s) of DSB relative to the SPB in FZ015 (WT) (n = 24), FZ074 (*myo3Δ* Myo5-AID) (n = 21), and FZ074 (*myo3Δ* Myo5-AID) + IAA (1 mM) (n = 19) 3 h after Gal-HO induction. Imaging and analysis done as described in Fig. 1C. Mean ± SEM is shown. Source data are provided as a Source Data file. **H** Rc from MSD analysis of strains in (**G**). Statistical analysis for the radius of confinement derived in PRISM using a one-way Anova (ns p ≥ 0.05, *p < 0.05, **p < 0.01, and ***p < 0.001). WT vs. *myo3Δ*, Myo5-AID (p = 0.0202), *myo3Δ*, Myo5-AID vs. *myo3Δ*, Myo5-AID + IAA (p = 0.0234), and WT vs. *myo3Δ*, Myo5-AID + IAA (p = 0.939). See Supplementary Data 1, 2 for significance and $R_C$ values. Mean ± SEM is shown. Source data are provided as a Source Data file.

---

0.7 kb site was not affected in *fun30Δ*, long-range resection past the 5 and 10 kb sites was markedly reduced (Fig. 6A and Supplementary Data 3). MSD analysis of a DSB showed the $R_C$ of *fun30Δ* (0.52 μm) was significantly lower than the WT (0.76 μm) (Fig. 6B, C, Supplementary Data 1).

Long-range resection requires both the exonuclease Exo1 and the helicase/endonuclease Sgs1-Top3-Rmi-Dna2 complex[41,42]. Since the

rate of resection is not affected by deleting either *EXO1* or *SGS1* alone[40], we blocked long-range resection by deleting *EXO1* and degrading Dna2-AID. There was no difference in DSB mobility between the wild type and *exo1Δ* Dna2-AID, but when IAA was added 2 h after DSB induction, the mobility of the DSB measured at 3 h was reduced: Rc WT (0.85 μm), *exo1Δ* DNA2-AID (0.82 μm), and *exo1Δ* DNA2-AID + IAA (0.55 μm). (Fig. 6D, E, Supplementary Data 1).

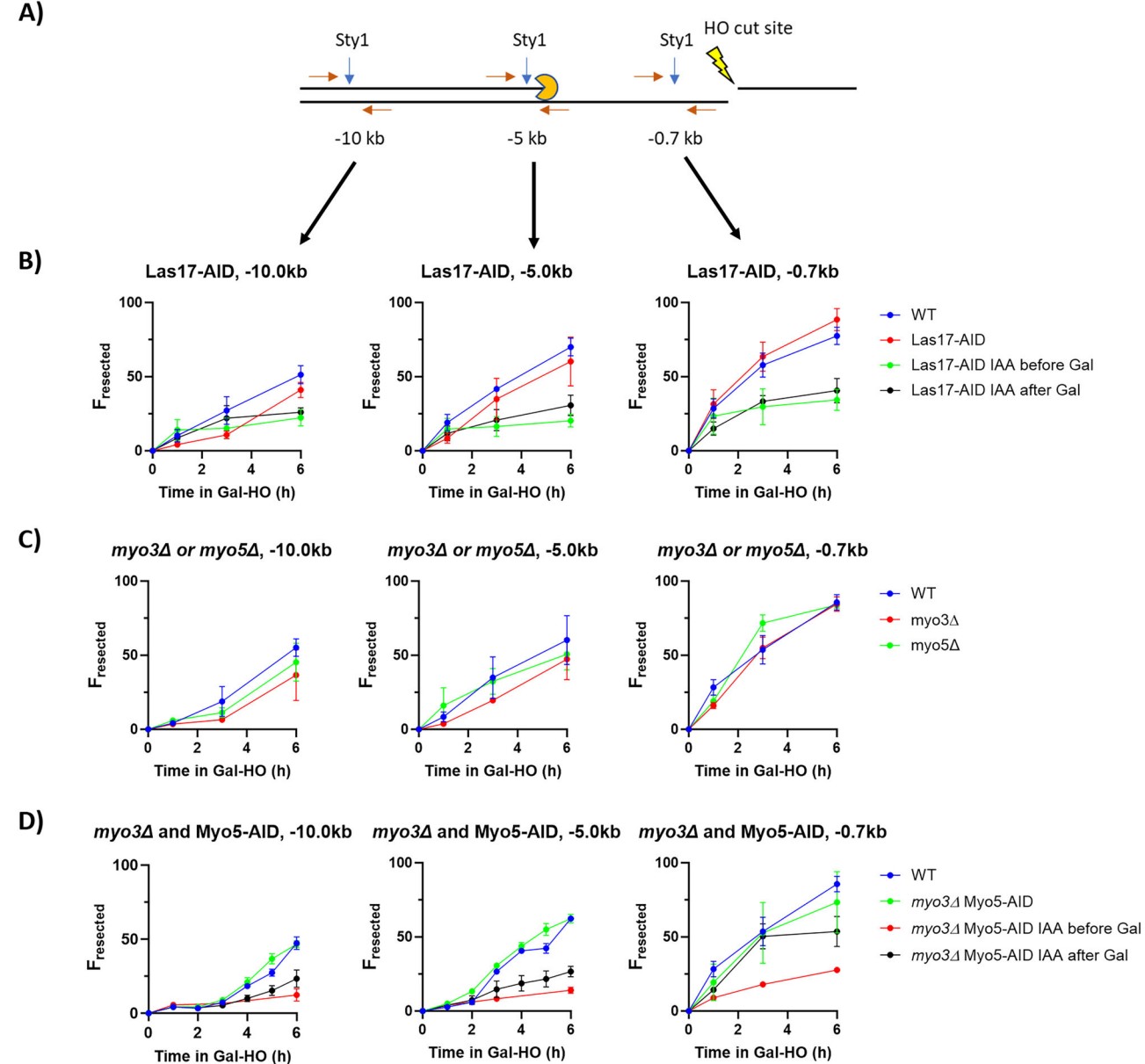

**Fig. 5 | Arp2/3 is required to initiate resection. A** qPCR-based resection assay. Pairs of primers are placed 10 kb, 5 kb, and 0.7 kb upstream of the HO-cut site. Between each set of primers is a *Sty*I cleavage site. *Sty*I is unable to cut ssDNA. Resection analysis was done from a minimum of 3 biological replicates collected in triplicate. See "Methods" section for more details. **B** Resection analysis of FZ015 (WT) and FZ075 (Las17-AID). IAA was added either 1 h before galactose or 2 h after galactose. Samples were collected at 0, 1, 3, and 6 h after adding galactose. Mean ± SEM is shown. Source data are provided as a Source Data file. **C** Resection analysis of FZ015 (WT), FZ033 (*myo3Δ*), and FZ034 (*myo5Δ*). IAA was added either 1 h before galactose or 2 h after galactose. Samples were collected at 0, 1, 3, and 6 h after adding galactose. Mean ± SEM is shown. Source data are provided as a Source Data file. **D** Resection analysis of FZ015 (WT) and FZ074 (*myo3Δ* Myo5-AID). IAA was added either 1 h before galactose or 1 h after galactose. Samples were collected at 0, 1, 2, 3, 4, 5, 6 h after adding galactose. Mean ± SEM is shown. Source data are provided as a Source Data file.

To test whether increasing resection affected DSB mobility, we measured mobility in a strain with a second copy of *EXO1* expressed under the control of a galactose promoter (pGal:*EXO1*) integrated at the *LEU2* locus[43]. Overexpression of *EXO1* increased resection (Fig. 7A and Supplementary Data 3), but we did not find a difference between the Rc in WT (0.78 μm) and when Exo1 (0.69 μm) was overexpressed (Fig. 7C, D and Supplementary Data 1). Interestingly, pGal::*EXO1* suppressed the reduction in Rc caused when CK-666 was added 1 h after HO induction (0.89 μm) (Fig. 7B–D, Supplementary Data 1), but *EXO1* overexpression did not rescue Ddc2-GFP focus formation in cells treated with CK-666 before DSB induction (Supplementary Fig. 5E). Thus CK-666 appears to block a step prior to the initiation of Exo1-dependent resection.

To further explore the relationship between Arp2/3 activity and Exo1 overexpression, we examined resection and DSB mobility in a pGal:*EXO1* Las17-AID double mutant. Without IAA, the pGal:*EXO1* Las17-AID double mutant saw a greater rate of resection than the wild type (Fig. 7E and Supplementary Data 3). DSB mobility of pGal:*EXO1* Las17-AID without IAA was the same as the wild type: R_C WT (0.93 μm) and R_C pGal:*EXO1* Las17-AID (0.87 μm) (Fig. 7F, G and Supplementary Data 1). When IAA was added before DSB induction to degrade Las17-AID (Fig. 7H, I), resection was significantly slower than the wild type (Fig. 7E and Supplementary Data 3). When IAA was added 2 h after DSB induction (Fig. 7H, J), the rate of resection was no longer significantly different than the wild type (Fig. 7E and Supplementary Data 3). There was no significant difference in the mobility of the DSB when IAA was

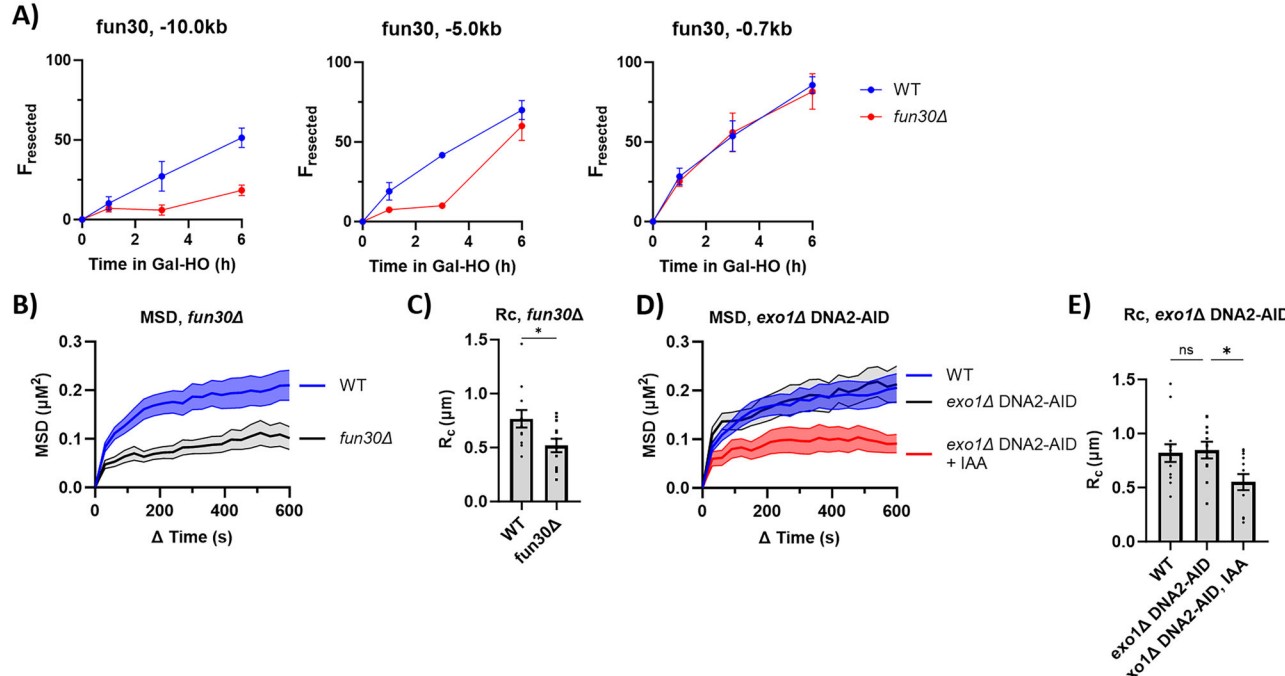

**Fig. 6 | Changes in the rate of resection lower DSB mobility. A** Resection analysis of FZ015 (WT) and FZ151 (*fun30Δ*). Samples were collected at 0, 1, 3, and 6 h after adding galactose. Resection analysis was done from a minimum of 3 biological replicates collected in triplicate. See "Methods" section for more details. Mean ± SEM is shown. Source data are provided as a Source Data file. **B** MSD analysis (Δt = 30 s) of DSB relative to the SPB in FZ015 (WT) (n = 12) and FZ151 (*fun30Δ*) (n = 12), 3 h after Gal-HO induction. Imaging and analysis done as described in Fig. 1C. Mean ± SEM is shown. Source data are provided as a Source Data file. **C** $R_C$ from MSD measurements in (**B**). Statistical analysis for the radius of confinement derived in PRISM using a two-sided unpaired t-test (ns p ≥ 0.05, *p < 0.05, **p < 0.01, and ***p < 0.001). WT vs. *fun30Δ* (p = 0.246). See Supplementary Data 1 and 2 for significance and $R_C$ values. Mean ± SEM is shown. Source data are provided as a Source Data file. **D** MSD analysis (Δt = 30 s) of DSB relative to the SPB in FZ015 (WT) (n = 13), FZ159 (*exo1Δ* DNA2-AID) (n = 11), and FZ159 (*exo1Δ* DNA2-AID) + IAA (n = 12) 3 h after Gal-HO induction. IAA (1 mM) was added 1 h before image collection. Imaging and analysis done as described in Fig. 1C. Mean ± SEM is shown. Source data are provided as a Source Data file. **E** $R_C$ from MSD analysis of strains in (**D**). Statistical analysis for the radius of confinement derived in PRISM using a one-way Anova (ns p ≥ 0.05, *p < 0.05, **p < 0.01, and ***p < 0.001). WT vs. *exo1Δ* DNA2-AID (p = 0.999), WT vs. *exo1Δ* DNA2-AID + IAA (p = 0.0266), and *exo1Δ* DNA2-AID vs. *exo1Δ* DNA2-AID + IAA (p = 0.0308). See Supplementary Data 1 and 2 for significance and $R_C$ values. Mean ± SEM is shown. Source data are provided as a Source Data file.

added to pGal:*EXO1* Las17-AID after DSB induction: $R_C$ pGal:*EXO1* Las17-AID + IAA (0.99 μm) (Fig. 7F, G and Supplementary Data 1).

These observations suggest that Exo1 does not have access to DSB ends when CK-666 blocks the initiation of resection, and again imply that the manner in which resection is initiated and how it is maintained are distinct. Moreover, in an *exo1Δ* Dna2-AID strain, when resection is blocked 2 h after DSB induction, there has been extensive resection, creating a large amount of ssDNA;[40] but the radius of confinement drops after Dna2 is inactivated. This result implies that the simple presence of a long ssDNA region is insufficient to activate DNA damage-associated chromosome mobility: continuous resection is required for the high mobility of DSBs.

**Activation of the TM checkpoint in response to blocking Arp2/3 activity**

When a DSB is detected, the DNA damage checkpoint (DDC) is activated to halt cell cycle progression and allow cells a chance to repair the DSB before proceeding through mitosis[44,45]. In budding yeast, activation of the DDC is mainly controlled through the effector kinase Mec1[ATR], with a minor contribution by the Tel1[ATM] kinase[46–49]. Mec1 is recruited to DSBs by its binding partner Ddc2[ATRIP], which binds to RPA loaded on ssDNA[50,51]. Arrest by the DDC is typically monitored by cell morphology, as cells shift towards a large-budded $G_2/M$ state, and by Western blot analysis of phosphorylation of the Mec1/Tel1 target, Rad53[52]. We hypothesized that preventing resection - by blocking Arp2/3 activity prior to HO induction - should interfere with cell cycle arrest through the DDC. Typically, a cell with a single-irreparable DSB

will arrest for 12–15 h before escaping arrest through a process known as adaptation[53]. Here we find that when Arp2/3 activity was blocked before DSB induction, either by treatment with CK-666 or degradation of Las17-AID with IAA (Fig. 8a), $G_2/M$ arrest was shortened to 4 h (Fig. 8B, C). Deletion of either Myo3 or Myo5 did not shorten checkpoint arrest (Fig. 8D, E).

While Mec1 is primarily responsible for checkpoint arrest by the DNA damage checkpoint, the Tel1-Mre11 (TM) checkpoint is an alternative checkpoint response in response to a DSB, notably in *mec1Δ* cells in which initial resection by the Mre11 complex is impaired[54]. We confirmed that the short checkpoint response we observed was attributable to the TM checkpoint, as *tel1Δ* and *mre11Δ* strains treated with CK-666 before DSB induction did not exhibit a checkpoint response (Fig. 8F, G). A *mre11Δ* Las17-AID double mutant treated with auxin also showed little or no $G_2/M$ arrest after DSB induction, and Western blot analysis showed that Rad53 was not phosphorylated (Fig. 8H, I).

Since degrading Las17 after DSB induction impaired further resection, we assayed whether Arp2/3 is required to maintain checkpoint arrest by adding CK-666 to our wild-type strain or IAA to Las17-AID two hours after adding galactose. By monitoring changes in cell morphology (Fig. 8A), we found that treatment with CK-666 or degrading Las17-AID resulted in cells escaping $G_2/M$ arrest early, despite Rad53 remaining hyperphosphorylated (Fig. 8J, K). These results show that the shortened DSB-induced checkpoint arrest was dependent on the TM checkpoint, which was activated when resection was impaired by inhibiting Arp2/3 or its associated factors.

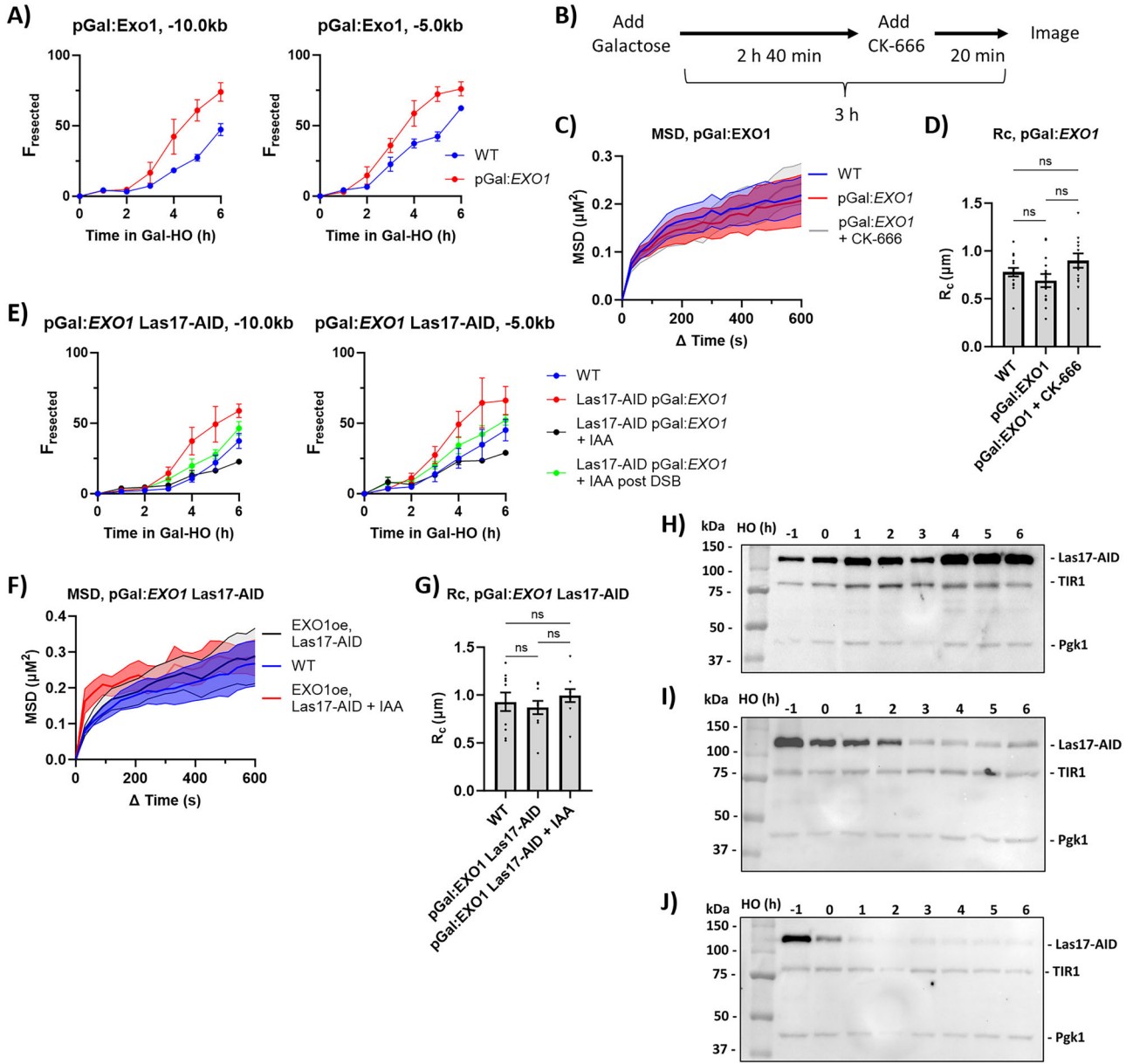

**Fig. 7 | Overexpression of *EXO1* rescues overcomes Arp2/3 inhibition.**
**A** Resection analysis of FZ201 (*pGal:exo1*). Samples were collected at 0, 1, 2, 3, 4, 5, and 6 h after adding galactose. Resection analysis was done from a minimum of 3 biological replicates collected in triplicate. See "Methods" section for more details. Mean ± SEM is shown. Source data are provided as a Source Data file. **B** Diagram of the experimental setup to measure the MSD of a DSB labeled with Ddc2-GFP. CK-666 (100 μM) was added 2 h 40 min after adding galactose. **C** MSD analysis (Δt = 30 s) of DSB relative to the SPB in FZ015 (n = 15), FZ201 (*pGal:EXO1*) (n = 14), and FZ201 with CK-666 (100 μM) (*pGal:EXO1*) (n = 13) 3 h after Gal-HO induction. CK-666 (100 μM) was added 20 min before image collection as described in Fig. 1C. Imaging and analysis were done as described in (**B**). Mean ± SEM is shown. Source data are provided as a Source Data file. **D** R$_C$ from MSD analysis of strains in (**C**). Statistical analysis for the radius of confinement derived in PRISM using a one-way Anova (ns p ≥ 0.05, *p < 0.05, **p < 0.01, and ***p < 0.001). WT vs. pGal::*EXO1* (p = 0.6748), WT vs. pGal::*EXO1* + CK-666 (p = 0.4505), and pGal::*EXO1* vs. pGal::*EXO1* + CK-666 (p = 0.0737). See Supplementary Data 1 and 2 for significance and R$_C$ values. Mean ± SEM is shown. Source data are provided as a Source Data file. **E** Resection analysis of FZ015 (WT) and FZ218 (Las17-AID pGal::*EXO1*). IAA was added either 1 h before galactose or 1 h after galactose. Samples were collected at 0, 1, 3, and 6 h after adding galactose. Resection analysis was done from a minimum of 3 biological replicates collected in triplicate. See "Methods" section for more

details. Mean ± SEM is shown. Source data are provided as a Source Data file. **F** MSD analysis (Δt = 30 s) of DSB relative to the SPB in FZ015 (n = 10), FZ218 (*pGal:EXO1* Las17-AID) (n = 10), and FZ218 with IAA (1 mM) 2 h after galactose (*pGal:EXO1* Las17-AID + IAA) (n = 10). 3 h after Gal-HO induction. CK-666 (100 μM) was added 20 min before image collection as described in Fig. 1C. Imaging and analysis done as described in Fig. 1C. Mean ± SEM is shown. Source data are provided as a Source Data file. **G** R$_C$ from MSD analysis of strains in (**F**). Statistical analysis for the radius of confinement derived in PRISM using a one-way Anova (ns p ≥ 0.05, *p < 0.05, **p < 0.01, and ***p < 0.001). WT vs. pGal:EXO1 Las17-AID (p = 09371), pGal:EXO1 Las17-AID vs. pGal:EXO1 Las17-AID + IAA (p = 0.6209), and WT vs. pGal:EXO1 Las17-AID + IAA (p = 0.9172). See Supplementary Data 1 and 2 for significance and R$_C$ values. Mean ± SEM is shown. Source data are provided as a Source Data file. **H** Western blot of Las17-AID pGal::*EXO1* probed with α-Myc to show Las17-AID degradation and phosphorylated protein, and α-Pgk1 as a loading control. Source data are provided as a Source Data file. **I** Western blot of Las17-AID pGal::*EXO1* with IAA added 1 h before DSB induction, probed with α-Myc to show Las17-AID degradation and α-Pgk1 as a loading control. Source data are provided as a Source Data file. **J** Western blot of Las17-AID pGal::*EXO1* with IAA added 2 h after DSB induction, probed with α-Myc to show Las17-AID degradation and α-Pgk1 as a loading control. Source data are provided as a Source Data file.

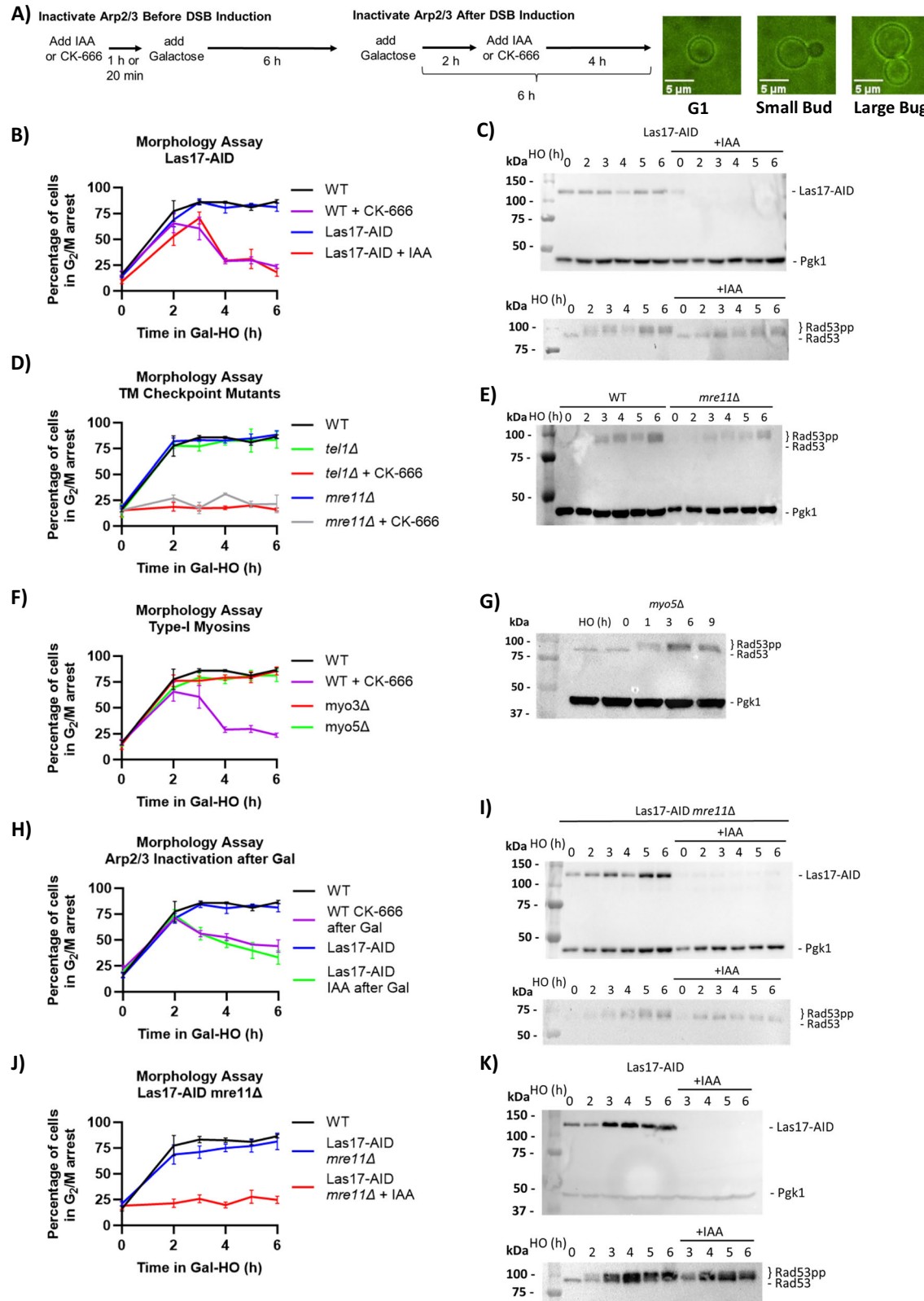

## DSB mobility and repair by gene conversion

If DSB-induced chromatin mobility facilitates homology search, then lowering the mobility of DSBs might impact the repair efficiency carried out by gene conversion (GC). To study how changes in chromatin mobility affect the efficiency of DSB repair, we modified the well-characterized strain YJK17, which has an ectopic, mutated copy of *MAT* (*MATa-inc*) that cannot be cut by the HO endonuclease and which

serves as a donor to repair the cleavage at *MATα* by interchromosomal gene conversion (Fig. 9A)[40,55]. With this strain, we could generate and monitor the repair of a single DSB event.

Deletion of *MYO3*, *MYO5*, or *MRE11* did not impact repair; however, Las17-AID showed a significant decrease in viability (Fig. 9B and Supplementary Data 4). An *mre11Δ* Las17-AID double mutant showed an even more significant drop in viability (10% on YP-Gal+ IAA versus

**Fig. 8 | The Tel1-Mre11 (TM) checkpoint prolongs checkpoint arrest when Arp2/3 is blocked before DSB induction. A** Experimental setup to measure the percentage of G$_2$/M arrested cells after DSB induction. CK-666 or IAA were added 20 min or 1 h before galactose to block Arp2/3 activity before DSB induction. To block Arp2/3 activity after DSB induction, CK-666 or IAA was added 2 h after galactose. Example images of G$_1$, small budded, and large-budded (G$_2$/M) cells. **B** Percentage of G$_2$/M arrested cells in FZ015 (WT) and FZ075 (Las17-AID) ± IAA after DSB induction. IAA was added before galactose as described in (**A**). Data was collected from 3 separate days (n = 3) with >100 cells counted at each timepoint per day. Mean ± SEM is shown. **C** Western blot of Las17-AID ± IAA probed with α-Myc to show Las17-AID degradation, α-Rad53 shows both an unphosphorylated and phosphorylated protein, and α-Pgk1 as a loading control. Source data are provided as a Source Data file. **D** Percentage of G$_2$/M arrested cells in FZ015 (WT) ± CK-666, FZ033 (myo3Δ), and FZ034 (myo5Δ) after DSB induction. CK-666 was added before galactose as described in (**A**). Data was collected from 3 separate days (n = 3) with >100 cells counted at each timepoint per day. Mean ± SEM is shown. Source data are provided as a Source Data file. **E** Western blot of myo5Δ probed with α-Rad53 shows both an unphosphorylated and phosphorylated protein, and α-Pgk1 as a loading control. Source data are provided as a Source Data file. **F** Percentage of G$_2$/M arrested cells in a FZ015 (WT), FZ019 (mre11Δ) ± CK-666, and YSL56 (tel1Δ) ± CK-666 strains, where CK-666 was added before galactose as described in (**A**). Data are

shown from 3 independent experiments with error bars representing the standard error of the mean (SEM). Data was collected from 3 separate days (n = 3) with >100 cells counted at each timepoint per day. Mean ± SEM is shown. Source data are provided as a Source Data file. **G** Western blot of WT and mre11Δ strains after DSB induction, probed with α-Rad53 shows both an unphosphorylated and phosphorylated protein, and α-Pgk1 as a loading control. Source data are provided as a Source Data file. **H** Percentage of G$_2$/M arrested cells in FZ015 (WT) and FZ192 (Las17-AID mre11Δ) ± IAA DSB induction. IAA was added before galactose as described in (**A**). Data was collected from 3 separate days (n = 3) with >100 cells counted at each timepoint per day. Mean ± SEM is shown. Source data are provided as a Source Data file. **I** Western blot of Las17-AID mre11Δ ± IAA probed with α-Myc to show Las17-AID degradation, α-Rad53 shows both an unphosphorylated and phosphorylated protein, and α-Pgk1 as a loading control. Source data are provided as a Source Data file. **J** Percentage of G$_2$/M arrested cells in FZ015 (WT) ± CK-666 and FZ075 (Las17-AID) ± IAA after DSB induction. CK-666 or IAA were added 2 h after galactose as described in (**A**). Data was collected from 3 separate days (n = 3) with >100 cells counted at each timepoint per day. Mean ± SEM is shown. Source data are provided as a Source Data file. **K** Western blot of Las17-AID ± IAA probed with α-Myc to show Las17-AID degradation, α-Rad53 shows both an unphosphorylated and phosphorylated protein, and α-Pgk1 as a loading control. Source data are provided as a Source Data file.

65% on YP-Gal) (Fig. 9B and Supplementary Data 4). To test whether the lower viability of the Las17-AID and mre11Δ Las17-AID mutants when treated with IAA was attributable to the shortened G2/M arrest, we added nocodazole (10 µg/ml) an hour before galactose to extend the length of G2/M arrest for up to 6 h. Because the addition of nocodazole lowered the viability of cells (Fig. 9C), we normalized the outcome of nocodazole-treated cells to the level of wildtype + nocodazole (Fig. 9D). When Las17-AID ± IAA and mre11Δ Las17-AID ± IAA were kept in nocodazole and galactose for 6 h before plating on YP-Gal plates, there was an increase in the number of survivors (Fig. 9D and Supplementary Data 4). Las17-AID survivors from the YP-Gal and YP-Gal+IAA plates grew normally when re-plated on YPD and YP-Gal (Supplementary Fig. 9), as expected.

Since the lower rate of repair by gene conversion was due to a shortened checkpoint arrest, we also monitored *MAT* switching, a highly efficient intrachromosomal gene conversion event that does not activate the DNA damage checkpoint[56]. Using a strain with an *HMLa-inc* donor and a Gal-inducible DSB in the *MAT*a locus (Fig. 9E), we found that mating-type switching was not affected by CK-666 (Fig. 9F and Supplementary Data 4).

To test further the effects of mobility-deficient mutants on interchromosomal gene conversion, we examined myo5Δ and Las17-AID + IAA in a series of strains we had previously characterized.[57] In these strains, a copy of *LEU2* with an HO cut site is located on chromosome V, and a *LEU2* donor sequence is inserted at different locations that vary in their contact probability with this HO-induced DSB site (Supplementary Fig. 9A, B). Repair efficiencies ranged from 46% to 9%[57]. None of the myo5Δ mutants showed diminished repair by gene conversion, although one strain had a small increase in repair efficiency (Supplementary Fig. 9C–F and Supplementary Data 3). However, Las17-AID + IAA showed a highly significant decrease in the percentage of survivors in all 4 strains (Supplementary Fig. 9G–J and Supplementary Data 4). As noted above, while myo5Δ exhibits a decreased R$_C$, it did not affect resection and cell cycle arrest, whereas inactivating las17 reduced R$_C$ and also blocked resection.

Since degradation of Las17 showed reduced resection, we wanted to know how this would affect repair by single-strand annealing (SSA) using the previously characterized strain YMV2, which has an HO cut site inserted in *LEU2* on chromosome 3 (leu2-cs) and a 1.3 kb fragment of the 3′ end of *LEU2* (U2) inserted 30 kb upstream of leu2-cs (Fig. 9G)[58]. The DSB can be repaired both by Rad51-dependent break-induced replication and by Rad51-independent SSA[40]. Degradation of Las17 reduced the survival rate from 86% to 63% (Fig. 9H). Deletion of *RAD51*

did not affect the repair efficiency (Fig. 9H), yet when Las17 was degraded in the *rad51Δ* background, repair efficiency dropped to 36% (Fig. 9H). This suggests that Las17 and Arp2/3 have a much more important role in repair by SSA, which requires extensive 5′ to 3′ resection, than on break-induced replication (BIR), where only a small extent of resection is required.

## Discussion

When a cell commits to repair a DSB through HR, there is an increase in chromatin mobility near the DSB[3,4,25]. Here we show that Arp2/3, previously shown to be required for DSB mobility in *Drosophila* and mammalian cells[7,8], has a conserved role in DSB mobility in budding yeast and that this process is separate from Arp2/3's well-characterized function in endocytosis[59]. It appears that Arp2/3 is required at 2 different steps during homology-based repair: for both the initiation and the maintenance of long-range resection. While it has been thought that increased chromatin mobility near a DSB facilitates homology search, we find that a change in the radius of confinement per se does not affect repair, whereas a reduced DNA damage checkpoint and reduced G2/M arrest have a profound effect.

Long-range resection was not seen when Arp2/3 activity was blocked before DSB induction, either through treatment with the Arp2/3 inhibitor CK-666, deletion of *VRP1*, degradation of Las17$^{WASP}$, or when both type-I myosins, Myo3 and Myo5, were not present. Unlike a *sae2Δ* mutant, which has been reported to have delayed DSB-induced chromatin mobility[4] and delayed resection[60], we saw little or no resection up to 6 h after inducing an HO break when Arp2/3 was blocked prior to DSB induction.

In previous studies, deletion of *YKU70/80* allowed Exo1-dependent resection of DSB ends in G1-arrested cells[19] and over-expression of *YKU70/80* interfered with Mre11 association with DSB ends and thus acted as a barrier for resection[61]. Also, Ku70/80 from HeLa cells interacted with polymerized F-actin, and that treatment with the drug cytochalasin D, which prevents actin polymerization[62], interfered with Ku70/80 chromatin retention[63]. Here we show that deleting *YKU70* did not overcome the CK-666 block to forming Ddc2-GFP foci. These data suggest that removal of yKu70/80 is not a barrier to resection when Arp2/3 is inhibited before inducing a DSB.

The connection between increased chromatin mobility and end-resection is complex. As in mammalian cells, slowing down resection reduces the mobility of DSB ends, though whether the same factors are involved remains unclear. In mammals, inhibition of Mre11 by mirin reduced mobility[8], while in budding yeast, inhibition of the Mre11

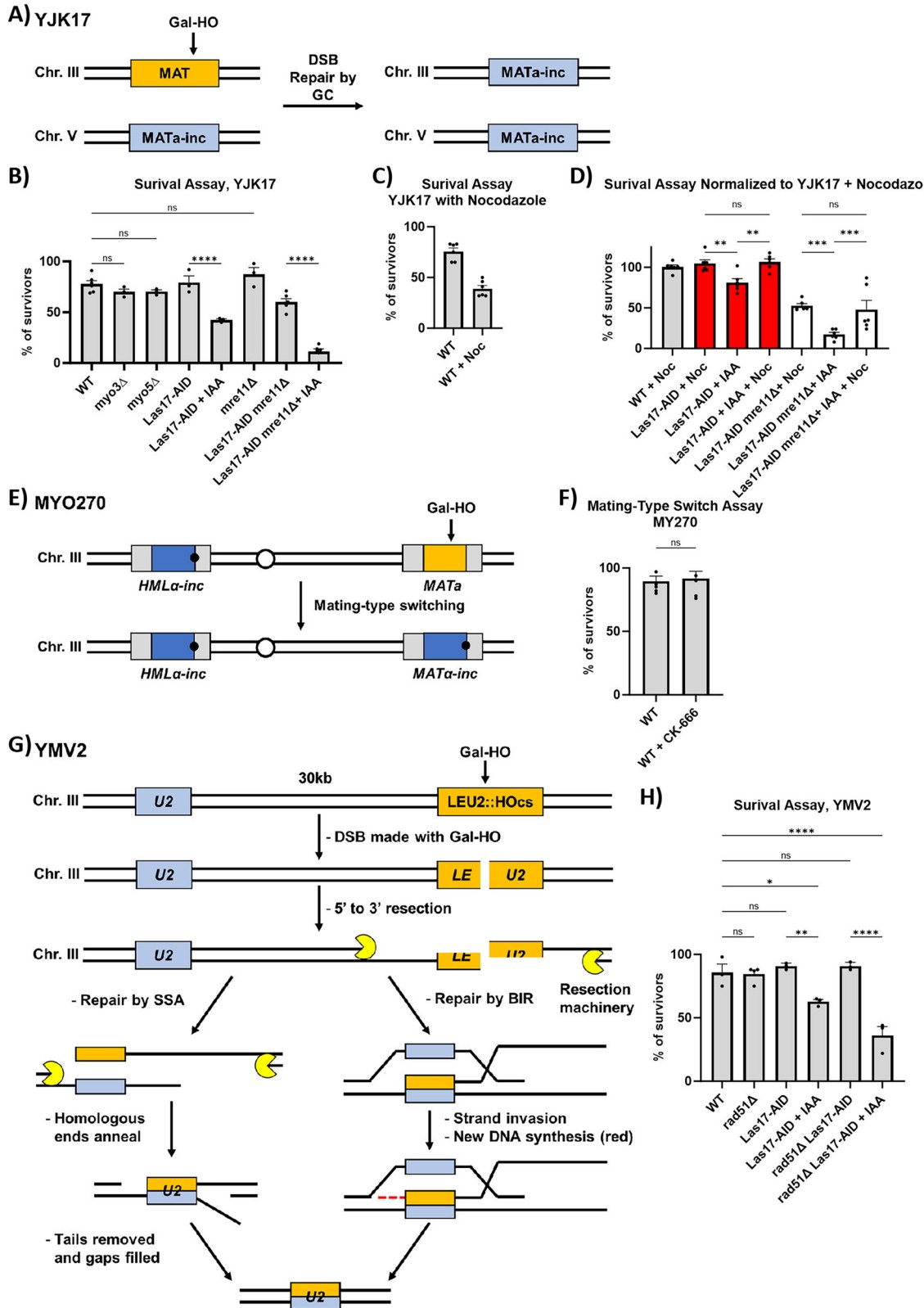

complex by deletion of *SAE2* only delays resection and the increase in $R_C$[4,60]. Here we found that $R_C$ was reduced by impairing long-range resection, either by deleting the chromatin remodeler *FUN30* or by inactivation of both Exo1 and DNA2. Despite extensive ssDNA formed by resection in the *exo1* deletion[40], stopping resection by degrading DNA2 lowered the Rc of the DSB, showing that a defect in long-range end-resection is sufficient to impair damage-induced chromatin mobility.

We also found that *EXO1* overexpression suppressed the reduction in Rc caused by adding CK-666 by maintaining wild-type levels of resection, but overexpression of *EXO1* was not able to bypass the block in initiating resection when Arp2/3 was inhibited prior to DSB induction. We were unable to test DSB mobility in a triple deletion mutant of the primary resection proteins because an *mre11-nd* (nuclease dead) *exo1 sgs1Δ* deletion is inviable[42]; but it would appear that inactivating

**Fig. 9 | The effect of NPFs on DSB repair by gene conversion. A** Repair strain YJK17. The donor *MAT*a-inc on chromosome 5 is used as a donor to repair the Gal-HO-cut in the *MAT* locus on chromosome 3 through gene conversion (GC). **B** Survival assay in YJK17 mutants: YJK17 (WT) (n = 6), FZ063 (*myo3Δ*) (n = 3), FZ040 (*myo5Δ*) (n = 3), FZ177 (Las17-AID) (n = 3), Las17-AID + IAA (1 mM) (n = 3), FZ172 (*mre11Δ*) (n = 3), FZ193 (Las17-AID *mre11Δ*) (n = 6), and Las17-AID *mre11Δ* + IAA (n = 6). -200–100 cells from a YP-Lac culture were plated on YPD, YP-Gal, and YP-Gal+IAA plates. The survival rate was calculated by the number of cells grown on a YEP-Gal or YEP-Gal+IAA plate divided by the number of cells grown on a YPD plate. Statistical analysis for the percentage of survivors was derived in PRISM using a one-way Anova (ns p ≥ 0.05, *p < 0.05, **p < 0.01, and ***p < 0.001). WT vs. *myo3Δ* (p = 0.6053), WT vs. *myo5Δ*, (p = 0.6505), WT vs. *mre11Δ*, (p = 0.3879), Las17-AID vs. Las17-AID + IAA (p < 0.0001), and Las17-AID *mre11Δ* vs. Las17-AID *mre11Δ* + IAA (p < 0.0001). See Supplementary Data 3 for significance. See "Methods" section for more details. Mean ± SEM is shown. Source data are provided as a Source data file. **C** Survival assay of YJK17 (n = 6) and YJK17 + nocodazole (Noc) (10 μg/ml) (n = 6). Noc was added 1 h before galactose. See "Methods" section for more details. Mean ± SEM is shown. Source data are provided as a Source Data file. **D** Survival assay of YJK17 mutants normalized to wildtype + nocodazole (10 μg/ml): YJK17 + Noc (WT + Noc) (n = 6), Las17-AID + Noc (n = 6), Las17-AID + IAA (n = 6), Las17-AID + IAA + Noc (n = 6), Las17-AID *mre11Δ* (n = 6), Las17-AID *mre11Δ* + IAA (n = 6), and Las17-AID *mre11Δ* + IAA + Noc (n = 6). Survival assay conducted as described in (**B**). Nocodazole (10 μg/ml) was added 1 h before galactose. IAA (1 mM) was added 1 h before galactose. See "Methods" section for more details. Statistical analysis for the percentage of survivors was derived in PRISM using a one-way Anova (ns p ≥ 0.05, *p < 0.05, **p < 0.01, and ***p < 0.001). Las17-AID + Nocodazole vs. Las17-AID + IAA (p = 0.0067), Las17-AID + IAA vs. Las17-AID + IAA Nocodazole (p = 0.0038), Las17-AID + Nocodazole vs. Las17-AID + IAA + Nocodazole (p = 0.8219), Las17-AID *mre11Δ*

+ Nocodazole vs. Las17-AID *mre11Δ* + IAA (p = 0.0001), Las17-AID *mre11Δ* + IAA vs. Las17-AID *mre11Δ* + IAA Nocodazole (p = 0.0007), and Las17-AID *mre11Δ* + Nocodazole vs. Las17-AID *mre11Δ* + IAA + Nocodazole (p = 0.5676). See Supplementary Data 3 for significance. Mean ± SEM is shown. Source data are provided as a Source Data file. **E** Mating-type switch strain MY270. The donor *HMLα-inc* is used as a donor to repair the Gal-HO cut in the *MAT* locus on chromosome 3 through gene conversion (GC). After repair, *MATα-inc* cannot be cut by Gal-HO[78]. **F** Survival assay of MY270 (n = 6) and MYO270 + CK-666 (100 μM) (n = 6). Survival assay conducted as described in (**B**). CK-666 (100 μM) or 200 proof ethanol was added 20 min before plating on YPD and YP-Gal plates. Statistical analysis for the percentage of survivors was derived in PRISM using a two-sided unpaired t-test (ns p ≥ 0.05, *p < 0.05, **p < 0.01, and ***p < 0.001). WT vs. WT + CK-666 (p = 0.7417). See Supplementary Data 3 for significance. Mean ± SEM is shown. Source data are provided as a Source Data file. **G** Repair strain YMV2 and repair by single-strand annealing (SSA) or break-induced replication (BIR). An HO cut site is inserted in LEU2 on chromosome 3 (*LEU2::HOcs*), and a 1.3 kb donor of the C-terminal end of LEU2 (*U2*) is inserted 30 kb upstream of *LEU2::HOcs*. The 30 kb sequence between the homologous sequences is deleted after repair by either SSA or BIR. **H** Survival assay of YMV2 strains grown on YP-Gal-IAA plates: YMV2 (WT) (n = 3), YMA65 (*rad51Δ*) (n = 3), YMA63 (Las17-AID) (n = 3), YMA63 + IAA (Las17-AID + IAA) (n = 3), YMA66 (*rad51Δ* Las17-AID) (n = 3), and YMA66 + IAA (*rad51Δ* Las17-AID + IAA) (n = 3). Survival assay conducted as described in (**B**). WT vs. *rad51Δ* (p > 0.9999), WT vs. Las17-AID (p = 0.9807), WT vs. *rad51Δ* Las17-AID (p = −0.9807), WT vs. Las17-AID + IAA (p = 0.0172), WT vs. *rad51Δ* Las17-AID + IAA (p < 0.0001), Las17-AID vs. Las17-AID + IAA (p = 0.0038), and *rad51Δ* Las17-AID vs. *rad51Δ* Las17-AID + IAA (p < 0.0001). See Supplementary Data 3 for significance. Mean ± SEM is shown. Source data are provided as a Source Data file.

Arp2/3 in an *mre11Δ* background eliminates both long- and short-range resection and thus recombination. These findings confirm and extend the analysis of the roles of Arp2/3 reported in mammalian cells, where treatment with CK-666 impaired both mobility and end-resection[8].

Detection of a DSB triggers cell cycle arrest through the DNA damage checkpoint (DDC) for 12–15 h[53]. We found that when Arp2/3 was blocked, cells experienced a short, 4-h checkpoint which was dependent on the Tel1-Mre11 (TM) checkpoint, which had previously been found only in *mec1Δ* cells when components of the Mre11-Rad50-Xrs2-Sae2 complex were mutated, although some TM-activating mutations have no apparent effect on end-resection[64–66]. Here, inhibiting Arp2/3 and NPFs mimics these Rad50-S and Sae2 mutants. Deleting Mre11 or Tel1 and blocking Arp2/3 or Las17 activity prevented cells from entering DSB-induced checkpoint arrest. Arp2/3 was also required to *maintain* checkpoint arrest, as cells escape arrest after 4 h, despite continued Rad53 phosphorylation. Impaired Mec1-mediated checkpoint arrest in budding yeast is consistent with the lower ATR signaling reported in WASP-deficient human cells[12].

Of the known Arp2/3 actin nucleation-promoting factors in budding yeast, we focused on Las17[WASP] and the type-I myosins Myo3 and Myo5. In humans, WASP has been shown to bind to RPA and is implicated in loading RPA onto ssDNA in response to a DSB[8]. We show that the WH2 domain, found in the highly conserved C-terminal region, of Las17 is required for DSB mobility. Human WASP binds to RPA through this highly conserved region and mediates RPA binding to ssDNA[12]. From a total budding yeast extract, pulldown of a peptide that binds to monomeric G-actin and polymerized F-actin brought down 2 subunits of RPA, but a peptide that binds solely to F-actin did not bring down RPA[67]. Since Las17 binds to G-actin through its WH2 domain, RPA loading onto ssDNA might be mediated by G-actin binding to Las17. Impaired RPA binding to ssDNA would prevent Ddc2[ATRIP]-mediated Mec1 activation of the DDC.

The budding yeast type-I myosins Myo3 and Myo5 are functionally redundant in endocytosis, and deletion of either Myo3 or Myo5 did not affect resection, checkpoint arrest, basal levels of chromatin mobility, or repair by gene conversion. However, a deletion of either myosin protein lowered DSB mobility, and the insertion of a second copy of the remaining type-I myosin, e.g., a second copy of Myo5 in a *myo3Δ* mutant, restored DSB mobility to wild-type levels. Through domain deletion mutants of *MYO5*, we found that the SH3 domain of Myo5, which mediates interactions with Vrp1 and recruitment to Arp2/3[68,69], is required for wild-type levels of DSB mobility. Since SH3 domains mediate a wide range of protein-protein interactions, the loss of either type-I myosin might result in an imbalance of interactions required for DSB mobility, and the deletion of *MYO3* and *MYO5* might restore that balance.

Surprisingly, the loss of both type-I myosins after DSB induction appeared to restore DSB mobility back to wild-type levels. However, a recent study has shown that depleting both type-I myosins from the nucleus, without a DSB, causes a dramatic change in chromosome confirmation[14]. Here, we showed that this change in overall chromatin organization was also seen in an increase in SPB mobility. Hence, it is difficult to assess the loss of both type-I myosins on chromosome mobility. We did, however, see that the double inactivation of these myosins blocked resection similarly to inactivating Las17.

Changes in DSB mobility do not necessarily affect overall DSB repair efficiency through HR. While the degradation of Las17 negatively affected the repair efficiency of DSBs through GC and SSA, deletion of either *MYO3* or *MYO5* did not affect the repair efficiency of DSBs, even when the donor location was moved to unfavorable locations. This difference in repair efficiencies among these mobility-deficient mutants instead correlates with the length of G2/M arrest imposed by the DDC. Since *myo3Δ* and *myo5Δ* mutants did not affect the length of checkpoint arrest, these mutants had the same amount of time to repair the break as the wild type. The shortened checkpoint arrest when Las17 was degraded might have triggered an early adaptation response. Lengthening the duration of checkpoint arrest by nocodazole treatment in Las17-AID + IAA and Las17-AID *mre11Δ* + IAA strains improved DSB repair efficiency through gene conversion. Further evidence that changes in DSB mobility do not correlate with DSB repair efficiency can be seen in the deletion of the ubiquitin ligase, Uls1, which lowers chromatin mobility near DSB sites, but does not affect repair of a single DSB through break-induced recombination[25].

In summary, by using single particle tracking and MSD analysis of a single DSB in budding yeast, we were able to show that Arp2/3 has an evolutionarily conserved role in the DNA damage response and plays an important part in both the initiation and facilitation of resection of DSB ends. Conversely, defects in 5′ to 3′ resection also impair mobility. Increased chromosome mobility depends on continuing resection even when there is a large amount of ssDNA generated around the break. The rate of resection was not the only determinant of DSB mobility, as a deletion of either Myo3 or Myo5 was sufficient to lower DSB mobility but did not affect the rate of resection. Further work will be required to determine how Arp2/3 and NPFs interact with the resection machinery and how increased chromosome movement is affected.

## Methods

### Strain and plasmid construction

All strains are derived from JKM179[17], a well-characterized strain with the *HML* and *HMR* donor domains deleted. Ddc2 and Rad51 were GFP-tagged with a 13 amino acid linker GGSGGSRIPGLIN-eGFP as previously described[15]. Ddc2-GGSGGSRIPGLIN-eGFP and Rad51-GGSGGSRIPGLIN-eGFP are referred to as Ddc2-GFP and Rad51-GFP, respectively, in this study. All AID-tagged mutant strains were derived from a modified version of JKM179 expressing osTIR1 at URA3 created by cutting pNHK53[28] with *Stu*I and integrating it into the genome. For degron-tagged derivatives, PCR products were generated with mixed oligos with homology to the C-terminal end and plasmid pJH2892 or pJH2899[27] to create a 9xMyc-AID (AID) insert with the KAN or NAT marker. Myo5-domain deletion plasmids[30] were digested with *Bam*HI/*Sac*I to excise a linear fragment for genome integration. Deletion of ORFs, *myo5* mutants, and AID tags were introduced with the one-step PCR homology cassette amplification and the standard yeast transformation method[70]. Las17 domain deletion mutants were created by CRISPR/Cas9 as previously described[71] with material listed in supplemental tables. Transformants were verified by PCR and Western blot. Strains are listed in Supplementary Data 5. Primers are listed in Supplementary Data 6. Plasmids are listed in Supplementary Data 7. Key Resources are listed in Supplementary Data 8.

### Growth conditions

Strains containing degron fusions and galactose-inducible HO were cultured using standard procedures. Briefly, a single colony grown on a YPD plate was inoculated in 5 ml YP-lactate for -15 h overnight at 30 °C with agitation. The following day, 500–100 ml of YP-lactate was inoculated with the starter culture so that the cell density reached $OD_{600} = 0.5$ the following day after overnight growth at 30 °C with agitation. Once cultures reached the appropriate density, HO expression was induced by the addition of 20% galactose to 2% vol/vol final concentration. For auxin treatment, indole-3-acetic acid (IAA) was resuspended in ethanol to a working concentration of 500 mM. For AID degradation, cultures were split at the indicated timepoints to add auxin, IAA was added to one culture to a final concentration of 1 mM either 1 h before or 2 h after adding galactose. The equivalent volume of 100% ethanol was added to the second culture. Cells were harvested at the indicated timepoint by centrifugation at 3000× for 3 min and prepared for microscopy, resection assays, and western blot analysis. Cells prepared for microscopy were washed 3 times with Leu-media.

### Plating assays

The efficiency of DSB repair by homologous recombination or single-strand annealing was determined as previously described for YJK17[55]. Briefly, cells were selected from a single colony on YPD plates and grown overnight in 5 ml of YEP-lactate. Cells were diluted to $OD_{600} = 0.2$ and allowed to grow until $OD_{600} = 0.5–1.0$. Approximately 100 cells from each culture were then placed on YEP-Gal (2% gal per volume) or YEP-Gal+IAA (1 mM) and YPD in triplicate and incubated at 30 °C until visible colonies formed. For plating experiments with CK-666, overnight YEP-lactate cultures were treated with CK-666 (100 μM) 20 min prior to being plated on YPD and YEP-Gal plates in triplicate, as previously described.

For plating assays with IAA and/or nocodazole, YP-Lac cultures were split and given 100% ethanol (control), IAA (1 mM), nocodazole (10 μg/ml)[72], or both IAA and nocodazole 1 h prior to galactose. Ethanol-treated cultures were directly plated on YPD and YP-Gal plates. Galactose (2%) was added to cultures treated with IAA and/or noco-dazole and allowed to grow at 30 °C for 6 h before plating on YEP-Gal or YEP-Gal+IAA plates. Viability was calculated by dividing the number of colonies on YEP-Gal or YEP-Gal+IAA plates by the number of colonies on YPD plates. Viability in nocodazole-treated cells was calculated by dividing the percentage of survivors in nocodazole-treated cells by the percentage of survivors of wild-type cells treated with nocodazole.

### Live cell microscopy and mean-squared displacement analysis

Live cell microscopy was performed using a spinning disc confocal microscope using Nikon Elements AR software on a Nikon Ni-E upright microscope equipped with a 100× (NA, 1.45) oil immersion objective, a Yokogawa CSU-W1 spinning-disk head, and an Andor iXon 897U EMCCD camera. Fluorophores were excited at 488 nm (GFP) and 561 nm (mCherry). Time-lapse series were acquired with 15 optical slices of 0.3 μm thickness every 30 s for 20 min. Time-lapse image stacks were analyzed using a custom MATLAB program designed by the Bloom lab as previously described[73]. Coordinates of DSB and the SPB were tracked with Speckle Tracker, a custom MATLAB program[74,75]. A custom PERL script was used to convert the pixels to nanometers and subtract the distance of the SPB from the Ddc2-GFP coordinates to eliminate cell and nuclear motion, subtract the mean position of the corrected GFP coordinates, calculate the MSD of each time lapse, and export MSD coordinates and radius of confinement ($R_c$) to an Excel spreadsheet[73]. In MATLAB, spot positions were fitted to $[\mu_x, \sigma_x] = \text{normfit}(x - x_{mean})$ and $[\mu_y, \sigma_y] = \text{normfit}(y - y_{mean})$. The variance of the distribution of spot position was then calculated as $\sigma^2 = \text{mean}(\sigma_x^2, \sigma_y^2)$. The average squared deviation from the mean position is $(\Delta r_0^2) = (\Delta x_0^2) + (\Delta y_0^2)$. Using $\sigma^2$ and $(\Delta r_0^2)$, we calculated $R_c$ as $R_C = \sqrt{\frac{5}{4}(2\sigma^2 + \Delta r_0^2)}$. Due to variations in day-to-day collection, wild-type data was only compared to data collected on the same day.

### Statistics and reproducibility

Statistical analysis for Rc comparison was done in PRISM using a two-tailed t-test or one-way ANOVA analysis. For one-way ANOVA analysis, the Dunnett test was used to correct for multiple comparisons. Wild-type comparisons were conducted with data collected from the same day. MSD and $R_c$ analysis were done from at least 10 cells ($n \geq 10$) collected across at least 2 days to account for variations in microscopy conditions. Resection analysis was done from 3 biological replicates collected in triplicate. Changes in cell morphology were measured across 3 days, with >100 cells counted at each timepoint per day. Western blots to show degradation of AID-tagged proteins were run at least twice: first to validate the creation of the strain and second to measure degradation of AID-tagged proteins under experimental conditions.

### Resection and cutting assays

Resection was measured by quantitative PCR (qPCR) analysis using a restriction enzyme digest as previously described[36]. Briefly, cells are grown in YEP-Lac as described above. 50 ml of culture was harvested and DNA extracted using a DNA extraction kit (Masterpure Yeast DNA Purification Kit, Cat# MPY80200). DNA was diluted to 10 ng/μl. Sty1-HF (digest) or an equivalent amount of water (mock) in Cutsmart buffer at 37 °C for 4 h. qPCR samples were run in triplicate on a Bio-Rad CFX384 Real-Time System C1000 Touch Thermal Cycler qPCR machine using

Bio-Rad CFX Maestro 1.1 Version 4.1.2433.1219. *ADH1* was used as a control gene. See Supplementary Data 5 for primers. Resection was calculated by measuring the fraction of cells that had passed the Sty1 restriction site (RS).

$$x = \frac{2}{\left(\frac{E_{RS}^{\Delta C_q(digest-mock)}}{E_{ADH1}^{\Delta C_q(digest-mock)}} + 1\right)*f} \tag{1}$$

*f* is the fraction of cells where HO has been cleaved. $E_{RS}$ and $E_{ADH1}$ are the primer efficiencies for the primer pairs 0.7 kb, 5 kb, and 10 kb away from the HO-cut site and the *ADH1* primers. $\Delta C_q$ (digest−mock) is the difference between the quantification cycles between the mock and digested samples.

For cutting assays, DNA was collected as described above. DNA was diluted to 10 ng/μl and run using primers flanking the HO-cut site in the *MAT* locus on chromosome III. *ADH1* was used as a control. Gal-HO cutting was measured by the fold increase with the following equation, using the 0 h timepoint as a control.

$$fold\ increase_t = 2^{-(\Delta C_t - \Delta C_{controlaverage})}. \tag{2}$$

## TCA protein extraction
Protein extracts were prepared for western blot analysis by the standard TCA protocol described in ref. 76. Briefly, 15–10 ml of harvested cells were incubated on ice in 1.5 ml microcentrifuge tubes with 20% TCA for 20 min. Cells were washed with acetone, and the pellet was air-dried. 200 μl of MURBs buffer (50 mM sodium phosphate, 25 mM MES, 3 M urea, 0.5% 2-mercaptoethanol, 1 mM sodium azide, and 1% SDS) was added to each sample, allowed to incubate with acid-washed glass beads. Cells were lysed by mechanical shearing with glass beads for 2 min. Supernatant was collected by poking a hole in the bottom of the 1.5 ml microcentrifuge tube and spun in a 15 ml conical tube. Samples were boiled at 95 °C for 10 min.

## Western blotting
Denatured protein samples prepared by TCA extraction were centrifuged at max speed for 1 min, and 8–20 μl of samples were loaded into a 10% or 8% SDS page gel. Proteins were separated by applying 90 V constant until the 37 kDa marker reached the bottom of the gel. Gels were then transferred to an Immun-Blot PVDF using a wet transfer apparatus set to 100 V constant voltage for 1 h. The resulting membranes were blocked in 5% nonfat dry milk or OneBlock buffer (Genesee Scientific, 20-313) for 1 h at room temperature or overnight at 4 °C with gentle agitation. After washing 3 times with 1× TBS-T, blots were incubated with either mouse anti-Myc [9E11] (Abcam, ab56) to detect Tir1 and AID fusions, rabbit anti-Rad53 (Abcam, ab104232), or mouse anti-Pgk1 antibody (Abcam Cat# ab113687, RRID:AB_10861977) for 1 h at room temperature. Blots were then washed 3 times with 1× TBS-T and incubated with anti-mouse or anti-rabbit HRP secondary antibody for 1 h at room temperature. After washing 3 times with 1× TBS-T, ECL Prime was added to fully coat the blots and left to incubate for 5 min at room temperature with gentle agitation. Blots were imaged using a Bio-Rad ChemiDoc XR+ imager and prepared for publication using Image Lab software (Bio-Rad) and Adobe Photoshop CC 2017. Uncropped and unprocessed Western blots are provided in the Source Data file.

## Reporting summary
Further information on research design is available in the Nature Portfolio Reporting Summary linked to this article.

## Data availability
The mean-square displacement analysis, Western blots, morphology analysis, and resection analysis data generated in this study are provided in the Supplementary Information/Source Data file. Further information and requests for reagents may be directed to and will be fulfilled by the corresponding author. Source data are provided with this paper.

## Code availability
MATLAB code is available at GitHub (https://github.com/zhflx001/auto_track)[77].

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

## Acknowledgements

We are grateful to Kerry Bloom for training in MSD measurements, Bruce Goode for donation of CK-666 and expertise, Helle Ulrich for donations of the AID and *TIR1* plasmids, David Drubin for donations of the Myo5 plasmids, and Susan Gasser for donation of strains and advice. Research was supported by NIH grant R35 GM127029. F.Y.Z., M.A., N.A., K.C., and K.B.F. were supported by NIH Genetics Training Grant TM32GM007122.

## Author contributions

Conceptualization and initial data collection were conducted by F.Y.Z. and J.E.H. Data collection, imaging, primer design, data analysis, and strain making were conducted by F.Y.Z., M.A., Y.J., N.A., K.C., and K.B.F. The manuscript was authored and edited by F.Y.Z. and J.E.H.

## Competing interests

The authors declare no competing interests.
