## [Transparent Peer Review file · Nature Communications]

Arp2/3 and Type-I myosins control chromosome mobility and end-resection at double-strand breaks in *S. cerevisiae*

Corresponding Author: Professor James Haber

Version 0:

Reviewer comments:

Reviewer #1

(Remarks to the Author)

Zhou et al., Arp2/3 and Type-I myosins control chromosome motility and end-resection at double-strand breaks in *S. cerevisiae*

In this manuscript, Zhou et al investigate the role of Arp2/3 and its nucleation promoting factors in DSB mobility and end resection. The manuscript in general uses elegant approaches, such as live imaging of DSB movement and yeast genetics to assess the effects on DNA repair pathways. However, the overall work remains largely descriptive, and the central message of the study is not always clear. The findings highlight a conserved role for the Arp2/3 actin branching complex in facilitating DSB movement and resection. While this is an important observation, similar roles for this complex have been reported in the literature, albeit in different contexts and model organisms. While evolutionary conservation of DNA repair mechanisms is valuable information, the manuscript would benefit from a more detailed exploration of the specific molecular mechanisms by which the Arp2/3 actin branching complex influences DSB movement and/or repair. Please find specific concerns below.

Major comments

- General. The link between Arp2-3 mediated-DSB mobility and end-resection remains rather inconclusive. The authors confirm the previously known connection between DSB mobility and end resection through depleting important end-resection proteins (Exo1 and Dna2) and analyzing the DSB mobility. In line with this, if the authors inhibit Arp2/3 (CK-666) or deplete Las17, DSB mobility as well as end-resection are inhibited. Surprisingly though, end resection and DSB repair by gene conversion are also decreased in the Myo3 and Myo5 double depleted background, while this genotype did not affect the DSB mobility. Even more surprisingly, the individual Myo3 or Myo5 depletions did affect DSB mobility, however depletion of these single proteins did not affect end-resection, neither did it influence DSB repair by gene conversion. These counteracting results actually suggest there is not a direct link between the impaired DSB mobility and end resection upon Arp2/3 inhibition (or depletion of nucleation promoting factors).

To be able to understand the role of the Arp2/3 complex in repair, the study would benefit from determining a clearer link between the two phenotypes: end-resection and movement. Or, if there is no clear link between the two phenotypes and for example, the effect of loss of Arp2/3 on resection is independent of DSB movement, what is the mechanism by which Arp2/3 then promotes end-resection (or DSB mobility)? These would be important questions to address to be able to add new insights to the field.

- The authors mention several times that by using the inhibitor CK-666 or by depleting Las17, Arp2/3 activity is blocked. Is the inhibitor specific for Arp2/3? And how do the authors know that 10 uM blocks the activity of Arp2/3? Would it be possible to reproduce the results using an Arp2 or Arp3 mutant / degron approach? Additionally, adding more background information on how depletion of Las17 exactly blocks the Arp2/3 activity would be helpful here.

- Fig.3E-F. The authors find that the Las17-CA mutant did not affect DSB mobility, but the Las17-WH-CA mutant did affect DSB mobility. Does a single Las17-WH mutant already display DSB mobility defects or is it truly the combined depletion of Las17-CA and WH that affects DSB mobility? Moreover, the explanation for why the Las17-CA mutant alone does not show a phenotype, but the CA + WH2 Las17-mutant does show a phenotype, remains unclear. If the CA domain is the Arp2/3 binding domain, shouldn't the CA domain mutant alone already be completely dead in activating Arp2/3? Please explain further.

- Fig.6H. Exo1 overexpression rescues the DSB movement defect of CK-666 treatment, but does not rescue Ddc2 recruitment (Fig.S5D). This would suggest that Exo1 overexpression can induce DSB movements, but does not overcome the end-resection defect of Arp2/3 inhibition. But instead the authors conclude that (p.11/line 236-237): ...'overexpression of Exo1 can overcome the reduced rate of resection caused by CK-666...' We don't think their experiments warrant this conclusion. What is the authors' explanation for this? Could the authors use a different read-out for end-resection than Ddc2 focus appearance, such as for example the analyses as done in Fig.6I to determine end-resection rates in the 'Exo1 overexpression + CK-666' condition?

Minor comments

- P.6 line 111 – 113. References to Fig.1B, G should be Fig.2B, G.
- P.5 line 80-83-88 : 'After 3h of galactose 80% of the cells have a DSB (line 80), while after 1h of galactose 90% of the cells are being cleaved (line 83)'. Why did the authors decide to measure DSB mobility 3h after adding galactose in Figure 1 (line 88)?
- Figure 1A: Would be helpful to include an image of the quantification in WT condition, e.g. showing the moving tracks?
- Figure 3A/line 120: This figure is not explained in the text.
- Fig. 3A/p.7 line 126-133. The authors employ Las17 mutants in these experiments, but the binding partners of the domains (WH2, CA) which, when mutated show a phenotype, are not indicated in the figure. It would be helpful to include Arp2/3 and actin as Las17 binding partners in Fig.3A to make it clearer for the reader.
- Line 137: Are Myo1a and Myo1b the Drosophila homologs of myo3 and myo5? If so, would be helpful to mention this.
- Fig S2. The authors determine which functional domain of Myo5 is required for DSB mobility. What do the authors mean with "MYO5 domain deletion plasmids"? And why did the authors choose to focus on Myo5, while Myo3 depletion also affects DSB mobility? And why was overexpression of Myo5 in a Myo3 depleted background tested and not the other way around (Myo3 overexpression in a Myo5 depleted background)?
- Fig.3B. There is no loading control for the western blot shown.
- Fig.4 Myo3 and Myo5 double depletion results in wild-type DSB movements, which are reduced upon loss of Las17 functionality. The authors conclude that 'las17 can compensate for the loss of both Myo3 and Myo5.'. Has redundancy between Las17 and Myo3 and Myo5 been found in other contexts as well?
- Fig.S4A: different mutants are shown (SH3del) than what is written in legend and main text (Las17-WH2-CA_{del}).
- Fig 5, 6, and 7. Statistics are missing in Fig 5, 6 and 7.
- Figure 5A: Would be helpful to include an example image.
- P.9 line 182-184 – reference to Fig.S5A/S5B-D should be S6A/B-D.
- P.9 line 178: first time mention of VRP1, but not explained in text what it is until much later in the manuscript (discussion).
- Fig.S5A VRP1 mutant is only used once in the manuscript and the authors find a reduction in Ddc2 focus appearance. Have the authors tested the effect of VRP1 loss on DSB movement? Since VRP1 binds both Las17 and Myo3/5 (fig.3A), VRP1 loss might provide insights into the differential effects of Las17 loss and myo3/myo5 depletions on DSB movement and DSB resection.
- Figure 7B: Would be helpful to include images of the morphology of the yeast. This will clarify how the morphology was assessed.
- Fig.7G -K: no legends present.
- Abstract: '...while overexpression of Exo1 suppressed resection inhibition by CK-666.' From the data shown by the authors, they can only conclude Exo1 suppresses DSB movement inhibition by CK-666 (Fig.6H), but not resection inhibition. In fact, Exo1 overexpression does not rescue the resection defect imposed by CK-666 treatment (Fig.S5D).
- Abstract: 'These results suggest that Arp2/3 regulation plays an unanticipated role in the regulation of processing of DSB ends.'. A role for Arp2/3 is not necessarily unanticipated, based on earlier results on a role for the Arp2/3-nucleator Wasp/Las17 in repair in both mammalian cells and budding yeast (ref.11), as well as in DSB movement in flies and mammalian cells (ref.6, 7).
- In general the readability and flow of the text could be improved by explaining the rationale behind certain experiments. Below some examples:
 - Line 162. The authors state "Since Las17 has a WH2 domain and acidic (CA) patch, it is possible that...". Why would this be possible?
 - Line 111: Why do the authors add CK666 before and after DSB induction? (same question for Line 193 + 194)
 - Line 154: Could the authors speculate what the data above suggest? Is it logical that the SH3 domain is important? Is this domain involved in binding to arp2/3?
 - Line 178-179: What's the rationale for depleting VRP1 and Ku70?
 - Line 204: Could the authors explain why Las17 is required for the initiation AND maintenance of resection.
 - Line 251: "G2/M arrest was shortened to 4h", could the authors speculate why the G2/M arrest was shortened?

(Remarks on code availability)

Reviewer #2

(Remarks to the Author)

Previous work in Drosophila and human cells has shown that nuclear actin is required for double-strand break (DSB)-induced chromatin mobility. In this study, the authors show that this function of actin is conserved in budding yeast. In addition, they link DSB-induced chromatin mobility to long-range end resection, DNA damage signaling and DSB repair.

The authors used HO endonuclease to generate a unique DSB in the budding yeast genome and a Ddc2-GFP fusion or a lacO array/GFP-Lacl integrated next to the HO cut site to visualize and track the DSB. Chromatin mobility was quantified by mean-square displacement (MSD) analysis. Inhibition of Arp2/3 activity by CK-666, Las17/WASP depletion, or loss of Myo3 or Myo5, resulted in reduced DSB mobility. Furthermore, depletion of Las17 reduced end resection and mutants deficient for long-range end resection (fun30 or exo1 DNA2-AID+IAA) showed reduced DSB mobility, linking end resection to mobility. Notably, the fun30 mutant is proficient for short-range end resection (700 bp from DSB) suggesting that kbs of single-stranded DNA (ssDNA) are required to mobilize DSBs. The reduction in end resection by CK-666 or Las17 depletion also prevented the Mec1-dependent DNA damage checkpoint. A key question in the field is whether DSB mobility is important for the homology search. The authors addressed this question by measuring inter-chromosomal gene conversion in cells depleted for Las17 or myosins. Although depletion of Las17 reduced inter-chromosomal gene conversion, especially using strains with low contact probability donors, the myo3 and myo5 mutants showed normal HR. The low frequency of inter-chromosomal HR for strains defective for checkpoint signaling and long-range end resection is consistent with a recent study by Kimble et al. (2023). The myo3 and myo5 results suggest that DSB mobility does not contribute significantly to HR efficiency, an interesting and unexpected finding.

One of the more confusing results in the manuscript is the finding that DSB mobility is normal in cells simultaneously depleted for Myo3 and Myo5. The authors suggest that Las17 can compensate for loss of both myosins, but it is unclear why Las17 would not compensate for loss of only one. Furthermore, the authors show that end resection is normal in myo3 and myo5 mutants, even though DSB mobility is compromised, whereas end resection is defective in the myo3 Myo5-AID strain where DSB mobility is restored. These findings raise the question of how (and if) end resection and DSB mobility are linked.

Main comments:

1. Figure 1: In Figure 1A, CK-666 added 2 h 40 min after HO induction reduced chromatin mobility. Was end resection by the qPCR-based method used under these conditions and was the percent cells with Ddc2-GFP and Rad51-GFP foci determined? The data in Figure S5 show the impact of CK-666 added 20 min before HO induction. There should be considerable end resection 2 h 40 min after HO induction suggesting a role for Arp2/3 in mobility independent of end resection.
2. Figure 3: Does the WH2 deletion alone impact DSB mobility?
3. Was chromatin mobility measured in myo3 and myo5 strains before HO induction using the lacO/GFP-Lacl system? Maybe there is a lower baseline level of chromatin mobility in these mutants that can be increased by a DSB.
4. Interestingly, over-expression of Exo1 was able to overcome the inhibitory effect of CK-666 on DSB mobility when the inhibitor was added 2 h and 40 min after HO induction. However, end resection was not measured by the qPCR assay under these conditions. The Ddc2 foci data in Figure S5 are for CK-666 added 20 min before HO induction. If CK-666 is added before Gal induction can Gal:Exo1 restore DSB mobility? If Exo1 over-expression is not increasing end resection how might it impact DSB mobility?
5. Can the HR defect of the Las17-AID strain be rescued by holding cells in nocodazole to maintain cell cycle arrest?
6. Is mating-type switching, or another HR events that does not provoke the DNA damage checkpoint, normal when Las17 is depleted?

Minor comments:

Line 122: change Lad17 to Las17.

Figure S2E: The Rc values of WT and myo5 are indicated as not significantly different contradicting data shown in Figure 4C. Is this an error?

Figure 6D,E: Is there a reason for adding IAA 2 h after galactose induction instead of 20 min after inducing HO? Was resection measured by qPCR in these cells.

Figure 6H: Are the Rc values of WT + CK-666 and pGal:EXO1 + CK-666 significantly different?

Figure 7: Labeling of the figure parts does not match the legend.

Figure 8B: Survival of the mre11 mutant seems high.

Figure 8D, E: It would be helpful to have a schematic in a supplementary figure showing the location of recipient and donor sites in the genome.

(Remarks on code availability)

Reviewer #3

(Remarks to the Author)

(Remarks on code availability)

Reviewer #4

(Remarks to the Author)

In their manuscript, the authors Zhou et al. investigate the role of the Arp2/3 complex and its nucleation promoting factors (NPFs) in regulating chromatin mobility and end-resection at double-strand breaks in yeast. In addition to what has already been described in mammalian cells, *Drosophila* and *Xenopus laevis* extracts, they find that the Arp2/3 complex also regulates DSB mobility and end-resection in yeast, suggesting an evolutionary conserved function. The authors examined this by using pharmacological inhibition of Arp2/3 or by degrading the NPFs Las17, Myo3 and Myo5.

In the abstract, they state that their results reveal an “unanticipated role for Arp2/3 in the regulation of DSB mobility”.

However, given that similar roles for the Arp2/3 complex in DSB mobility, clustering and resection have already been demonstrated in mammalian cells (Schrank et al, 2018) and in the relocalization of breaks from heterochromatic regions to the nuclear periphery in *Drosophila* (Caridi et al, 2018), these findings may not be as surprising as suggested.

The authors suggest that Las17 and Arp2/3, in their role as actin nucleators, are required for DSB mobility. However, they do not directly demonstrate the involvement of (branched) actin filaments or active movement of the DSBs, nor do they perturb (nuclear) actin filament formation in any other way. This leaves several open questions about the precise mechanism by which the Arp2/3 complex is involved in the processes under investigation.

Overall, their results seem to reinforce the idea of an evolutionarily conserved role for the Arp2/3 complex, rather than introducing a completely novel function. Therefore, the manuscript may be better suited for a different journal.

Major points:

1. Reintroduction of a second copy of Myo5 into the Myo3del mutant appears to rescue the DSB mobility (Fig. 4). Does this also work the other way round, meaning reintroduction of a second Myo3 copy into the Myo5del strain? Also, either specifically deleting the type I myosins in the yeast nucleus or reintroduction of a nuclear targeted version of Myo3/Myo5 in a deletion strain would strengthen the results, as it would highlight the nuclear-specific role for the studied processes.
2. How do the authors explain the unexpected result that deletion of both Myo3 and Myo5 can restore the DSB mobility (as opposed to single deletion)? Why is Las17 only able to rescue the phenotype of the double deletion and not the single deletions?
3. In line 330 of the discussion, the authors conclude that “a defect in long-range end-resection is sufficient to impair damage-induced chromatin mobility.” However, the single deletion of either Myo3 or Myo5 did not greatly affect resection, yet it still led to reduced DSB mobility, as the authors correctly acknowledge later in lines 384-386. Therefore, this claim may be somewhat overstated at this point.

Minor points:

1. The concentration of 10 μ M CK666 used for their assays appears to be rather low. It would be helpful to show, that Arp2/3 function is indeed inhibited at this concentration. Also, a good control for CK666 treatment would be to include the inactive CK-689 compound.

Generally though, it would be really important to achieve nuclear-specific inhibition of Arp2/3, like nuclear uncaring of the inhibitor or depletion of Arp2/3 and reintroduction of a cytoplasmic derivative. The field heavily suffers from the global use of CK666 and its over interpretation for nuclear effects. How can one exclude that all these effects arise from cytoplasmic Arp2/3 inhibition?

2. There should be a space between the numerical value and the unit symbol for concentrations (e.g. line 92 “10 μ M”, line 122 “1 mM, etc.).
3. Lines 98-102: The rationale for the experiments shown in Fig 1 D and E is not very clear as endocytosis takes place at the plasma membrane whereas DSB processing is happening in the nucleus. How would these processes be connected? Also, the authors state that “deletion of SLA2, a key component of endocytosis, did not affect the behavior of the DSB (Figure 1D-E).” However, to be more precise, I suggest rephrasing it to “did not significantly affect...” since there appears to be a slight decrease in MSD.
4. Lines 111 and 113: Wrong Figure callout; should be Fig. 2B instead of 1B.
5. Line 120: Figure 3A should also be referenced in the text where Vrp1 and Myo3/Myo5 and its domains are discussed.
6. Line 178: The abbreviation and function of Vrp1 should be explained the first time its mentioned in the manuscript. Also, it might be helpful to reference Figure 3A at this point.
7. Lines 193-194: “Resection in cells treated with CK-666 after 1 h after DSB induction was also significantly impaired.” While the data suggest this, no statistical test has been performed to support the claim. The statement should either be rephrased or a statistical test should be conducted to confirm the significance of the observed differences.
8. Line 252: A reference for Figure 7E is wrongly included at this position of the text or wrongly labeled (myo5del instead of CK666 treatment).
9. Line 285: A series of strains that had been previously characterized with repair efficiencies ranging from 46% to 9% is mentioned. However, further explanation or description of the individual strains would be helpful for the reader at this point.
10. Line 288: “...Las17-AID + IAA showed a significant decrease in the percentage of survivors in all 4 strains.” There are no statistical tests performed for these data nor for any other data presented in Fig. 8. The sentence should either be rephrased or significance tested.
11. Line 294: Please introduce the abbreviation for break-induced replication (BIR) here.
12. The authors should specify the statistical tests performed for each analysis directly in each figure legend, rather than referencing previous figure legends. (e.g. in Fig legend 1B and E). Also, the number of independent biological replicates should be clearly stated in the figure legends for transparency.
13. The authors refer to their quantification in Figures 1-4, 6 and 8 as “boxplots” even though the data are presented as bar graphs. The labelling needs to be corrected, or preferably, boxplots should be displayed instead of bar graphs.
14. Fig 2D: There is a mislabeling in the figure legend. Panel D shows data for the treatment group “cut + CK666 before gal, 3h”, while the legend incorrectly states “cut + CK-666 after gal, 3h”. It appears that the labeling (or figure panels) between D and F may have been mixed up and should be corrected for clarity.
15. Fig. 2: It appears that the same data for the treatment groups “cut, 3h” and “uncut, 0h” have been used in panels B, D and

F, as the trend of the graphs are exactly the same. If the authors prefer to show them in different graphs due to better clarity of the data, they should be more transparent about this and clearly state it in the figure legend.

16. Fig 2G: The authors reference (B) for the statistical test used, however, there is no test performed for Fig. 2B. I assume they reference Fig. 1B. Also, instead of a t-test an ANOVA should be performed as more than 2 groups are being compared.

17. Fig. 3D, F: Statistical test should be mentioned in the figure legend and ANOVA should be used instead for multiple comparison of data.

18. Fig. 5 C-E: Legend missing for y-axis labeling. What exactly is "Fresected"?

19. Fig 7: Panel labeling does not match the figure legends.

20. Fig. 8C is not referenced in the text.

(Remarks on code availability)

Version 1:

Reviewer comments:

Reviewer #1

(Remarks to the Author)

We appreciate the significant effort the authors have made to address our concerns, and we acknowledge that the manuscript has improved considerably in response to our previous feedback. The authors have successfully resolved several of the key issues initially raised. While we still find the data to be primarily descriptive and feel that the study does not deliver a conclusive mechanistic model for how the Arp2/3 complex facilitates DSB movement and repair, the new findings do offer valuable insights. In particular, the observation that checkpoint activation defects, rather than impaired DSB mobility per se, may be more central to the repair deficiency in the absence of Arp2/3 activity is intriguing. The text, however, remains quite technical and may be challenging for readers outside the immediate field. Additionally, some references to figures in the main text are incorrect. We would encourage the authors to verify these references and to consider input from a colleague less familiar with the field to improve clarity and accessibility of the text.

We have a few comments left on this revised manuscript.

1) We are not fully satisfied with one answer to one of our previous comments ('=Fig.6H: Exo1 overexpression.... + CK-666' condition?') in which we asked to better determine the effect of Exo1 overexpression on Arp2/3 inhibition with CK-666. The authors now performed resection analyses with overexpressing exo1 in the LAS17-AID + IAA mutant and they find a rescue (Fig.6I). The authors therefore now conclude in line 383: 'We also found that EXO1 overexpression suppressed the reduction in Rc caused by adding CK-666 by maintaining wildtype levels of resection.'

However, we are still of the opinion that the rescue of Exo1 overexpression on DSB movement and resection in the presence of CK-666 remains rather inconclusive. We feel there are two separate pieces of data: On the one hand, Exo1 overexpression rescues the CK-666-induced movement defect (Fig.6G), but not the defect in end-resection (reduced Ddc1-GFP foci formation, Fig. S5D). On the other hand, the authors find that Exo1 overexpression rescues resection in the Las17-AID + IAA background (Fig.6I), but movement was not tested in this background as far as we can tell.

Therefore, while the authors did test the effect of Exo1 overexpression on DSB movement upon CK-666 treatment, they did not test this effect in the LAS17-AID + IAA background (where they find a rescue of the resection defect). We would therefore suggest to either test the DSB movement effect of Exo1 overexpression in the Las17-AID + IAA background or tone down their conclusion that EXO1 overexpression suppressed the reduction in Rc caused by CK-666 by maintaining wildtype levels of resection.'

2) Although the authors now included that the assays shown in Fig.5-7 are averages + SEM, they did not include statistical tests for the line graphs in figure 5B, 5C, 5D, 6A, 6H, 6I, 7B, 7D, 7F, 7H, 7J, 8C, 8H.

3) Line 140: Please include what the abbreviation WH2 stands for.

4) Line 403-404: "We show that the highly conserved C-terminal region across WASP orthologs (WH2 and CA domains) of Las17 is required for DSB mobility". The authors show that only the WH2 domain is required for DSB mobility, not the CA domain and would have to revise this conclusion.

Reviewer #2

(Remarks to the Author)

The authors have adequately addressed my concerns. One minor comment, the vpr1 data are presented in Fig S5A not Fig 3A (line 198).

Reviewer #3

(Remarks to the Author)

Reviewer #5

(Remarks to the Author)

> I was asked to step in for the previous Reviewer #4 to assess the authors' responses to the reviewers' comments. I will therefore go through the comments point by point:<

Overall, their results seem to reinforce the idea of an evolutionarily conserved role for the Arp2/3 complex, rather than introducing a completely novel function. Therefore, the manuscript may be better suited for a different journal.

We beg to differ with the reviewer on several grounds. We agree that we have shown clearly that Arp2/3 plays a key role in DSB-induced chromosome mobility in yeast, as in mammals and flies. Also, in mammalian cells there was an effect on RPA focus formation that suggested a role in 5' to 3' resection. Here we have documented the effect on resection in detail and shown that Arp2/3 is necessary both to initiate long-range resection and to maintain it (and that these two phases are distinct). Moreover, we show that continuing resection is essential for the increased radius of confinement and that just having a long ssDNA tail is not sufficient. We go further to show that mobility is tied to continuing resection by (a) inactivating the long-range resection machinery itself and (b) overexpressing Exo1 in a Las17-inactivated state. In addition, we show that higher mobility per se is not the key factor in facilitating ectopic recombination; it is instead the activation of the DNA damage checkpoint to prevent cells from dividing before repair is complete. We show this by improving repair by holding Arp2/3-inactivated cells in nocodazole for 6 h. Also we show that inactivating Arp2/3 by CK-666 or inactivating Las17 triggers and "TM" checkpoint that has only been shown to be induced by mutations in Rad50 and Sae2.

> I do agree with the authors that their study provides more value than the demonstration of an evolutionary conserved role of the ARP2/3 complex in DSB repair.

Zhou and colleagues use beautiful yeast genetics and imaging analyses to address the relationship between actin branching, DNA end resection and chromatin mobility upon DSB induction in yeast. They provide novel insight into the role of ARP2/3's activity for the processing of DNA ends, checkpoint activation and DSB repair. The data are of high technical quality and well presented with a good structure making the manuscript easy to read. Their findings pave the way for future studies where the model organism *Saccharomyces cerevisiae* can be utilized to investigate basic molecular mechanisms of actin cytoskeletal functions in genome stability. I consider the manuscript well suited for publication in *Nature Communications*.<

Major points:

1. Reintroduction of a second copy of Myo5 into the Myo3del mutant appears to rescue the DSB mobility (Fig. 4). Does this also work the other way round, meaning reintroduction of a second Myo3 copy into the Myo5del strain? Also, either specifically deleting the type I myosins in the yeast nucleus or reintroduction of a nuclear targeted version of Myo3/Myo5 in a deletion strain would strengthen the results, as it would highlight the nuclear-specific role for the studied processes. We have now shown that two copies of Myo3 compensate for the deletion of Myo5.

> While the authors replied to the first part of the question, they missed addressing the second part.

The previous reviewer #4 asked to study if the observed phenotype for type I myosins in DSB mobility is specific for their nuclear pool by either (1) deleting the nuclear Myo3/5 pool or (2) reconstituting the respective deletions with a nuclear targeted variant. I would like to ask the authors to address this point.<

2. How do the authors explain the unexpected result that deletion of both Myo3 and Myo5 can restore the DSB mobility (as opposed to single deletion)? Why is Las17 only able to rescue the phenotype of the double deletion and not the single deletions?

As noted above, a recent pre-print showed that type-I myosins have a role in nuclear organization [21]. It should be noted that Myo3 was depleted in a *myo5Δ* background and Myo5 was depleted in a *myo3Δ* background, so either Myo3 or Myo5 alone was sufficient for maintaining genome organization. Depletion of both type-I myosins from the nucleus significantly affected cell growth, caused changes in genome organization, and changed the shape of the nuclear envelope. The dramatic changes in chromosome arrangement and nuclear properties most likely explain our unexpected finding that the double mutant had a higher radius of confinement.

Although we found that inactivating Las17 decrease the unexpected high radius of confinement in the *myo3Δ myo5Δ* double depletion, we have chosen to remove this experiment, given that the organization of the nucleus in the *myo3 myo5* double depletion is aberrant. We confirmed this exceptional behavior by measuring a change in the MSD of the Spindle Pole Bodies relative to the cell's bud neck.

> This issue was also raised by the other reviewers and both comments were properly addressed.<

3. In line 330 of the discussion, the authors conclude that "a defect in long-range end-resection is sufficient to impair damage-induced chromatin mobility." However, the single deletion of either Myo3 or Myo5 did not greatly affect resection, yet it still led to reduced DSB mobility, as the authors correctly acknowledge later in lines 384-386. Therefore, this claim may be somewhat overstated at this point.

We have addressed this point for Reviewer 1. Slowing down the rate of resection with the *fun30* deletion mutant or stopping long-range resection by getting rid of Exo1 and Dna2 lowered the mobility of the DSBs. Increasing resection by overexpressing Exo1 was able to compensate for the loss of Arp2/3 activity further strengthening the link between the rate of resection and DSB mobility. Single deletions of Myo3 and Myo5 did not affect the rate of resection, nor did they impair the efficiency of repair.

The rate of resection (4 kb/h) is one factor that contributes to DSB mobility. In budding yeast, it has been shown that activation of the DNA damage checkpoint through *Ddc1/Ddc2* targeting to a chromatin locus [22] was sufficient to increase

chromatin mobility [23]. In this instance there was no resection of DNA ends, but increased Mec1 activity and phosphorylation of Rad9 and Rad53 coincided with increased chromatin mobility [23]. It is unclear why deleting myo3 or myo5 lowers DSB mobility when resection is unperturbed (Figure 5D) and Rad53 is still readily phosphorylated (Figure 7E).

> This point was also raised by the other reviewers and was properly addressed by the authors.<

Minor points:

1. The concentration of 10 μM CK666 used for their assays appears to be rather low. It would be helpful to show, that Arp2/3 function is indeed inhibited at this concentration. Also, a good control for CK666 treatment would be to include the inactive CK-689 compound. Generally though, it would be really important to achieve nuclear-specific inhibition of Arp2/3, like nuclear uncaring of the inhibitor or depletion of Arp2/3 and reintroduction of a cytoplasmic derivative. The field heavily suffers from the global use of CK666 and its over interpretation for nuclear effects. How can one exclude that all these effects arise from cytoplasmic Arp2/3 inhibition?

The concentration of CK-666 was actually 100 μM and was incorrectly reported. We apologize for the mistake.

In budding yeast the primary role of Arp2/3 in the cytoplasm is in endocytosis. Our *sla2* deletion mutant, which is defective in cell membrane invagination [14], did not exhibit any defect in DSB mobility, suggesting that if we were able to specifically inhibit Arp2/3's primary role in endocytosis in the cytoplasm while maintaining its role in the nucleus, it would not affect mobility of the DSB.

> The CK666 concentration was corrected by the authors. However, I do not see that the authors have used the inactive CK-689 compound as negative control in one of their assays. Therefore, I would like to ask the authors to show this control as it was requested by the previous reviewer. Regarding the second point of a nuclear-specific inhibition of Arp2/3, I understand the reviewer's concern and the current "sloppiness" in the field in terms of claiming nucleus-specific effects upon usage of a compartment unspecific drug. However, the authors do not claim compartment specific inhibition or localization of the Arp2/3 complex throughout their entire manuscript and provide the *Sla2* endocytosis control, which covers the major function of cytoplasmic Arp2/3. Secondly, they were asked to provide a nucleus-specific function of the type I myosins (see above) as a major point raised by the previous reviewer which will also address this question. In that context and with the notion that the previous reviewer #4 had listed this as a minor point, I do not insist on a further experimental revision here.<

2. There should be a space between the numerical value and the unit symbol for concentrations (e.g. line 92 "10 μM ", line 122 "1 mM, etc.).

Text has been updated.

>Fine<

3. Lines 98-102: The rationale for the experiments shown in Fig 1 D and E is not very clear as endocytosis takes place at the plasma membrane whereas DSB processing is happening in the nucleus. How would these processes be connected? Also, the authors state that "deletion of *SLA2*, a key component of endocytosis, did not affect the behavior of the DSB (Figure 1D-E)." However, to be more precise, I suggest rephrasing it to "did not significantly affect..." since there appears to be a slight decrease in MSD.

We state that Arp2/3 is primarily studied for its role in endocytosis, and we asked whether Arp2/3 dependent DSB mobility was linked to the role of Arp2/3 in endocytosis. To answer this question we targeted another protein, *Sla2*, which is important for forming pre-endocytic patches and measured DSB mobility in this deletion mutant.

We have updated the text to note that there is a statistically insignificant decrease in the *Rc* of the *sla2* deletion mutant (Figure 1A and Table S1).

>Fine<

4. Lines 111 and 113: Wrong Figure callout; should be Fig. 2B instead of 1B.

Text has been updated.

>Fine<

5. Line 120: Figure 3A should also be referenced in the text where *Vrp1* and *Myo3/Myo5* and its domains are discussed.

Text has been updated to reference Figure 3A when referring to *Vrp1* and the functional domains of *Myo5*.

>Fine<

6. Line 178: The abbreviation and function of *Vrp1* should be explained the first time its mentioned in the manuscript. Also, it might be helpful to reference Figure 3A at this point.

Text has been updated to show the full name and primary function of verprolin (*Vrp1*).

>Fine<

7. Lines 193-194: "Resection in cells treated with CK-666 after 1 h after DSB induction was also significantly impaired."

While the data suggest this, no statistical test has been performed to support the claim. The statement should either be rephrased or a statistical test should be conducted to confirm the significance of the observed differences.

Text has been rephrased to add clarity.

> Fine – could not find this part, most likely re-written.<

8. Line 252: A reference for Figure 7E is wrongly included at this position of the text or wrongly labeled (myo5del instead of CK666 treatment).

Figure 7E is a western blot probing for Rad53 and Pgc1 in a myo5 Δ strain. We did not run a western blot of CK-666 treated cells due to the amount of drug that we would need to run this experiment. Instead, we used the Las17-AID mutant \pm IAA to test how disrupting Arp2/3 activity before DSB induction affected activation of the DNA damage checkpoint.

>Fine<

9. Line 285: A series of strains that had been previously characterized with repair efficiencies ranging from 46% to 9% is mentioned. However, further explanation or description of the individual strains would be helpful for the reader at this point. We added a description of the strains used as well as a map of where the HO-cut site is and where the various LEU2 donors are in supplemental Figure S8.

>Fine<

10. Line 288: "...Las17-AID + IAA showed a significant decrease in the percentage of survivors in all 4 strains." There are no statistical tests performed for these data nor for any other data presented in Fig. 8. The sentence should either be rephrased or significance tested.

This figure has been moved to Figure S9 and divided into 4 graphs based on the background strain. A one-way Anova was performed to show that the Las17-AID + IAA is significantly different than Las17-AID not treated with IAA and the WT (Figure S9G-J).

>Fine<

11. Line 294: Please introduce the abbreviation for break-induced replication (BIR) here.
Text has been updated.

>Fine<

12. The authors should specify the statistical tests performed for each analysis directly in each figure legend, rather than referencing previous figure legends. (e.g. in Fig legend 1B and E). Also, the number of independent biological replicates should be clearly stated in the figure legends for transparency.

For MSD curves and their accompanying Rc graphs the number of biological replicates is included after the strain name in the figure legend e.g. Figure 1A WT (n=14) and WT with CK-666 (n=34).

The statistical tests used in the Rc graphs have been included in the figure legends and are also listed in Table S1 along with the P value and significance.

>Fine<

13. The authors refer to their quantification in Figures 1-4, 6 and 8 as "boxplots" even though the data are presented as bar graphs. The labelling needs to be corrected, or preferably, boxplots should be displayed instead of bar graphs.
Legends have been updated.

>Fine<

14. Fig 2D: There is a mislabeling in the figure legend. Panel D shows data for the treatment group "cut + CK666 before gal, 3h", while the legend incorrectly states "cut + CK-666 after gal, 3h". It appears that the labeling (or figure panels) between D and F may have been mixed up and should be corrected for clarity.

Text has been updated so the legends match their appropriate graph.

>Fine<

15. Fig. 2: It appears that the same data for the treatment groups "cut, 3h" and "uncut, 0h" have been used in panels B, D and F, as the trend of the graphs are exactly the same. If the authors prefer to show them in different graphs due to better clarity of the data, they should be more transparent about this and clearly state it in the figure legend.

The "uncut, 0h" and "cut, 3h" are our wildtype data, confirming what others have reported that following a DSB there is an increase in chromatin mobility near a DSB site [24, 25]. We note that these are the same data.

These data are shown in Figures 2B, 2D, and 2F to compare how CK-666 affected baseline levels of chromatin mobility and chromatin mobility after DSB induction with Gal-HO where CK-666 was added either before or after galactose.

> Although the authors explain the usage of the same data for the "uncut, 0h" and "cut, 3h" for all three panels in their response to the reviewer #4, this statement is still missing from the figure legend of Figure 2. Please add.<

16. Fig 2G: The authors reference (B) for the statistical test used, however, there is no test performed for Fig. 2B. I assume they reference Fig. 1B. Also, instead of a t-test an ANOVA should be performed as more than 2 groups are being compared. The legend now includes which statistical test was used to analyze significance in Rc value and percentage of survivors.

This information is also included in Tables S1 and S3.

>Fine<

17. Fig. 3D, F: Statistical test should be mentioned in the figure legend and ANOVA should be used instead for multiple comparison of data.

The statistical tests used for each comparison is now listed in each figure legend and in Table S1 and S3 along with the P value and which correctional test was used.

>Fine<

18. Fig. 5 C-E: Legend missing for y-axis labeling. What exactly is "Fresected"?

Fresected is the fraction of resected DNA and is the unit used to measure how much resected DNA is measured through the restriction enzyme based qPCR assay we used as illustrated in Figure 5A and described in Grugge et al. Processing the DNA Double-Strand Breaks in Yeast Methods Enzymology [26].

>Fine<

19. Fig 7: Panel labeling does not match the figure legends.
Updated legend.

>Fine<

20. Fig. 8C is not referenced in the text.

Figure 8C has been moved to Figure S8A and is referenced in the text along with Figure S8B to describe the series of strains with the donor for homologous recombination moved to different locations in the genome.

>Fine – There are no Figure S8A and S8B; only Figure S8, but Figures S9A and B show the series of strains which are mentioned here. Figure S8 and S9A/B are correctly referenced in the main text.<

> New comments:

1. There is a problem with Figure 6I where the 10 kb plot is shown twice.

2. Furthermore, in the main text, references do not align in line 257 where Figure 6I is referenced but should say Figure 6H. In line 263 it should say Figure 6I instead of Figure 6J and it seems that all consecutive references for Figure 6 are misaligned plus there is a reference to Figure 6M in line 266 where the corresponding Figure does not exist.<

Response to reviewers

We thank the reviewers for their insightful comments. We have endeavored to address all of the points they raise in the revised manuscript and respond specifically to each point below

Reviewer #1 (Remarks to the Author)

In this manuscript, Zhou et al investigate the role of Arp2/3 and its nucleation promoting factors in DSB mobility and end resection. The manuscript in general uses elegant approaches, such as live imaging of DSB movement and yeast genetics to assess the effects on DNA repair pathways. However, the overall work remains largely descriptive, and the central message of the study is not always clear. The findings highlight a conserved role for the Arp2/3 actin branching complex in facilitating DSB movement and resection. While this is an important observation, similar roles for this complex have been reported in the literature, albeit in different contexts and model organisms. While evolutionary conservation of DNA repair mechanisms is valuable information, the manuscript would benefit from a more detailed exploration of the specific molecular mechanisms by which the Arp2/3 actin branching complex influences DSB movement and/or repair. Please find specific concerns below.

Major comments

- General. The link between Arp2-3 mediated-DSB mobility and end-resection remains rather inconclusive. The authors confirm the previously known connection between DSB mobility and end resection through depleting important end-resection proteins (Exo1 and Dna2) and analyzing the DSB mobility. In line with this, if the authors inhibit Arp2/3 (CK-666) or deplete Las17, DSB mobility as well as end-resection are inhibited. Surprisingly though, end resection and DSB repair by gene conversion are also decreased in the Myo3 and Myo5 double depleted background, while this genotype did not affect the DSB mobility. Even more surprisingly, the individual Myo3 or Myo5 depletions did affect DSB mobility, however depletion of these single proteins did not affect end-resection, neither did it influence DSB repair by gene conversion. These counteracting results actually suggest there is not a direct link between the impaired DSB mobility and end resection upon Arp2/3 inhibition (or depletion of nucleation promoting factors).

Response:

We believe we have now addressed these concerns. We conclude that the key step that affects ectopic recombinational repair of a DSB is not the mobility of the broken ends *per se*, but the role of the Arp2/3-Las17-Myo1 proteins in end-resection and in the activation of the DNA damage checkpoint. Thus, while deleting Myo3 or Myo5 individually reduces the radius of confinement, these deletions do not impair resection and the DNA damage checkpoint is activated normally, giving cells the time to accomplish repair. In contrast, inactivating Las17 or adding CK-666 to inhibit Arp2/3 not only reduces mobility but also impairs resection and shortens the DNA damage checkpoint. Further, by eliminating short-distance resection in addition to Arp2/3 or Las17 inactivation almost completely eliminates both resection and G2/M arrest. So, we conclude that impairing mobility *per se* is not a barrier to repair, while impairing resection and checkpoint activation is critical.

How then to account for the inactivation of both Myo3 and Myo5? A recent paper from Brian Freeman (doi: <https://doi.org/10.1101/2024.09.26.615191>) has enlightened us: using an anchors-away approach, these authors depleted Myo3 and/or Myo5 from the nucleus. They find that depleting both type I myosins has a dramatic effect on the nucleus; this “triggered 3D genome disorganization, nucleolar disruption, broad gene expression changes, and nuclear membrane morphology collapse. Hence it is likely that the changes in the radius of confinement that we observe reflect these dramatic changes in the clustering of centromeres and other effects on chromosome behavior.

To test whether there was a significant change in chromatin organization when both Myo3 and Myo5 are inactivated, we performed MSD analysis of the mobility of the spindle pole bodies (SPBs) relative to the bud neck to see if there were global changes in nuclear organization. The MSD of the SPBs was not affected by CK-666 or deleting *myo5* alone (**Figure S1A-B, S4A-B**). However, when both type-I myosins are depleted, the MSD of the SPBs relative to the bud neck increases (**Figure S4C-D**). We interpret this finding as evidence that the organization of the nucleus and likely behavior of chromosomes is exceptional when both type I myosins are inactivated.

Although in the future it would be possible to use the anchors-away approach in our assays, this would take us many months to accomplish, as we would have to introduce all the elements of the FKBP system (Tor1 mutation, tagged ribosomal and myosin proteins, changes of markers of mutations we already use, etc.).

- *The authors mention several times that by using the inhibitor CK-666 or by depleting Las17, Arp2/3 activity is blocked. Is the inhibitor specific for Arp2/3? And how do the authors know that 10 μ M blocks the activity of Arp2/3? Would it be possible to reproduce the results using an Arp2 or Arp3 mutant / degenon approach? Additionally, adding more background information on how the depletion of Las17 exactly blocks the Arp2/3 activity would be helpful here.*

CK-666 directly binds to Arp2/3 to prevent it from switching from its inactive to active state [1]. In the paper we incorrectly reported that we used 10 μ M of CK-666. We used 100 μ M of CK-666 which has been shown to inhibit Arp2/3 activity in budding yeast as shown in Guo et al. [2]. We apologize for this error in reporting the concentration.

Las17 is the yeast homologue for the human protein WASP which is required for Arp2/3 activity in human cells [3]. We believe that the fact that CK-666 has similar phenotypes as inactivating Las17 has addressed this issue. Moreover, having shown that inactivating Arp2/3 or Las17 impairs 5' to 3' resection, we then show that directly impairing the resection apparatus reflects the inactivation of Arp2/3 and nucleation promoting factors.

- *Fig.3E-F. The authors find that the Las17-CA mutant did not affect DSB mobility, but the Las17-WH-CA mutant did affect DSB mobility. Does a single Las17-WH mutant already display DSB mobility defects or is it truly the combined depletion of Las17-CA and WH that affects DSB mobility? Moreover, the explanation for why the Las17-CA mutant alone does not show a phenotype, but the CA + WH2 Las17-mutant does*

show a phenotype, remains unclear. If the CA domain is the Arp2/3 binding domain, shouldn't the CA domain mutant alone already be completely dead in activating Arp2/3? Please explain further.

We now have deleted only the WH2 domain of Las17 and find that the MSD of the DSB decreased to *las17-WH2CAΔ* levels. This suggests that the WH2 domain of Las17 contributes to DSB mobility and the CA domain of Las17 may be dispensable for this function.

Mutations in the WH2 and CA domains of Las17 do not have the same effect in endocytosis. The CA domains of Myo3 and Myo5 are functionally redundant with the CA domain of Las17 [4]. Hence, it is possible that the CA domain of Las17 can be compensated by these other Cas domains. As noted above, removing both Myo3 and Myo5 from the nucleus creates a more severe phenotype that may well mask and attempt to delete these domains. In endocytosis, deletion of the CA domain of Las17 delays the invagination of endocytic patches, but does not impede invagination once it begins [5]. The WH2 domain of Las17 binds to the subdomain 3 of g-actin and a *las17-I555D* mutant in WH2 was defective in the invagination stage [5, 6].

- Fig.6H. Exo1 overexpression rescues the DSB movement defect of CK-666 treatment, but does not rescue Ddc2 recruitment (Fig.S5D). This would suggest that Exo1 overexpression can induce DSB movements, but does not overcome the end-resection defect of Arp2/3 inhibition. But instead the authors conclude that (p.11/line 236-237): ...'overexpression of Exo1 can overcome the reduced rate of resection caused by CK-666...' We don't think their experiments warrant this conclusion. What is the authors' explanation for this? Could the authors use a different read-out for end-resection than Ddc2 focus appearance, such as for example the analyses as done in Fig.6I to determine end-resection rates in the 'Exo1 overexpression + CK-666' condition?

In addition to loss of the Ddc2-GFP signal we now have measured resection by qPCR when Exo1 is overexpressed when Arp2/3 is inactivated, either before or after DSB induction. Without IAA, the *Las17-AID pGal:EXO1* double mutant exhibited a higher rate of resection than the wildtype, much like the *pGal:EXO1* strain. When *Las17-AID* was degraded *after* adding galactose, the rate of resection was reduced to wildtype levels (that is, *pGal::Exo1* increased the rate of resection compared to just inactivating *Las17*). However, when *las17* was blocked *before* DSB induction, *Exo1* overexpression did not overcome the inhibition. These results, along with the other data in this paper, suggest that Arp2/3 activity is required both to initiate resection and to sustain it. These are apparently different activities, reminiscent of the difference of the *las17-CA* deletion in initiating and sustaining invagination of endocytic patches. *EXO1* overexpression is unable to overcome this barrier to starting resection; however once resection has begun, overexpression of *EXO1* facilitates resection in the absence of Arp2/3 activity.

Minor comments

1. P.6 line 111 – 113. References to Fig.1B, G should be Fig.2B, G.

- We have revised the text

2. P.5 line 80-83-88 : ‘After 3h of galactose 80% of the cells have a DSB (line 80), while after 1h of galactose 90% of the cells are being cleaved (line 83)’. Why did the authors decide to measure DSB mobility 3h after adding galactose in Figure 1 (line 88)?
 - As shown in our 2019 paper characterizing Ddc2-GFP and Rad51-GFP damage-dependent foci formation, we found that the percentage of cells with a Ddc2-GFP focus at the DSB peaked 3 h after DSB induction with Gal-HO [6]. This led us to believe that by 3 h after adding galactose, our cells had undergone sufficient resection, loading of Ddc2, and had begun the search for homology.
3. Figure 1A: Would be helpful to include an image of the quantification in WT condition, e.g. showing the moving tracks?
 - We added a series of images from a time course tracking a Ddc2-GFP focus and the spindle pole bodies labeled with Spc42-mCherry in a single cell (**Figure 1A**).
4. Figure 3A/line 120: This figure is not explained in the text.
 - Text has been updated to explain that these are the known interactions between the nucleation promoting factors Las17 and the type-I myosins Myo3 and Myo5. The manuscript has been updated to refer back to this figure when talking about the different functional domains of these proteins.
5. Fig. 3A/p.7 line 126-133. The authors employ Las17 mutants in these experiments, but the binding partners of the domains (WH2, CA) which, when mutated show a phenotype, are not indicated in the figure. It would be helpful to include Arp2/3 and actin as Las17 binding partners in Fig.3A to make it clearer for the reader.
 - For the *las17-WH2CAA* and *las17-CAA* mutants we added models to illustrate which domains have been deleted.
 - The model in 3A showing the interactions between the NPFs Las17, Vrp1, Myo3, and Myo5 has been updated to show that the WH2 and CA domains of Las17 binds to monomeric g-actin and directly bind to Arp2/3 respectively.
6. Line 137: Are Myo1a and Myo1b the *Drosophila* homologs of myo3 and myo5? If so, would be helpful to mention this.
 - Myo1a and Myo1b are type-I myosins in *Drosophila* and Myo3 and Myo5 are the only type-I myosins in budding yeast. The text has been updated to clarify that these are all type-I myosins.
7. Fig S2. The authors determine which functional domain of Myo5 is required for DSB mobility. What do the authors mean with “MYO5 domain deletion plasmids”? And why did the authors choose to focus on Myo5, while Myo3 depletion also affects DSB mobility? And why was overexpression of Myo5 in a Myo3 depleted background tested and not the other way around (Myo3 overexpression in a Myo5 depleted background)?
 - We have clarified this statement. “We integrated a set of plasmids that contain deletions of different domains of the MYO5 open reading frame [7]. These MYO5 domain deletion plasmids were integrated into a *myo5* deletion background to see which domains of Myo5 are required for DSB mobility.”
 - Myo5 and Myo3 have the same functional domains and are known to be functionally redundant in endocytosis [8].

- For the revision, we have built a plasmid with a copy of Myo3 with its endogenous promoter and inserted it into a *myo5* deletion strain. We found that the second plasmid copy of Myo3 was able to restore mobility of the DSBs when *myo5* was deleted (**Figure 4E-F**). Thus, two copies of *MYO3* in a *myo5* deletion behaves the same as two copies of *MYO5* in a *myo3* deletion (**Figure 4D, F**).
8. Fig.3B. There is no loading control for the western blot shown.
- For Figure 3B we ran a new western blot of the Las17-AID strain \pm IAA to show that adding IAA readily degrades Las17-AID 1 h within adding IAA. For the loading control, α -PGK1 was used as a probe.
9. Fig.4 Myo3 and Myo5 double depletion results in wild-type DSB movements, which are reduced upon loss of Las17 functionality. The authors conclude that ‘las17 can compensate for the loss of both Myo3 and Myo5.’. Has redundancy between Las17 and Myo3 and Myo5 been found in other contexts as well?
- We have revised this section, given that the *myo3 myo5* double inactivation appears to dramatically alter chromosome conformations. Although it is interesting that inactivating Las17 reduces mobility in the *myo3 myo5* double inactivation, we do not think we can assess how this occurs. Hence we have elected to remove this experiment.
10. Fig.S4A: different mutants are shown (SH3del) than what is written in legend and main text (Las17-WH2-CAdel).
- This figure has been updated with the correct MSD graph.
11. Fig 5, 6, and 7. Statistics are missing in Fig 5, 6 and 7.
- Text has been updated to say that the “Mean \pm SEM is shown” in the resection assays and morphology assays.
12. Figure 5A: Would be helpful to include an example image.
- We’ve added example images of the GFP-LacI *lacO::MAT* strain 0, 1, 2, 3, and 4 h after adding galactose.
13. P.9 line 182-184 – reference to Fig.S5A/S5B-D should be S6A/B-D.
- Text has been updated.
14. P.9 line 178: first time mention of VRP1 but not explained in text what it is until much later in the manuscript (discussion).
- In Figure 3A we now mention that Vrp1 mediates the interaction between Las17 and the type-I myosins.
15. Fig.S5A VRP1 mutant is only used once in the manuscript and the authors find a reduction in Ddc2 focus appearance. Have the authors tested the effect of VRP1 loss on DSB movement? Since VRP1 binds both Las17 and Myo3/5 (fig.3A), VPR1 loss might provide insights into the differential effects of Las17 loss and *myo3/myo5* depletions on DSB movement and DSB resection.
- As noted in Figure S5A, a deletion of *vrp1* impairs Ddc2-GFP focus formation following Gal-HO induction. A similar phenomenon is seen when both type-I myosins are depleted before Gal-HO induction. We believe that the *myo3 Δ Myo5-AID* strain is able to give us an idea of how losing Vrp1 either before or after DSB induction would affect DSB repair processes.
16. Figure 7B: Would be helpful to include images of the morphology of the yeast. This will clarify how the morphology was assessed.

- We have added example images of G1, small bud, and large bud cells.
17. Fig.7G -K: no legends present.
- Updated legends for Fig. 7.
18. Abstract: ‘...while overexpression of Exo1 suppressed resection inhibition by CK-666.’ From the data shown by the authors, they can only conclude Exo1 suppresses DSB movement inhibition by CK-666 (Fig.6H), but not resection inhibition. In fact, Exo1 overexpression does not rescue the resection defect imposed by CK-666 treatment (Fig.S5D).
- We made a Las17-AID pGal:Exo1 strain to test how Exo1 overexpression affects resection when Arp2/3 is inactivated. Using our qPCR resection assay (**Figure 5A**), we found that when Las17-AID was degraded before DSB induction (**Figure 6L**), overexpression of Exo1 was not able to initiate long range resection (**Figure 6J**). However, when Las17-AID was degraded 2 h after DSB induction (**Figure 6M**), overexpression of Exo1 was able to maintain wildtype levels of resection (**Figure 6J**) which could account for why CK-666 treatment did not affect DSB mobility when Exo was overexpressed.
19. Abstract: ‘These results suggest that Arp2/3 regulation plays an unanticipated role in the regulation of processing of DSB ends..’. A role for Arp2/3 is not necessarily unanticipated, based on earlier results on a role for the Arp2/3-nucleator Wasp/Las17 in repair in both mammalian cells and budding yeast (ref.11), as well as in DSB movement in flies and mammalian cells (ref.6, 7).
- The language in the abstract has been changed to reflect the importance of Arp2/3 in facilitating resection. Previous work showed that Arp2/3 had a role in single-stranded DNA (ssDNA) formation in mammalian cells [9] and that WASP helped load RPA onto ssDNA [10]. Our work adds to this by showing that Arp2/3 and Las17 are required for long range resection and that slowing down the rate of resection affects the mobility of DSBs.
20. In general the readability and flow of the text could be improved by explaining the rationale behind certain experiments. Below some examples:
- Line 162. The authors state “Since Las17 has a WH2 domain and acidic (CA) patch, it is possible that...”. Why would this be possible?
- We have revised this paragraph; see l. 136
- Line 111: Why do the authors add CK666 before and after DSB induction? (same question for Line 193 + 194)
- Turning off Arp2/3 activity either before or after DSB induction caused different effects on DSB repair processes. We have updated the text to include this clarification.
- Line 154: Could the authors speculate what the data above suggest? Is it logical that the SH3 domain is important? Is this domain involved in binding to arp2/3?
- We discuss the importance of the SH3 domain on Myo5 in the Discussion. The SH3 domain of Myo5 is required for Myo5 localization to pre-endocytic patches through Vrp1 and is therefore the SH3 domain is important for the NPF activity of Myo5 [7].
- Line 178-179: What’s the rationale for depleting VRP1 and Ku70?
- It is known that G1-arrested cells fail to initiate long-range resection and fail to activate the DNA damage checkpoint [11, 12]. However, when Ku70 or Ku80 are deleted resection is restored [13]. We therefore asked if Arp2/3 helps to remove Ku70/80 from the ends of DSBs to initiate resection

but found that Ddc2-GFP damage-dependent focus formation was still inhibited when Arp2/3 was inactivated with CK-666 in an *yku70* deletion mutant.

- Moreover, recruitment of Vrp1 to pre-endocytic patches is important for initiating endocytosis [14]. The primary role of Vrp1 in endocytosis is to bring the type-I myosins Myo3 and Myo5 to Las17; indeed a fusion protein of Las17 and Myo5 is able to rescue the defect in endocytosis found in *vrp1* deletion mutants [7]. Since Vrp1 has a role in Arp2/3 actin nucleation in endocytosis, we deleted *vrp1* in our Ddc2-GFP strain to see if it had an effect on DSB repair and we found that deleting *vrp1* disrupted damage-dependent Ddc2-GFP focus formation, as does inactivating Las17.

Line 204: Could the authors explain why Las17 is required for the initiation AND maintenance of resection.

- We do not yet know how the initiation and maintenance of resection are separately controlled. Previous work from several labs has shown that in G1-arrested cells, the barrier to Exo1 is the binding of Ku70/Ku80 at DSB ends; however we deleted *YKU70* and found that Ddc2-GFP focus formation was still inhibited when CK-666 was added to cells before DSB induction with Gal-HO (**Figure S6A-D**). Future research will investigate how Arp2/3 controls the initiation as well as the maintenance of 5' to 3' resection.

Line 251: "G2/M arrest was shortened to 4h", could the authors speculate why the G2/M arrest was shortened?

- We thought we had made this clear before but have expanded our explanation. In otherwise wildtype cells, inhibition of resection prevents activation of the Mec1-dependent G2/M arrest (i.e. when both Exo1 and Dna2 are inactivated or – here – when Arp2/3 is inhibited). Moreover, when Mec1 is deleted, there is usually no checkpoint. And no G2/M arrest. However, Usui and Petrini [15] found that some mutations in the MRX/Sae2 complex would trigger the activation of Mec1's homolog, Tel1 in the absence of Mec1. This "TM" (Tel1-MRX) checkpoint is short-lived compared to the normal Mec1-regulated cell cycle delay; cells resume cell cycle progression after about 4 h, compared to 10-12 h when Mec1 is activated. We conclude that inhibiting Arp2/3 by CK-666 or inactivating Las17 mimics the MRX/Sae2 defects, triggering the TM checkpoint. Indeed, as we had shown in the MS, deleting Tel1 or Mre11 eliminates this short arrest. These data place Arp2/3 at a step where MRX begins to initiate resection but does not yet show us exactly what is occurring. Note again that deleting Mre11 – like deleting Tel1 – abolishes this checkpoint, so it is a novel activation of this TM checkpoint when Mec1 is not activated.

Reviewer #2 (Remarks to the Author):

*One of the more confusing results in the manuscript is the finding that DSB mobility is normal in cells simultaneously depleted for Myo3 and Myo5. The authors suggest that Las17 can compensate for loss of both myosins, but it is unclear why Las17 would not compensate for loss of only one. Furthermore, the authors show that end resection is normal in *myo3* and *myo5* mutants, even though DSB mobility is*

compromised, whereas end resection is defective in the myo3 Myo5-AID strain where DSB mobility is restored. These findings raise the question of how (and if) end resection and DSB mobility are linked.

Please see the explanation provided to Reviewer 1. Recent data show that depleting both Myo3 and Myo5 disrupts many aspects of chromosome organization and most likely accounts for the apparent restoration of high mobility after inducing a break.

Main comments:

1. Figure 1: In Figure 1A, CK-666 added 2 h 40 min after HO induction reduced chromatin mobility. Was end resection by the qPCR-based method used under these conditions and was the percent cells with Dd2-GFP and Rad51-GFP foci determined? The data in Figure S5 show the impact of CK-666 added 20 min before HO induction. There should be considerable end resection 2 h 40 min after HO induction suggesting a role for Arp2/3 in mobility independent of end resection.

We used the drug CK-666 and the Las17-AID mutant to inhibit Arp2/3 activity. For measuring resection, we opted to use the Las17-AID mutant because using CK-666 in the resection assay would have been prohibitively expensive. In **Figure 5B** we degraded Las17-AID 2 h after HO induction. The resection assay showed that resection did not progress further when Las17-AID was degraded suggesting that *continuing* Arp2/3 activity is required to facilitate resection under normal circumstances.

This same question can be posed when we inactivate DNA2-AID after 160 min in an *exo1Δ* background. There is a lot of ssDNA at this point, as evidenced by the robust Ddc2-GFP foci. In and of itself ssDNA does not guarantee a high radius of confinement: We conclude that there needs to be continued resection. We thank the reviewer for this question and we have endeavored to make this important point more clear.

2. Figure 3: Does the WH2 deletion alone impact DSB mobility?

We found that deleting the WH2 domain of Las17 lowers the mobility of the DSB, as addressed to Reviewer 1.

3. Was chromatin mobility measured in myo3 and myo5 strains before HO induction using the lacO/GFP-LacI system? Maybe there is a lower baseline level of chromatin mobility in these mutants that can be increased by a DSB.

In the GFP-LacI lacO::MAT strain we deleted either Myo3 or Myo5 to measure how these type-I myosins affect baseline levels of chromatin mobility and we found that deletion of either type-I myosin did not affect baseline levels of chromatin mobility. (**Figure 4B-C**).

4. Interestingly, over-expression of Exo1 was able to overcome the inhibitory effect of CK-666 on DSB mobility when the inhibitor was added 2 h and 40 min after HO induction. However, end resection was not measured by the qPCR assay under these conditions. The Ddc2 foci data in Figure S5 are for CK-666 added

20 min before HO induction. If CK-666 is added before Gal induction can Gal:Exo1 restore DSB mobility? If Exo1 over-expression is not increasing end resection how might it impact DSB mobility?

As noted to Reviewer 1, we have now done qPCR measurements to confirm that overexpressing Exo1 does restore mobility if Arp2/3 is inactivated after the induction of a DSB, but not if inactivated before.

5. Can the HR defect of the Las17-AID strain be rescued by holding cells in nocodazole to maintain cell cycle arrest?

We thank the reviewer for this excellent suggestion. Indeed, holding cells in G2/M by nocodazole inhibition allowed a larger fraction of cells to complete DSB repair, confirming our conclusion that the decrease in gene conversion efficiency in the Las17-AID and Las17-AID *mre11Δ* mutants might be due to a shortened DNA damage checkpoint response.

6. Is mating-type switching, or another HR events that does not provoke the DNA damage checkpoint, normal when Las17 is depleted?

Mating-type switching (intrachromosomal and facilitated by a special donor preference RE sequence) happens very quickly and doesn't activate the Mec1/Rad53 checkpoint [16] [17]. We now show (**Figure 8F and Table S3**) that highly efficient *MAT* switching is unaffected by treating cells when CK-666 is added prior to HO induction (p value = 0.5). Here, long-range resection is not needed to accomplish *MAT* switching. Again we thank the reviewer for this question that helps also clarify when MRX is needed to allow DSB repair.

Minor comments:

1. Line 122: change Lad17 to Las17.

- Updated text.

2. Figure S2E: The Rc values of WT and *myo5* are indicated as not significantly different contradicting data shown in Figure 4C. Is this an error?

- The difference in the Rc between the WT and the *myo5* deletion strain is significantly different in Figure S2E, but we made an error when putting our data together. We analyzed 17 cells from the *myo5* deletion strain and showed that in the MSD curves (**Figure S2A-D**) but only reported the Rc from 15 of those cells analyzed. We've updated **Figure S2E** and **Table S2** to show that updated Rc numbers and the number of cells analyzed.
- We have gone through all other MSD and Rc graphs to ensure consistent reporting.

3. Figure 6D,E: Is there a reason for adding IAA 2 h after galactose induction instead of 20 min after inducing HO? Was resection measured by qPCR in these cells.

- We waited 2 h after adding galactose to degrade DNA2-AID to ensure efficient DSB formation through Gal-HO and allow resection to proceed. Gal-HO creates a DSB in ~90% of cells 1 h after

adding galactose [18] and we have previously shown that 2 h after DSB induction with Gal-HO that there is significant resection around the HO-cut site [19].

- Since we added IAA to our other AID-mutants 2 h after DSB induction, we did the same for our *exo1Δ* DNA2-AID strain to ensure consistency in conditions between mutants tested for DSB mobility.
- We did not measure resection by qPCR in the *exo1Δ* DNA2-AID ± IAA strain because it has previously been shown that deleting *exo1* and a component of the DNA2-Sgs1 complex prevents resection [20].

4. Figure 6H: Are the Rc values of WT + CK-666 and pGal:EXO1 + CK-666 significantly different?

- A one-way Anova test showed a significant difference between the WT + CK-666 and pGal:EXO1 + CK-666 (**Figure 6H, Table S1**). There was not a significant difference between the WT and pGal:EXO1 (**Figure 6H, Table S1**).

5. Figure 7: Labeling of the figure parts does not match the legend.

- Legend has been fixed to match the figure.

6. Figure 8B: Survival of the *mre11* mutant seems high.

- There was no significant difference between the survival rate of the wildtype and the *mre11Δ* mutant. We performed a one-way Anova test comparing the survival rate of the wildtype to the *myo3*, *myo5*, and *mre11* deletion mutants and found that there was no significant difference between the % of survivors in these strains (**Figure 8B and Table S3**).
- There is a frequent misunderstanding in the field that Mre11 is the “gatekeeper” that regulates whether there is resection or not. This is not the case. Mre11 in yeast is needed for NHEJ but is not essential for HR, because resection proceeds – albeit with a delay – by Exo1 and Dna2-Sgs1 and HR is therefore reasonably efficient.

7. Figure 8D, E: It would be helpful to have a schematic in a supplementary figure showing the location of recipient and donor sites in the genome.

- Figure 8D-E have been moved to a new supplemental figure (**Figure S9**) which includes a model of the repair assay (**Figure S9A**) and a map of where the *LEU2* donor is in the genome (**Figure S9B**).

Reviewer #3 (Remarks to the Author)

We have responded above.

Reviewer #4 (Remarks to the Author)

Overall, their results seem to reinforce the idea of an evolutionarily conserved role for the Arp2/3 complex, rather than introducing a completely novel function. Therefore, the manuscript may be better suited for a different journal.

We beg to differ with the reviewer on several grounds. We agree that we have shown clearly that Arp2/3 plays a key role in DSB-induced chromosome mobility in yeast, as in mammals and flies. Also, in mammalian cells there was an effect on RPA focus formation that suggested a role in 5' to 3' resection. Here we have documented the effect on resection in detail and shown that Arp2/3 is necessary both to initiate long-range resection and to maintain it (and that these two phases are distinct). Moreover, we show that continuing resection is essential for the increased radius of confinement and that just having a long ssDNA tail is not sufficient. We go further to show that mobility is tied to *continuing* resection by (a) inactivating the long-range resection machinery itself and (b) overexpressing Exo1 in a Las17-inactivated state. In addition, we show that higher mobility per se is not the key factor in facilitating ectopic recombination; it is instead the activation of the DNA damage checkpoint to prevent cells from dividing before repair is complete. We show this by improving repair by holding Arp2/3-inactivated cells in nocodazole for 6 h. Also we show that inactivating Arp2/3 by CK-666 or inactivating Las17 triggers and “TM” checkpoint that has only been shown to be induced by mutations in Rad50 and Sae2.

Major points:

1. Reintroduction of a second copy of Myo5 into the Myo3del mutant appears to rescue the DSB mobility (Fig. 4). Does this also work the other way round, meaning reintroduction of a second Myo3 copy into the Myo5del strain? Also, either specifically deleting the type I myosins in the yeast nucleus or reintroduction of a nuclear targeted version of Myo3/Myo5 in a deletion strain would strengthen the results, as it would highlight the nuclear-specific role for the studied processes.

We have now shown that two copies of Myo3 compensate for the deletion of Myo5.

2. How do the authors explain the unexpected result that deletion of both Myo3 and Myo5 can restore the DSB mobility (as opposed to single deletion)? Why is Las17 only able to rescue the phenotype of the double deletion and not the single deletions?

As noted above, a recent pre-print showed that type-I myosins have a role in nuclear organization [21]. It should be noted that Myo3 was depleted in a *myo5Δ* background and Myo5 was depleted in a *myo3Δ* background, so either Myo3 or Myo5 alone was sufficient for maintaining genome organization. Depletion of both type-I myosins from the nucleus significantly affected cell growth, caused changes in genome organization, and changed the shape of the nuclear envelope. The dramatic changes in chromosome arrangement and nuclear properties most likely explain our unexpected finding that the double mutant had a higher radius of confinement.

Although we found that inactivating Las17 decrease the unexpected high radius of confinement in the *myo3Δ myo5Δ* double depletion, we have chosen to remove this experiment, given that the organization of the nucleus in the *myo3 myo5* double depletion is aberrant. We confirmed this exceptional behavior by measuring a change in the MSD of the Spindle Pole Bodies relative to the cell's bud neck.

3. In line 330 of the discussion, the authors conclude that “a defect in long-range end-resection is sufficient to impair damage-induced chromatin mobility.” However, the single deletion of either *Myo3* or *Myo5* did not greatly affect resection, yet it still led to reduced DSB mobility, as the authors correctly acknowledge later in lines 384-386. Therefore, this claim may be somewhat overstated at this point.

We have addressed this point for Reviewer 1. Slowing down the rate of resection with the *fun30* deletion mutant or stopping long-range resection by getting rid of Exo1 and Dna2 lowered the mobility of the DSBs. Increasing resection by overexpressing Exo1 was able to compensate for the loss of Arp2/3 activity further strengthening the link between the rate of resection and DSB mobility. Single deletions of *Myo3* and *Myo5* did not affect the rate of resection, nor did they impair the efficiency of repair.

The rate of resection (4 kb/h) is one factor that contributes to DSB mobility. In budding yeast, it has been shown that activation of the DNA damage checkpoint through Ddc1/Ddc2 targeting to a chromatin locus [22] was sufficient to increase chromatin mobility [23]. In this instance there was no resection of DNA ends, but increased Mec1 activity and phosphorylation of Rad9 and Rad53 coincided with increased chromatin mobility [23]. It is unclear why deleting *myo3* or *myo5* lowers DSB mobility when resection is unperturbed (**Figure 5D**) and Rad53 is still readily phosphorylated (**Figure 7E**).

Minor points:

1. The concentration of 10 μ M CK666 used for their assays appears to be rather low. It would be helpful to show, that Arp2/3 function is indeed inhibited at this concentration. Also, a good control for CK666 treatment would be to include the inactive CK-689 compound. Generally though, it would be really important to achieve nuclear-specific inhibition of Arp2/3, like nuclear uncaging of the inhibitor or depletion of Arp2/3 and reintroduction of a cytoplasmic derivative. The field heavily suffers from the global use of CK666 and its over interpretation for nuclear effects. How can one exclude that all these effects arise from cytoplasmic Arp2/3 inhibition?
 - The concentration of CK-666 was actually 100 μ M and was incorrectly reported. We apologize for the mistake.
 - In budding yeast the primary role of Arp2/3 in the cytoplasm is in endocytosis. Our *sla2* deletion mutant, which is defective in cell membrane invagination [14], did not exhibit any defect in DSB mobility, suggesting that if we were able to specifically inhibit Arp2/3's primary role in endocytosis in the cytoplasm while maintaining its role in the nucleus, it would not affect mobility of the DSB.
2. There should be a space between the numerical value and the unit symbol for concentrations (e.g. line 92 “10 μ M”, line 122 “1 mM, etc.).
 - Text has been updated.
3. Lines 98-102: The rationale for the experiments shown in Fig 1 D and E is not very clear as endocytosis takes place at the plasma membrane whereas DSB processing is happening in the nucleus. How would these processes be connected? Also, the authors state that “deletion of SLA2, a key component of endocytosis, did not affect the behavior of the DSB (Figure 1D-E).” However, to be more precise, I suggest rephrasing it to “did not significantly affect...” since there appears to be a slight decrease in MSD.

- We state that Arp2/3 is primarily studied for its role in endocytosis, and we asked whether Arp2/3 dependent DSB mobility was linked to the role of Arp2/3 in endocytosis. To answer this question we targeted another protein, Sla2, which is important for forming pre-endocytic patches and measured DSB mobility in this deletion mutant.
 - We have updated the text to note that there is a statistically insignificant decrease in the Rc of the *sla2* deletion mutant (**Figure 1A and Table S1**).
4. Lines 111 and 113: Wrong Figure callout; should be Fig. 2B instead of 1B.
 - Text has been updated.
 5. Line 120: Figure 3A should also be referenced in the text where Vrp1 and Myo3/Myo5 and its domains are discussed.
 - Text has been updated to reference Figure 3A when referring to Vrp1 and the functional domains of Myo5.
 6. Line 178: The abbreviation and function of Vrp1 should be explained the first time its mentioned in the manuscript. Also, it might be helpful to reference Figure 3A at this point.
 - Text has been updated to show the full name and primary function of verprolin (Vrp1).
 7. Lines 193-194: "Resection in cells treated with CK-666 after 1 h after DSB induction was also significantly impaired." While the data suggest this, no statistical test has been performed to support the claim. The statement should either be rephrased or a statistical test should be conducted to confirm the significance of the observed differences.
 - Text has been rephrased to add clarity.
 8. Line 252: A reference for Figure 7E is wrongly included at this position of the text or wrongly labeled (*myo5del* instead of CK666 treatment).
 - Figure 7E is a western blot probing for Rad53 and Pgc1 in a *myo5Δ* strain. We did not run a western blot of CK-666 treated cells due to the amount of drug that we would need to run this experiment. Instead, we used the Las17-AID mutant ± IAA to test how disrupting Arp2/3 activity before DSB induction affected activation of the DNA damage checkpoint.
 9. Line 285: A series of strains that had been previously characterized with repair efficiencies ranging from 46% to 9% is mentioned. However, further explanation or description of the individual strains would be helpful for the reader at this point.
 - We added a description of the strains used as well as a map of where the HO-cut site is and where the various *LEU2* donors are in supplemental **Figure S8**.
 10. Line 288: "...Las17-AID + IAA showed a significant decrease in the percentage of survivors in all 4 strains." There are no statistical tests performed for these data nor for any other data presented in Fig. 8. The sentence should either be rephrased or significance tested.
 - This figure has been moved to **Figure S9** and divided into 4 graphs based on the background strain. A one-way Anova was performed to show that the Las17-AID + IAA is significantly different than Las17-AID not treated with IAA and the WT (**Figure S9G-J**).
 11. Line 294: Please introduce the abbreviation for break-induced replication (BIR) here.
 - Text has been updated.

12. The authors should specify the statistical tests performed for each analysis directly in each figure legend, rather than referencing previous figure legends. (e.g. in Fig legend 1B and E). Also, the number of independent biological replicates should be clearly stated in the figure legends for transparency.
 - For MSD curves and their accompanying Rc graphs the number of biological replicates is included after the strain name in the figure legend e.g. Figure 1A WT (n=14) and WT with CK-666 (n=34).
 - The statistical tests used in the Rc graphs have been included in the figure legends and are also listed in Table S1 along with the P value and significance.
13. The authors refer to their quantification in Figures 1-4, 6 and 8 as “boxplots” even though the data are presented as bar graphs. The labelling needs to be corrected, or preferably, boxplots should be displayed instead of bar graphs.
 - Legends have been updated.
14. Fig 2D: There is a mislabeling in the figure legend. Panel D shows data for the treatment group “cut + CK666 before gal, 3h”, while the legend incorrectly states “cut + CK-666 after gal, 3h”. It appears that the labeling (or figure panels) between D and F may have been mixed up and should be corrected for clarity.
 - Text has been updated so the legends match their appropriate graph.
15. Fig. 2: It appears that the same data for the treatment groups “cut, 3h” and “uncut, 0h” have been used in panels B, D and F, as the trend of the graphs are exactly the same. If the authors prefer to show them in different graphs due to better clarity of the data, they should be more transparent about this and clearly state it in the figure legend.
 - The “uncut, 0h” and “cut, 3h” are our wildtype data, confirming what others have reported that following a DSB there is an increase in chromatin mobility near a DSB site [24, 25]. We note that these are the same data.
 - These data are shown in Figures 2B, 2D, and 2F to compare how CK-666 affected baseline levels of chromatin mobility and chromatin mobility after DSB induction with Gal-HO where CK-666 was added either before or after galactose.
16. Fig 2G: The authors reference (B) for the statistical test used, however, there is no test performed for Fig. 2B. I assume they reference Fig. 1B. Also, instead of a t-test an ANOVA should be performed as more than 2 groups are being compared.
 - The legend now includes which statistical test was used to analyze significance in Rc value and percentage of survivors. This information is also included in **Tables S1 and S3**.
17. Fig. 3D, F: Statistical test should be mentioned in the figure legend and ANOVA should be used instead for multiple comparison of data.
 - The statistical tests used for each comparison is now listed in each figure legend and in Table S1 and S3 along with the P value and which correctional test was used.
18. Fig. 5 C-E: Legend missing for y-axis labeling. What exactly is “Fresected”?
 - Fresected is the fraction of resected DNA and is the unit used to measure how much resected DNA is measured through the restriction enzyme based qPCR assay we used as illustrated in **Figure 5A** and described in Gnugge et al. *Processing the DNA Double-Strand Breaks in Yeast Methods Enzymology* [26].
19. Fig 7: Panel labeling does not match the figure legends.

- Updated legend.
20. Fig. 8C is not referenced in the text.
- Figure 8C has been moved to Figure S8A and is referenced in the text along with Figure S8B to describe the series of strains with the donor for homologous recombination moved to different locations in the genome.

References

1. Hetrick, B., et al., *Small molecules CK-666 and CK-869 inhibit actin-related protein 2/3 complex by blocking an activating conformational change*. Chem Biol, 2013. **20**(5): p. 701-12.
2. Guo, S., et al., *Dynamic remodeling of actin networks by cyclase-associated protein and CAP-Abp1 complexes*. Curr Biol, 2023. **33**(20): p. 4484-4495 e5.
3. Robertson, A.S., et al., *The WASP homologue Las17 activates the novel actin-regulatory activity of Ysc84 to promote endocytosis in yeast*. Mol Biol Cell, 2009. **20**(6): p. 1618-28.
4. Evangelista, M., et al., *A role for myosin-I in actin assembly through interactions with Vrp1p, Bee1p, and the Arp2/3 complex*. J Cell Biol, 2000. **148**(2): p. 353-62.
5. Galletta, B.J., D.Y. Chuang, and J.A. Cooper, *Distinct roles for Arp2/3 regulators in actin assembly and endocytosis*. PLoS Biol, 2008. **6**(1): p. e1.
6. Waterman, D.P., et al., *Live cell monitoring of double strand breaks in S. cerevisiae*. PLoS Genet, 2019. **15**(3): p. e1008001.
7. Lewellyn, E.B., et al., *An Engineered Minimal WASP-Myosin Fusion Protein Reveals Essential Functions for Endocytosis*. Dev Cell, 2015. **35**(3): p. 281-94.
8. Goodson, H.V., et al., *Synthetic lethality screen identifies a novel yeast myosin I gene (MYO5): myosin I proteins are required for polarization of the actin cytoskeleton*. J Cell Biol, 1996. **133**(6): p. 1277-91.
9. Schrank, B.R., et al., *Nuclear ARP2/3 drives DNA break clustering for homology-directed repair*. Nature, 2018. **559**(7712): p. 61-66.
10. Han, S.S., et al., *WAsp modulates RPA function on single-stranded DNA in response to replication stress and DNA damage*. Nat Commun, 2022. **13**(1): p. 3743.
11. Mimitou, E.P. and L.S. Symington, *Ku prevents Exo1 and Sgs1-dependent resection of DNA ends in the absence of a functional MRX complex or Sae2*. EMBO J, 2010. **29**(19): p. 3358-69.
12. Mimitou, E.P. and L.S. Symington, *DNA end resection: many nucleases make light work*. DNA Repair (Amst), 2009. **8**(9): p. 983-95.
13. Li, K., et al., *Yeast ATM and ATR kinases use different mechanisms to spread histone H2A phosphorylation around a DNA double-strand break*. Proc Natl Acad Sci U S A, 2020. **117**(35): p. 21354-21363.
14. Idrissi, F.Z., et al., *Ultrastructural dynamics of proteins involved in endocytic budding*. Proc Natl Acad Sci U S A, 2012. **109**(39): p. E2587-94.
15. Usui, T., H. Ogawa, and J.H. Petrini, *A DNA damage response pathway controlled by Tel1 and the Mre11 complex*. Mol Cell, 2001. **7**(6): p. 1255-66.
16. White, C.I. and J.E. Haber, *Intermediates of recombination during mating type switching in Saccharomyces cerevisiae*. EMBO J, 1990. **9**(3): p. 663-73.
17. Yamaguchi, M. and J.E. Haber, *Monitoring Gene Conversion in Budding Yeast by Southern Blot Analysis*. Methods Mol Biol, 2021. **2153**: p. 221-238.

18. Lee, S.E., et al., *Saccharomyces Ku70, mre11/rad50 and RPA proteins regulate adaptation to G2/M arrest after DNA damage*. Cell, 1998. **94**(3): p. 399-409.
19. Eapen, V.V., et al., *The Saccharomyces cerevisiae chromatin remodeler Fun30 regulates DNA end resection and checkpoint deactivation*. Mol Cell Biol, 2012. **32**(22): p. 4727-40.
20. Zhu, Z., et al., *Sgs1 helicase and two nucleases Dna2 and Exo1 resect DNA double-strand break ends*. Cell, 2008. **134**(6): p. 981-94.
21. Peng, A.Y.T., J. Li, and B.C. Freeman, *Nuclear Type I Myosins are Essential for Life and Genome Organization*. bioRxiv, 2024.
22. Bonilla, C.Y., J.A. Melo, and D.P. Toczyski, *Colocalization of sensors is sufficient to activate the DNA damage checkpoint in the absence of damage*. Mol Cell, 2008. **30**(3): p. 267-76.
23. Seeber, A., V. Dion, and S.M. Gasser, *Checkpoint kinases and the INO80 nucleosome remodeling complex enhance global chromatin mobility in response to DNA damage*. Genes Dev, 2013. **27**(18): p. 1999-2008.
24. Dion, V., et al., *Increased mobility of double-strand breaks requires Mec1, Rad9 and the homologous recombination machinery*. Nat Cell Biol, 2012. **14**(5): p. 502-9.
25. Mine-Hattab, J. and R. Rothstein, *Increased chromosome mobility facilitates homology search during recombination*. Nat Cell Biol, 2012. **14**(5): p. 510-7.
26. Gnugge, R., J. Oh, and L.S. Symington, *Processing of DNA Double-Strand Breaks in Yeast*. Methods Enzymol, 2018. **600**: p. 1-24.

Response to reviewers.

The Thread of reviews and responses is below. The points that required our further comment are now in blue font.

Reviewer #1 (Remarks to the Author)

We appreciate the significant effort the authors have made to address our concerns, and we acknowledge that the manuscript has improved considerably in response to our previous feedback. The authors have successfully resolved several of the key issues initially raised. While we still find the data to be primarily descriptive and feel that the study does not deliver a conclusive mechanistic model for how the Arp2/3 complex facilitates DSB movement and repair, the new findings do offer valuable insights. In particular, the observation that checkpoint activation defects, rather than impaired DSB mobility per se, may be more central to the repair deficiency in the absence of Arp2/3 activity is intriguing. The text, however, remains quite technical and may be challenging for readers outside the immediate field. Additionally, some references to figures in the main text are incorrect. We would encourage the authors to verify these references and to consider input from a colleague less familiar with the field to improve clarity and accessibility of the text.

We have a few comments left on this revised manuscript.

1) We are not fully satisfied with one answer to one of our previous comments (=Fig.6H: Exo1 overexpression.... + CK-666' condition?) in which we asked to better determine the effect of Exo1 overexpression on Arp2/3 inhibition with CK-666. The authors now performed resection analyses with overexpressing exo1 in the LAS17-AID + IAA mutant and they find a rescue (Fig.6I). The authors therefore now conclude in line 383: 'We also found that EXO1 overexpression suppressed the reduction in Rc caused by adding CK-666 by maintaining wildtype levels of resection.'

However, we are still of the opinion that the rescue of Exo1 overexpression on DSB movement and resection in the presence of CK-666 remains rather inconclusive. We feel there are two separate pieces of data: On the one hand, Exo1 overexpression rescues the CK-666-induced movement defect (Fig.6G), but not the defect in end-resection (reduced Ddc1-GFP foci formation, Fig. S5D). On the other hand, the authors find that Exo1 overexpression rescues resection in the Las17-AID + IAA background (Fig.6I), but movement was not tested in this background as far as we can tell.

Therefore, while the authors did test the effect of Exo1 overexpression on DSB movement upon CK-666 treatment, they did not test this effect in the LAS17-AID + IAA background (where they find a rescue of the resection defect). We would therefore suggest to either test the DSB movement effect of Exo1 overexpression in the Las17-AID + IAA background or tone down their conclusion that EXO1 overexpression suppressed the reduction in Rc caused by CK-666 by maintaining wildtype levels of resection.'

We thank the reviewer for this suggestion. Previously we used CK-666 treatment and Las17-AID + IAA interchangeably to test the effects of inhibiting Arp2/3 activity. To complement our resection data from the Las17-AID strain with *exo1* overexpression (pGal:*EXO1* Las17-AID), we measured DSB mobility in a Ddc2-GFP strain where IAA was added 2 h after DSB induction and DSB mobility was measured 3 h after DSB induction as previously described. We found that there was no significant difference in DSB mobility when *exo1* was overexpressed in the Las17-AID background (**Figure 7E-F**), which is consistent with the MSD data from pGal:*EXO1* (**Figure 7B-C**). Degradation of Las17-AID 2 h after DSB induction when *exo1* was overexpressed did not affect DSB mobility or the radius of confinement of the DSB (**Figure 7E-F**).

2) Although the authors now included that the assays shown in Fig.5-7 are averages + SEM, they did not include statistical tests for the line graphs in figure 5B, 5C, 5D, 6A, 6H, 6I, 7B, 7D, 7F, 7H, 7J, 8C, 8H.

We compared the amount of resected DNA at the 6 h timepoint between the wildtype and the mutants to see if there was a significant change in resection. We used one-way Anova tests with Dunnetts test to correct for multiple comparisons or t-tests. This data is included in Table S3.

3) Line 140: Please include what the abbreviation WH2 stands for.

Text has been updated.

4) Line 403-404: “We show that the highly conserved C-terminal region across WASP orthologs (WH2 and CA domains) of Las17 is required for DSB mobility”. The authors show that only the WH2 domain is required for DSB mobility, not the CA domain and would have to revise this conclusion.

Text has been updated.

Reviewer #2 (Remarks to the Author)

The authors have adequately addressed my concerns. One minor comment, the *vpr1* data are presented in Fig S5A not Fig 3A (line 198).

Text has been updated to refer to the correct figure. Fig. 3A is a model that shows the interactions between Las17, Vrp1, Myo3, and Myo5.

Reviewer #3 (Remarks to the Author)

Reviewer #5 (Remarks to the Author)

> I was asked to step in for the previous Reviewer #4 to assess the authors' responses to the reviewers' comments. I will therefore go through the comments point by point:<

Overall, their results seem to reinforce the idea of an evolutionarily conserved role for the Arp2/3 complex, rather than introducing a completely novel function. Therefore, the manuscript may be better suited for a different journal.

We beg to differ with the reviewer on several grounds. We agree that we have shown clearly that Arp2/3 plays a key role in DSB-induced chromosome mobility in yeast, as in mammals and flies. Also, in mammalian cells there was an effect on RPA focus formation that suggested a role in 5' to 3' resection. Here we have documented the effect on resection in detail and shown that Arp2/3 is necessary both to initiate long-range resection and to maintain it (and that these two phases are distinct). Moreover, we show that continuing resection is essential for the increased radius of confinement and that just having a long ssDNA tail is not sufficient. We go further to show that mobility is tied to continuing resection by (a) inactivating the long-range resection machinery itself and (b) overexpressing Exo1 in a Las17-inactivated state. In addition, we show that higher mobility per se is not the key factor in facilitating ectopic recombination; it is instead the activation of the DNA damage checkpoint to prevent cells from dividing before repair is complete. We show this by improving repair by holding Arp2/3-inactivated cells in nocodazole for 6 h. Also we show that inactivating Arp2/3 by CK-666 or inactivating Las17 triggers and "TM" checkpoint that has only been shown to be induced by mutations in Rad50 and Sae2.

> I do agree with the authors that their study provides more value than the demonstration of an evolutionary conserved role of the ARP2/3 complex in DSB repair.

Zhou and colleagues use beautiful yeast genetics and imaging analyses to address the relationship between actin branching, DNA end resection and chromatin mobility upon DSB induction in yeast. They provide novel insight into the role of ARP2/3's activity for the processing of DNA ends, checkpoint activation and DSB repair. The data are of high technical quality and well presented with a good structure making the manuscript easy to read. Their findings pave the way for future studies where the model organism *Saccharomyces cerevisiae* can be utilized to investigate basic molecular mechanisms of actin cytoskeletal functions in genome stability. I consider the manuscript well suited for publication in Nature Communications.<

Major points:

1. Reintroduction of a second copy of Myo5 into the Myo3del mutant appears to rescue the DSB mobility (Fig. 4). Does this also work the other way round, meaning reintroduction of a second Myo3 copy into the Myo5del strain? Also, either specifically deleting the type I myosins in the yeast nucleus or reintroduction of a nuclear targeted version of Myo3/Myo5 in a deletion strain would strengthen the results, as it would highlight the nuclear-specific role for the studied processes.

We have now shown that two copies of Myo3 compensate for the deletion of Myo5.

> While the authors replied to the first part of the question, they missed addressing the second part. The previous reviewer #4 asked to study if the observed phenotype for type I myosins in DSB mobility is specific for their nuclear pool by either (1) deleting the nuclear Myo3/5 pool or (2) reconstituting the

respective deletions with a nuclear targeted variant. I would like to ask the authors to address this point.<

Deleting the nuclear pool of Myo3/Myo5 is possible using the “anchors-away” method developed by Brian Freeman’s lab at the University of Illinois Urbana-Champaign [1]. However, in order to adapt this system for use in our strains, it would require extensive gene editing and troubleshooting which makes it prohibitively difficult for us to answer this question.

In terms of rescuing DSB mobility by making a nuclear targeting version of Myo3/Myo5 it would be difficult to control the amount of the NLS-tagged Myo3/Myo5 that is recruited to the nucleus. Since we have shown that the number of copies of Myo3/Myo5 affects DSB mobility we would need a way to verify that the NLS-tagged Myo3/Myo5 is present at wildtype levels of Myo3/Myo5 in the nucleus in the deletion mutant of *myo3* or *myo5* (Figure 4D-F and Table S1).

2. How do the authors explain the unexpected result that deletion of both *Myo3* and *Myo5* can restore the DSB mobility (as opposed to single deletion)? Why is *Las17* only able to rescue the phenotype of the double deletion and not the single deletions?

*As noted above, a recent pre-print showed that type-I myosins have a role in nuclear organization [21]. It should be noted that *Myo3* was depleted in a *myo5Δ* background and *Myo5* was depleted in a *myo3Δ* background, so either *Myo3* or *Myo5* alone was sufficient for maintaining genome organization. Depletion of both type-I myosins from the nucleus significantly affected cell growth, caused changes in genome organization, and changed the shape of the nuclear envelope. The dramatic changes in chromosome arrangement and nuclear properties most likely explain our unexpected finding that the double mutant had a higher radius of confinement.*

*Although we found that inactivating *Las17* decrease the unexpected high radius of confinement in the *myo3Δ myo5Δ* double depletion, we have chosen to remove this experiment, given that the organization of the nucleus in the *myo3 myo5* double depletion is aberrant. We confirmed this exceptional behavior by measuring a change in the MSD of the Spindle Pole Bodies relative to the cell’s bud neck.*

> This issue was also raised by the other reviewers and both comments were properly addressed.<

3. In line 330 of the discussion, the authors conclude that “a defect in long-range end-resection is sufficient to impair damage-induced chromatin mobility.” However, the single deletion of either *Myo3* or *Myo5* did not greatly affect resection, yet it still led to reduced DSB mobility, as the authors correctly acknowledge later in lines 384-386. Therefore, this claim may be somewhat overstated at this point. We have addressed this point for Reviewer 1. Slowing down the rate of resection with the *fun30* deletion mutant or stopping long-range resection by getting rid of *Exo1* and *Dna2* lowered the mobility of the DSBs. Increasing resection by overexpressing *Exo1* was able to compensate for the loss of *Arp2/3* activity further strengthening the link between the rate of resection and DSB mobility. Single deletions of *Myo3* and *Myo5* did not affect the rate of resection, nor did they impair the efficiency of repair.

*The rate of resection (4 kb/h) is one factor that contributes to DSB mobility. In budding yeast, it has been shown that activation of the DNA damage checkpoint through *Ddc1/Ddc2* targeting to a chromatin locus [22] was sufficient to increase chromatin mobility [23]. In this instance there was no resection of DNA ends, but increased *Mec1* activity and phosphorylation of *Rad9* and *Rad53* coincided with increased*

chromatin mobility [23]. It is unclear why deleting myo3 or myo5 lowers DSB mobility when resection is unperturbed (Figure 5D) and Rad53 is still readily phosphorylated (Figure 7E).

> This point was also raised by the other reviewers and was properly addressed by the authors.<

Minor points:

1. The concentration of 10 μ M CK666 used for their assays appears to be rather low. It would be helpful to show, that Arp2/3 function is indeed inhibited at this concentration. Also, a good control for CK666 treatment would be to include the inactive CK-689 compound. Generally though, it would be really important to achieve nuclear-specific inhibition of Arp2/3, like nuclear uncaring of the inhibitor or depletion of Arp2/3 and reintroduction of a cytoplasmic derivative. The field heavily suffers from the global use of CK666 and its over interpretation for nuclear effects. How can one exclude that all these effects arise from cytoplasmic Arp2/3 inhibition?

The concentration of CK-666 was actually 100 μ M and was incorrectly reported. We apologize for the mistake.

In budding yeast the primary role of Arp2/3 in the cytoplasm is in endocytosis. Our sla2 deletion mutant, which is defective in cell membrane invagination [14], did not exhibit any defect in DSB mobility, suggesting that if we were able to specifically inhibit Arp2/3's primary role in endocytosis in the cytoplasm while maintaining its role in the nucleus, it would not affect mobility of the DSB.

> The CK666 concentration was corrected by the authors. However, I do not see that the authors have used the inactive CK-689 compound as negative control in one of their assays. Therefore, I would like to ask the authors to show this control as it was requested by the previous reviewer. Regarding the second point of a nuclear-specific inhibition of Arp2/3, I understand the reviewer's concern and the current "sloppiness" in the field in terms of claiming nucleus-specific effects upon usage of a compartment unspecific drug. However, the authors do not claim compartment specific inhibition or localization of the Arp2/3 complex throughout their entire manuscript and provide the Sla2 endocytosis control, which covers the major function of cytoplasmic Arp2/3. Secondly, they were asked to provide a nucleus-specific function of the type I myosins (see above) as a major point raised by the previous reviewer which will also address this question. In that context and with the notion that the previous reviewer #4 had listed this as a minor point, I do not insist on a further experimental revision here.<

CK-689 is an inactive form of CK-666 that binds to Arp2/3 without inhibiting Arp2/3 activity [2, 3]. We show that in **Figure 1C-D** that CK-689 does not affect the mobility of the DSB.

2. There should be a space between the numerical value and the unit symbol for concentrations (e.g. line 92 "10 μ M", line 122 "1 mM, etc.).

Text has been updated.

>Fine<

3. Lines 98-102: The rationale for the experiments shown in Fig 1 D and E is not very clear as endocytosis takes place at the plasma membrane whereas DSB processing is happening in the nucleus. How would these processes be connected? Also, the authors state that "deletion of SLA2, a key component of

endocytosis, did not affect the behavior of the DSB (Figure 1D-E).” However, to be more precise, I suggest rephrasing it to “did not significantly affect...” since there appears to be a slight decrease in MSD. We state that Arp2/3 is primarily studied for its role in endocytosis, and we asked whether Arp2/3 dependent DSB mobility was linked to the role of Arp2/3 in endocytosis. To answer this question we targeted another protein, Sla2, which is important for forming pre-endocytic patches and measured DSB mobility in this deletion mutant.

We have updated the text to note that there is a statistically insignificant decrease in the Rc of the sla2 deletion mutant (Figure 1A and Table S1).

>Fine<

4. Lines 111 and 113: Wrong Figure callout; should be Fig. 2B instead of 1B.

Text has been updated.

>Fine<

5. Line 120: Figure 3A should also be referenced in the text where Vrp1 and Myo3/Myo5 and its domains are discussed.

Text has been updated to reference Figure 3A when referring to Vrp1 and the functional domains of Myo5.

>Fine<

6. Line 178: The abbreviation and function of Vrp1 should be explained the first time its mentioned in the manuscript. Also, it might be helpful to reference Figure 3A at this point.

Text has been updated to show the full name and primary function of verprolin (Vrp1).

>Fine<

7. Lines 193-194: “Resection in cells treated with CK-666 after 1 h after DSB induction was also significantly impaired.” While the data suggest this, no statistical test has been performed to support the claim. The statement should either be rephrased or a statistical test should be conducted to confirm the significance of the observed differences.

Text has been rephrased to add clarity.

> Fine – could not find this part, most likely re-written.<

8. Line 252: A reference for Figure 7E is wrongly included at this position of the text or wrongly labeled (myo5del instead of CK666 treatment).

Figure 7E is a western blot probing for Rad53 and Pgc1 in a myo5Δ strain. We did not run a western blot of CK-666 treated cells due to the amount of drug that we would need to run this experiment. Instead, we used the Las17-AID mutant ± IAA to test how disrupting Arp2/3 activity before DSB induction affected activation of the DNA damage checkpoint.

>Fine<

9. Line 285: A series of strains that had been previously characterized with repair efficiencies ranging from 46% to 9% is mentioned. However, further explanation or description of the individual strains would be helpful for the reader at this point.

We added a description of the strains used as well as a map of where the HO-cut site is and where the various LEU2 donors are in supplemental Figure S8.

>Fine<

10. Line 288: "...Las17-AID + IAA showed a significant decrease in the percentage of survivors in all 4 strains." There are no statistical tests performed for these data nor for any other data presented in Fig. 8. The sentence should either be rephrased or significance tested.

This figure has been moved to Figure S9 and divided into 4 graphs based on the background strain. A one-way Anova was performed to show that the Las17-AID + IAA is significantly different than Las17-AID not treated with IAA and the WT (Figure S9G-J).

>Fine<

11. Line 294: Please introduce the abbreviation for break-induced replication (BIR) here.
Text has been updated.

>Fine<

12. The authors should specify the statistical tests performed for each analysis directly in each figure legend, rather than referencing previous figure legends. (e.g. in Fig legend 1B and E). Also, the number of independent biological replicates should be clearly stated in the figure legends for transparency.

For MSD curves and their accompanying Rc graphs the number of biological replicates is included after the strain name in the figure legend e.g. Figure 1A WT (n=14) and WT with CK-666 (n=34).

The statistical tests used in the Rc graphs have been included in the figure legends and are also listed in Table S1 along with the P value and significance.

>Fine<

13. The authors refer to their quantification in Figures 1-4, 6 and 8 as "boxplots" even though the data are presented as bar graphs. The labelling needs to be corrected, or preferably, boxplots should be displayed instead of bar graphs.

Legends have been updated.

>Fine<

14. Fig 2D: There is a mislabeling in the figure legend. Panel D shows data for the treatment group "cut + CK666 before gal, 3h", while the legend incorrectly states "cut + CK-666 after gal, 3h". It appears that

the labeling (or figure panels) between D and F may have been mixed up and should be corrected for clarity.

Text has been updated so the legends match their appropriate graph.

>Fine<

15. Fig. 2: It appears that the same data for the treatment groups “cut, 3h” and “uncut, 0h” have been used in panels B, D and F, as the trend of the graphs are exactly the same. If the authors prefer to show them in different graphs due to better clarity of the data, they should be more transparent about this and clearly state it in the figure legend.

The “uncut, 0h” and “cut, 3h” are our wildtype data, confirming what others have reported that following a DSB there is an increase in chromatin mobility near a DSB site [24, 25]. We note that these are the same data.

These data are shown in Figures 2B, 2D, and 2F to compare how CK-666 affected baseline levels of chromatin mobility and chromatin mobility after DSB induction with Gal-HO where CK-666 was added either before or after galactose.

> Although the authors explain the usage of the same data for the “uncut, 0h” and “cut, 3h” for all three panels in their response to the reviewer #4, this statement is still missing from the figure legend of Figure 2. Please add.<

Legend has been updated with this information to improve clarity.

16. Fig 2G: The authors reference (B) for the statistical test used, however, there is no test performed for Fig. 2B. I assume they reference Fig. 1B. Also, instead of a t-test an ANOVA should be performed as more than 2 groups are being compared.

The legend now includes which statistical test was used to analyze significance in Rc value and percentage of survivors. This information is also included in Tables S1 and S3.

>Fine<

17. Fig. 3D, F: Statistical test should be mentioned in the figure legend and ANOVA should be used instead for multiple comparison of data.

The statistical tests used for each comparison is now listed in each figure legend and in Table S1 and S3 along with the P value and which correctional test was used.

>Fine<

18. Fig. 5 C-E: Legend missing for y-axis labeling. What exactly is “Fresected”?

Fresected is the fraction of resected DNA and is the unit used to measure how much resected DNA is measured through the restriction enzyme based qPCR assay we used as illustrated in Figure 5A and described in Gnugge et al. Processing the DNA Double-Strand Breaks in Yeast Methods Enzymology [26].

>Fine<

19. Fig 7: Panel labeling does not match the figure legends.

Updated legend.

>Fine<

20. Fig. 8C is not referenced in the text.

Figure 8C has been moved to Figure S8A and is referenced in the text along with Figure S8B to describe the series of strains with the donor for homologous recombination moved to different locations in the genome.

>Fine – There are no Figure S8A and S8B; only Figure S8, but Figures S9A and B show the series of strains which are mentioned here. Figure S8 and S9A/B are correctly referenced in the main text.<

> New comments:

1. There is a problem with Figure 6I where the 10 kb plot is shown twice.

Sorry for this mistake. The 10 kb and 5 kb plot has been added and moved to Figure 7D.

2. Furthermore, in the main text, references do not align in line 257 where Figure 6I is referenced but should say Figure 6H. In line 263 it should say Figure 6I instead of Figure 6J and it seems that all consecutive references for Figure 6 are misaligned plus there is a reference to Figure 6M in line 266 where the corresponding Figure does not exist.<

Figure 6 was split into 2 figures: Figure 6 and Figure 7.

1. Peng, A.Y.T., J. Li, and B.C. Freeman, *Nuclear Type I Myosins are Essential for Life and Genome Organization*. bioRxiv, 2024.
2. Hetrick, B., et al., *Small molecules CK-666 and CK-869 inhibit actin-related protein 2/3 complex by blocking an activating conformational change*. Chem Biol, 2013. **20**(5): p. 701-12.
3. Nolen, B.J., et al., *Characterization of two classes of small molecule inhibitors of Arp2/3 complex*. Nature, 2009. **460**(7258): p. 1031-4.